# Relationship between cognitive abilities and mental health as represented by cognitive abilities at the neural and genetic levels of analysis

Yue Wang[1], Richard Anney[2], Narun Pat[1]*

[1]Department of Psychology, University of Otago, Dunedin, New Zealand; [2]MRC Centre for Neuropsychiatric Genetics and Genomics, Division of Psychological Medicine and Clinical Neurosciences, School of Medicine and Wolfson Centre for Young People's Mental Health, Cardiff University, Cardiff, United Kingdom

## eLife Assessment

This **important** study examines the relationship between cognition and mental health and investigates how brain, genetics, and environmental measures mediate that relationship. The methods and results are **compelling** and well-executed. Overall, this study will be of interest in the field of population neuroscience and in studies of mental health.

*For correspondence:
narun.pat@otago.ac.nz

**Abstract** Cognitive abilities are closely tied to mental health from early childhood. This study explores how neurobiological units of analysis of cognitive abilities—multimodal neuroimaging and polygenic scores (PGS)—represent this connection. Using data from over 11,000 children (ages 9–10) in the Adolescent Brain Cognitive Development (ABCD) Study, we applied multivariate models to predict cognitive abilities from mental health, neuroimaging, PGS, and environmental factors. Neuroimaging included 45 MRI-derived features (e.g. task/resting-state fMRI, structural MRI, diffusion imaging). Environmental factors encompassed socio-demographics (e.g. parental income/education), lifestyle (e.g. sleep, extracurricular activities), and developmental adverse events (e.g. parental use of alcohol/tobacco, pregnancy complications). Cognitive abilities were predicted by mental health (r = 0.36), neuroimaging (r = 0.54), PGS (r = 0.25), and environmental factors (r = 0.49). Commonality analyses showed that neuroimaging (66%) and PGS (21%) explained most of the cognitive–mental health link. Environmental factors accounted for 63% of the cognitive–mental health link, with neuroimaging and PGS explaining 58% and 21% of this environmental contribution, respectively. These patterns remained consistent over two years. Findings highlight the importance of neurobiological units of analysis for cognitive abilities in understanding the cognitive–mental health connection and its overlap with environmental factors.

## Introduction

Cognitive abilities across various domains, such as attention, working memory, declarative memory, verbal fluency, and cognitive control, are often altered in several psychiatric disorders (*Millan et al., 2012*). This is evident in recent meta-analyses of case-control studies involving patients with mood and anxiety disorders, obsessive-compulsive disorder, posttraumatic stress disorder, and attention-deficit/hyperactivity disorder (ADHD), among others (*Abramovitch et al., 2021*; *East-Richard et al., 2020*). Beyond typical case-control studies, the association between cognitive abilities and mental

health is also observed when mental health varies from normal to abnormal in normative samples (*Morris et al., 2022*). For instance, our study *Pat et al., 2022a* found an association between cognitive abilities and mental health in a relatively large, non-referred sample of 9–10 year-old children from the ABCD study (*Casey et al., 2018*). In this study, we measured cognitive abilities using behavioural performance across cognitive tasks (*Luciana et al., 2018*) while measuring mental health using a broad range of emotional and behavioural problems (*Achenbach et al., 2017*). Thus, cognitive abilities are frequently considered crucial for understanding mental health issues throughout life, beginning in childhood (*Abramovitch et al., 2021*; *Hankin et al., 2016*; *Morris and Cuthbert, 2012*).

According to the National Institute of Mental Health's Research Domain Criteria (RDoC) framework (*Insel et al., 2010*), cognitive abilities should be investigated not only behaviourally but also neurobiologically, from the brain to genes. It remains unclear to what extent the relationship between cognitive abilities and mental health is represented in part by different neurobiological units of analysis -- such as neural and genetic levels measured by multimodal neuroimaging and polygenic scores (PGS). To fully comprehend the role of neurobiology in the relationship between cognitive abilities and mental health, we must also consider how these neurobiological units capture variations due to environmental factors, such as socio-demographics, lifestyles, and childhood developmental adverse events (*Morris et al., 2022*). Our study investigated the extent to which (a) environmental factors explain the relationship between cognitive abilities and mental health, and (b) cognitive abilities at the neural and genetic levels capture these associations due to environmental factors. Specifically, we conducted these investigations in a large normative group of children from the ABCD study (*Casey et al., 2018*). We chose to examine children because, while their emotional and behavioural problems might not meet full diagnostic criteria (*Kessler et al., 2007*), issues at a young age often forecast adult psychopathology (*Reef et al., 2010*; *Roza et al., 2003*). Moreover, the associations among different emotional and behavioural problems in children reflect transdiagnostic dimensions of psychopathology (*Michelini et al., 2019*; *Pat et al., 2022a*), making children an appropriate population to study the transdiagnostic aetiology of mental health, especially within a framework that emphasises normative variation from normal to abnormal, such as the RDoC (*Morris et al., 2022*).

Recently, several neuroscientists have developed predictive models using neuroimaging data from brain magnetic resonance imaging (MRI) of various modalities in the so-called Brain-Wide Association Studies (BWAS) (*Marek et al., 2022*; *Sui et al., 2020*). BWAS aims to create models from MRI data that can accurately predict behavioural phenotypes in participants not included in the model-building process (*Dadi et al., 2021*). In one of the most extensive BWAS benchmarks to date, *Marek et al., 2022* concluded, 'More robust BWAS effects were detected for functional MRI (versus structural), cognitive tests (versus mental health questionnaires), and multivariate methods (versus univariate).' This benchmark has significant implications for using neuroimaging as a neural unit of analysis for cognitive abilities. First, while current BWAS may not be robust enough to predict mental health directly, it is more suitable for predicting cognitive abilities (see *Zhi et al., 2024* for a similar conclusion). This aligns with the Research Domain Criteria (RDoC) framework, which emphasises neurobiological units of analysis for *functional domains*, such as cognitive abilities, rather than mental health itself (*Cuthbert and Insel, 2013*). RDoC's functional domains capture basic human functioning and include cognitive abilities along with negative/positive valence, arousal, and regulation, and social and sensory processes (*Morris et al., 2022*). Accordingly, the current study conducted BWAS to capture cognitive abilities rather than mental health.

The second implication of *Marek et al., 2022* benchmark is the support it provides for using multivariate algorithms, which draw MRI information simultaneously across regions/voxels, over massively univariate algorithms that draw data from one area/voxel at a time. Similar to *Marek et al., 2022* study, which focused on resting-state functional MRI (rs-fMRI), our recent study on task-fMRI also found that multivariate algorithms performed superiorly, up to several folds, in predicting cognitive abilities compared to massively univariate algorithms (*Pat et al., 2023*). The third implication is that the performance of neuroimaging in predicting cognitive abilities depends on MRI modalities. Previous research has used brain MRI data of different modalities to predict cognitive abilities (*Vieira et al., 2022*). For instance, many studies have used rs-fMRI, which reflects functional connectivity between regions during rest (*Dubois et al., 2018*; *Keller et al., 2023*; *Rasero et al., 2021*; *Sripada et al., 2020*; *Sripada et al., 2021*). Others have utilised structural MRI (sMRI), which reflects anatomical morphology based on thickness, area, and volume in cortical/subcortical areas, and diffusion tensor

imaging (DTI), which reflects diffusion distribution within white matter tracts (*Mihalik et al., 2019*; *Rasero et al., 2021*). While less common, task-fMRI, which reflects blood-oxygen-level-dependent (BOLD) activity relevant to each task condition, shows relatively good predictive performance, especially from specific contrasts, such as the 2-Back vs 0-Back from the N-Back working-memory task (*Barch et al., 2013*) nor (*Makowski et al., 2024*; *Pat et al., 2023*; *Pat et al., 2022b*; *Sripada et al., 2020*; *Tetereva et al., 2022*; *Zhao et al., 2023*). A recent meta-analysis estimated the performance of multivariate methods in predicting cognitive abilities from MRI of different modalities at around an out-of-sample *r* of 0.42 (*Vieira et al., 2022*). However, we and others found that this predictive performance could be further boosted by drawing information across different MRI modalities, rather than relying on only one modality (*Pat et al., 2022b*; *Rasero et al., 2021*; *Tetereva et al., 2022*; *Tetereva and Pat, 2024*). Therefore, the current study used opportunistic stacking (*Engemann et al., 2020*; *Pat et al., 2022b*). This multivariate modelling technique allowed us to combine information across MRI modalities with the added benefit of handling missing values. With opportunistic stacking, we created a 'proxy' measure of cognitive abilities (i.e. predicted value from the model) at the neural unit of analysis using multimodal neuroimaging.

Geneticists, like neuroscientists, have conducted Genome-Wide Association Studies (GWAS) to explore the links between single-nucleotide polymorphisms (SNPs) and various behavioural phenotypes (*Bogdan et al., 2018*). Similar to BWAS, GWAS can develop predictive models from genetic profiles, resulting in polygenic scores (PGS) that predict behavioural phenotypes in participants not included in the model-building process (*Choi et al., 2020*). Several large-scale GWAS on cognitive abilities have been conducted, with some studies involving over 250,000 participants (*Davies et al., 2018*; *Lee et al., 2018*; *Savage et al., 2018*). Recently, researchers have used these large-scale GWAS to compute PGS for cognitive abilities and applied these scores to predict cognitive abilities in children (*Allegrini et al., 2019*; *Pat et al., 2022b*). For example, *Allegrini et al., 2019* found that PGS based on Savage et al.'s (2018) GWAS accounted for approximately 5.3% of the variance in cognitive abilities among 12-year-old children. The current study adopted this approach with children of a similar age in the ABCD study, creating a proxy measure of cognitive abilities at the genetic unit of analysis using PGS.

Environmental factors, broadly defined, significantly influence cognitive abilities (*Duyme et al., 1999*; *Pietschnig and Voracek, 2015*). A classic example is the Flynn Effect (*Flynn, 1984*; *Flynn, 2009*; *Rundquist, 1936*; *Williams, 2013*), which describes the observed rise in cognitive abilities, as measured by various cognitive tasks, across generations in the general population over time, particularly in high-income countries during the 20th century (*Pietschnig and Voracek, 2015*; *Trahan et al., 2014*; *Wongupparaj et al., 2017*). Experts attribute the Flynn Effect to environmental factors such as improved living standards and better education (*Baker et al., 2015*; *Rindermann et al., 2017*). Recently, researchers have used multivariate algorithms to create proxy measures of cognitive abilities in children based on environmental factors, similar to approaches used in neuroimaging and polygenic scores (PGS) (*Kirlic et al., 2021*; *Pat et al., 2022b*). These environmental factors often include socio-demographic variables (e.g., parental income/education, area deprivation index, parental marital status), lifestyle factors (e.g. screen/video game use, extracurricular activities), and developmental adverse events (e.g. parental use of alcohol/tobacco before and after pregnancy, birth complications). Studies, including ours, *Kirlic et al., 2021*; *Pat et al., 2022b* have applied multivariate algorithms to predict cognitive abilities from various environmental factors in the ABCD study (*Casey et al., 2018*). In these predictive models, parental income/education, area deprivation index, and extracurricular activities are particularly important predictors of cognitive abilities (*Kirlic et al., 2021*; *Pat et al., 2022b*). Following this approach, the current study created another proxy measure of cognitive abilities based on socio-demographics, lifestyles, and developmental adverse events.

In this study, inspired by RDoC (*Insel et al., 2010*), we (a) focused on cognitive abilities as a functional domain, (b) created predictive models to capture the continuous individual variation (as opposed to distinct categories) in cognitive abilities, (c) computed two neurobiological units of analysis of cognitive abilities: multimodal neuroimaging and PGS, and (d) investigated the potential contributions of environmental factors. To operationalise cognitive abilities, we estimated a latent variable representing behavioural performance across various cognitive tasks, commonly referred to as general cognitive ability or the g-factor (*Deary, 2012*). The g-factor was computed from various cognitive tasks pertinent to RDoC constructs, including attention, working memory, declarative memory, language, and

cognitive control. However, using the g-factor to operationalise cognitive abilities caused this study to diverge from the original conceptualisation of RDoC, which emphasises studying separate constructs within cognitive abilities (*Morris et al., 2022*; *Morris and Cuthbert, 2012*). Recent studies suggest that including a general factor, such as the g-factor, in the model, rather than treating each construct separately, improved model fit (*Beam et al., 2021*; *Quah et al., 2025*). The g-factor in children is also longitudinally stable and can forecast future health outcomes (*Calvin et al., 2017*; *Deary et al., 2013*). Notably, our previous research found that neuroimaging predicts the g-factor more accurately than predicting performance from separate individual cognitive tasks (*Pat et al., 2023*). Accordingly, we decided to conduct predictive models on the g-factor while keeping the RDoC's holistic, neurobiological, and basic-functioning characteristics.

Using the ABCD study (*Casey et al., 2018*), we first developed predictive models to estimate the cognitive abilities of unseen children based on their mental health. These models enabled us to quantify the relationship between cognitive abilities and mental health, thereby creating a proxy measure of cognitive abilities derived from mental health data. The mental health variables included children's emotional and behavioural problems (*Achenbach et al., 2017*) and temperaments, such as behavioural inhibition/activation (*Carver and White, 1994*) and impulsivity (*Zapolski et al., 2010*). These temperaments are linked to externalising and internalising aspects of mental health and are associated with disorders like depression, anxiety, and substance use (*Carver and Johnson, 2018*; *Johnson et al., 2003*). Next, we built predictive models of cognitive abilities using neuroimaging, polygenic scores (PGS), and socio-demographic, lifestyle, and developmental adverse event data, resulting in various proxy measures of cognitive abilities. For neuroimaging, we included 45 types of brain MRI data from task-fMRI, rs-fMRI, sMRI, and DTI. For PGS, we used three definitions of cognitive abilities based on previous large-scale GWAS (*Davies et al., 2018*; *Lee et al., 2018*; *Savage et al., 2018*). For socio-demographic, lifestyle, and developmental adverse events, we included 44 features, covering variables such as parental income/education, screen use, and birth/pregnancy complications. Finally, we conducted a series of commonality analyses (*Nimon et al., 2008*) using these proxy measures of cognitive abilities to address three specific questions. First, we examined the extent to which the relationship between cognitive abilities and mental health was represented in part by cognitive abilities at the neural and genetic levels, as measured by multimodal neuroimaging and PGS, respectively. Second, we assessed the extent to which this relationship was partly explained by environmental factors, as measured by socio-demographic, lifestyle, and developmental adverse events. Third, we tested whether the two neurobiological units of analysis for cognitive abilities, measured by multimodal neuroimaging and PGS, could account for the variance due to environmental factors. To ensure the stability of our results, we repeated the analyses at two time points (ages 9–10 and 11–12).

## Results

### Predictive modelling

#### Predicting cognitive abilities from mental health

*Figure 1a* and *Table 1* illustrate the predictive performance of the Partial Least Square (PLS) models in predicting cognitive abilities from mental health features. These features included: (1) emotional and behavioral problems assessed by the Child Behaviour Checklist (CBCL) (*Achenbach et al., 2017*), and (2) children's temperaments assessed by the Behavioural Inhibition System/Behavioural Activation System (BIS/BAS) (*Carver and White, 1994*) and the Urgency, Premeditation, Perseverance, Sensation seeking, and Positive urgency (UPPS-P) impulsive behaviour scale (*Zapolski et al., 2010*). Using these two sets of mental health features separately resulted in moderate predictive performance, with correlation coefficients ranging from $r=0.24$ to $r=0.31$. Combining them into a single set of features, termed 'mental health,' improved the performance to approximately $r=0.36$, consistent across the two time points.

*Figure 1b* illustrates the loadings and the proportion of variance in cognitive abilities explained by each PLS components. The first PLS component accounted for the highest proportion of variance, ranging from 22.3 to 25.7%. This component was primarily influenced by factors such as attention and social problems, rule-breaking and aggressive behaviours and behavioural activation system drive. A similar pattern was observed across both time points.

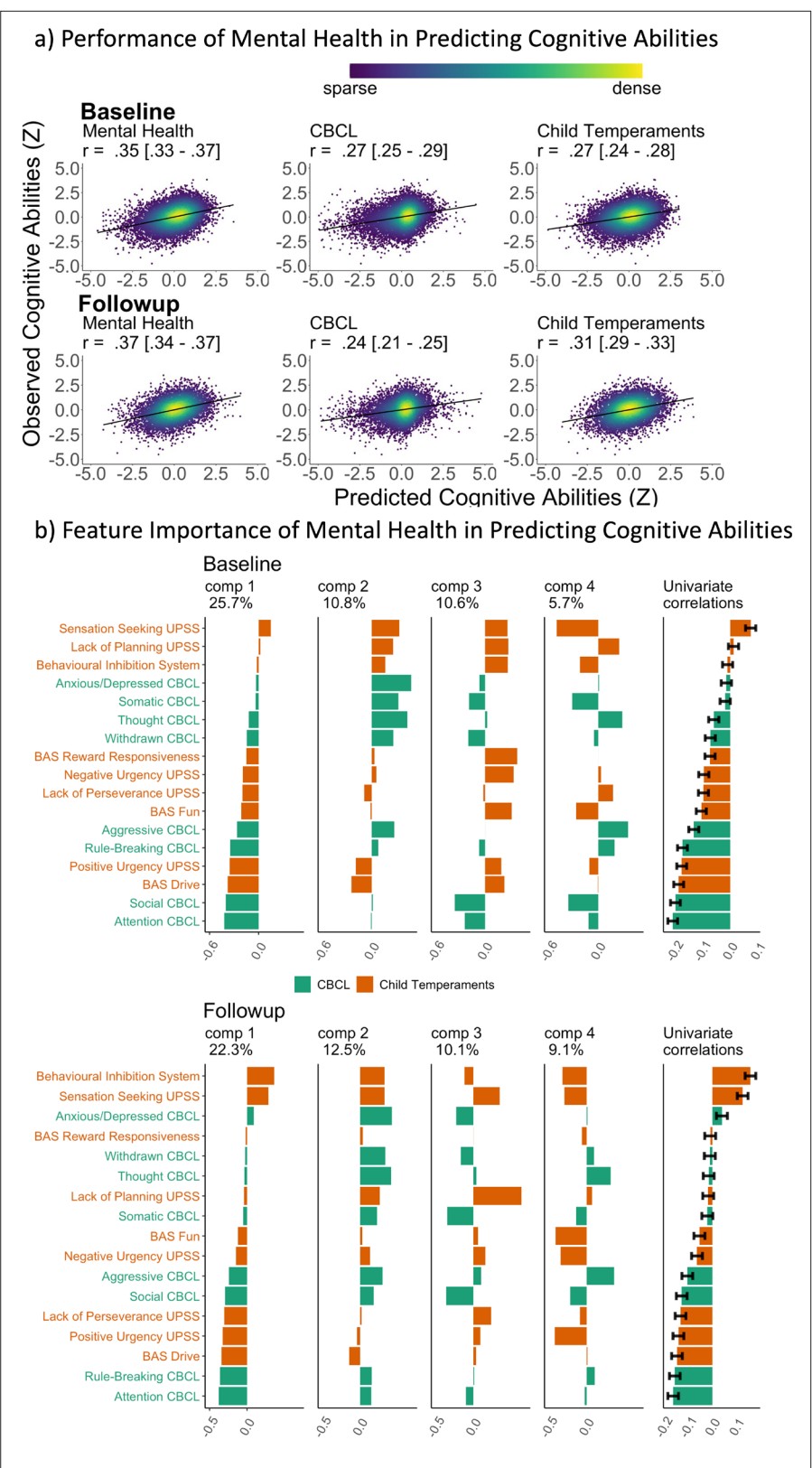

**Figure 1.** Predictive models, predicting cognitive abilities from mental-health features via Partial Least Square (PLS). (**a**) Predictive performance of the models, indicated by scatter plots between observed vs predicted cognitive abilities based on mental health. Cognitive abilities are based on the second-order latent variable, the g-factor, based on a confirmatory factor analysis of six cognitive tasks. All data points are from test sets. *r*

*Figure 1 continued*

is the average Pearson's *r* across 21 test sites. The parentheses following the *r* indicate bootstrapped 95% CIs, calculated based on observed vs predicted cognitive abilities from all test sites combined. UPPS-P Impulsive and Behaviour Scale and the Behavioural Inhibition System/Behavioural Activation System (BIS/BAS) were used for child temperaments, conceptualised as risk factors for mental issues. Mental health includes features from CBCL and child temperaments. (**b**) Feature importance of mental health, predicting cognitive abilities via PLS. The features were ordered based on the loading of the first PLS component. Univariate correlations were Pearson's *r* between each mental-health feature and cognitive abilities. Error bars reflect 95% CIs of the correlations. CBCL = Child Behavioural Checklist (in green), reflecting children's emotional and behavioural problems; UPPS-P = Urgency, Premeditation, Perseverance, Sensation seeking, and Positive urgency Impulsive Behaviour Scale; BAS = Behavioural Activation System (in orange).

## Predicting cognitive abilities from neuroimaging

*Figure 2*, *Figure 2—figure supplements 1 and 2*, and *Tables 1–3* illustrate the predictive performance of the opportunistic stacking models in predicting cognitive abilities from 45 sets of neuroimaging features. The predictive performance of each set of neuroimaging features varied significantly, with correlation coefficients ranging from approximately 0 (ENBack: Negative vs. Neutral Face) to around 0.4 (ENBack: 2-Back vs. 0-Back). Combining information from all 45 sets of neuroimaging features into a stacked model improved the performance to approximately *r*=0.54, consistent across both time points. The stacked model ($R^2 \approx 0.29$) explained almost twice as much variance in cognitive abilities as the model based on the best single set of neuroimaging features (ENBack: 2-Back vs. 0-Back, $R^2 \approx 0.15$). *Figures 2 and 3*, *Figure 3—figure supplements 1–11* highlight the feature importance of the opportunistic stacking models. Across both time points, the top contributing neuroimaging features, as indicated by SHAP values, were ENBack task-fMRI contrasts, rs-fMRI, and cortical thickness.

## Predicting cognitive abilities from polygenic scores

*Figure 2a* and *Table 1* illustrate the predictive performance of the Elastic Net models in predicting cognitive abilities using three polygenic scores (PGSs). The predictive accuracy of these PGSs was *r*=0.25 at baseline and *r*=0.25 at follow-up. (*Figure 2c*) highlights the feature importance within these models, indicating a stronger contribution from the PGS based on *Savage et al., 2018* GWAS.

**Table 1.** Performance metrics for predictive models, predicting cognitive abilities from mental health, neuroimaging, polygenic scores, and socio-demographics, lifestyles, and developments.
The metrics were averaged across test sites with standard deviations in parentheses.

| Features | Correlation | $R^2$ | MAE | RMSE |
|---|---|---|---|---|
| Baseline | | | | |
| Mental Health | 0.353 (0.051) | 0.124 (0.038) | 0.736 (0.019) | 0.934 (0.02) |
| CBCL | 0.272 (0.048) | 0.074 (0.028) | 0.758 (0.014) | 0.961 (0.015) |
| Child personality | 0.268 (0.058) | 0.071 (0.034) | 0.759 (0.019) | 0.962 (0.017) |
| Neuroimaging | 0.539 (0.073) | 0.291 (0.082) | 0.658 (0.039) | 0.839 (0.05) |
| Polygenic scores | 0.252 (0.056) | 0.02 (0.075) | 0.696 (0.055) | 0.884 (0.066) |
| Socio-demo Life Dev Adv | 0.486 (0.081) | 0.239 (0.084) | 0.686 (0.041) | 0.87 (0.049) |
| Follow-up | | | | |
| Mental Health | 0.36 (0.07) | 0.116 (0.061) | 0.715 (0.043) | 0.903 (0.051) |
| CBCL | 0.24 (0.056) | 0.043 (0.034) | 0.746 (0.045) | 0.94 (0.053) |
| Child personality | 0.311 (0.076) | 0.084 (0.059) | 0.728 (0.046) | 0.919 (0.051) |
| Neuroimaging | 0.524 (0.097) | 0.266 (0.112) | 0.645 (0.038) | 0.818 (0.053) |
| Polygenic scores | 0.25 (0.075) | 0.031 (0.068) | 0.672 (0.053) | 0.854 (0.068) |
| Socio-demo Life Dev Adv | 0.488 (0.093) | 0.226 (0.096) | 0.664 (0.044) | 0.843 (0.05) |

$R^2$=coefficient of determination; MAE = mean-absolute error; RMSE = root mean square error.

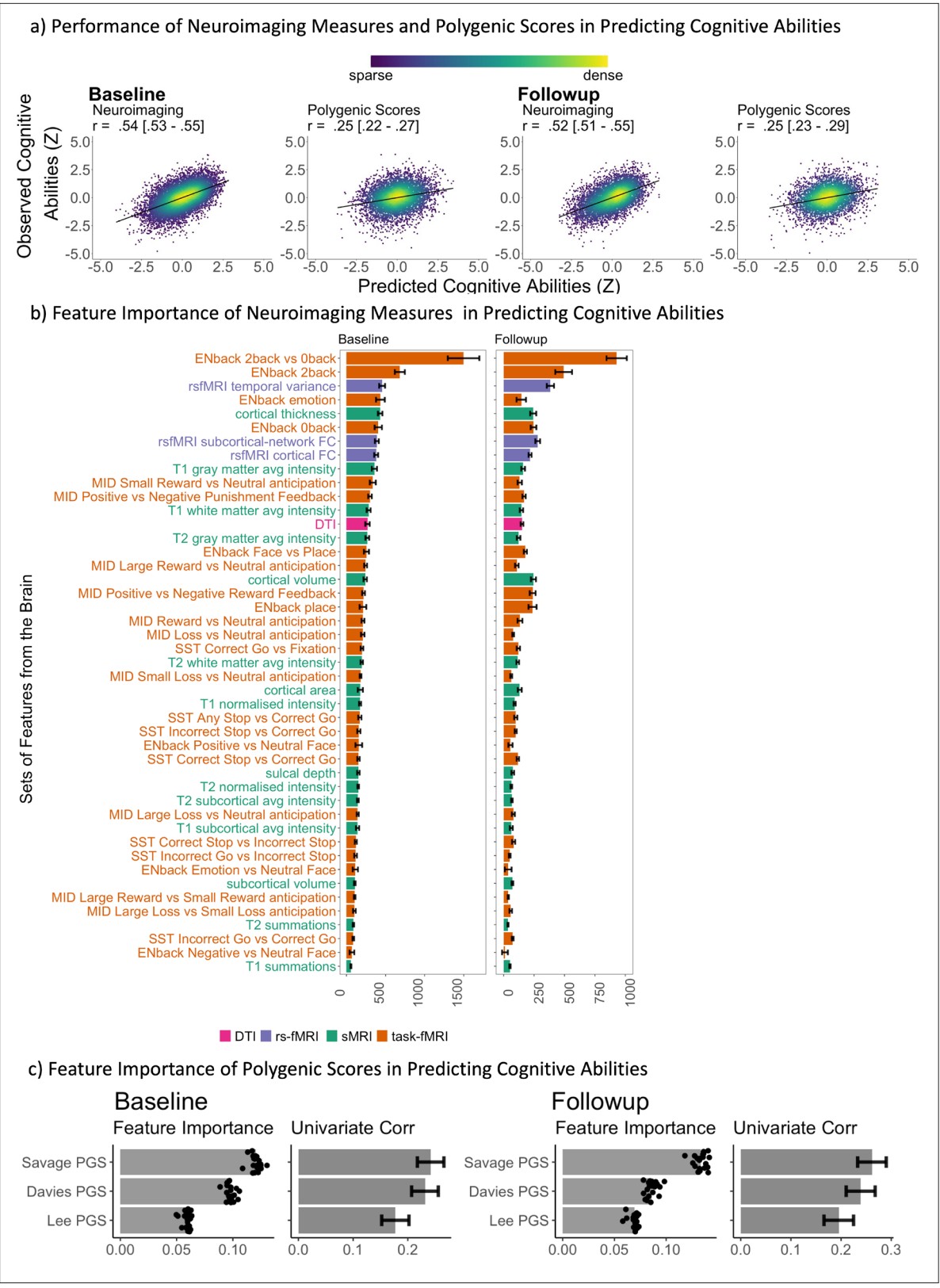

**Figure 2.** Predictive models predicting cognitive abilities from neuroimaging via opportunistic stacking and polygenic scores via Elastic Net. (**a**) Scatter plots between observed vs predicted cognitive abilities based on neuroimaging and polygenic scores. Cognitive abilities are based on the second-order latent variable, the g-factor, based on a confirmatory factor analysis of six cognitive tasks. The parentheses following the *r* indicate the bootstrapped 95% CIs, calculated based on observed vs predicted cognitive abilities from all test sites combined. All data points are from test sets. *r* is

*Figure 2 continued*

the average Pearson's *r* across 21 test sites. The parentheses following the *r* indicate bootstrapped 95% CIs, calculated based on observed vs predicted cognitive abilities from all test sites combined. (**b**) Feature importance of the stacking layer of neuroimaging, predicting cognitive abilities via Random Forest. For the stacking layer of neuroimaging, the feature importance was based on the absolute value of SHapley Additive exPlanations (SHAP), averaged across test sites. A higher absolute value of SHAP indicates a higher contribution to the prediction. Error bars reflect standard deviations across sites. Different sets of neuroimaging features were filled with different colours: pink for dMRI, orange for fMRI, purple for resting-state functional MRI (rsMRI), and green for structural MRI (sMRI). (**c**) Feature importance of polygenic scores, predicting cognitive abilities via Elastic Net. For polygenic scores, the feature importance was based on the Elastic Net coefficients, averaged across test sites. We also plotted Pearson's correlations between each polygenic score and cognitive abilities computed from the full data. Error bars reflect 95% CIs of these correlations.

The online version of this article includes the following figure supplement(s) for figure 2:

**Figure supplement 1.** Scatter plots between observed vs predicted cognitive abilities based on each set of 45 neuroimaging features in the baseline data.

**Figure supplement 2.** Scatter plots between observed vs predicted cognitive abilities based on each set of 45 neuroimaging features in the follow-up data.

## Predicting cognitive abilities from socio-demographics, lifestyles, and developmental adverse events

*Figure 4a* and *Table 1* illustrate the predictive performance of the PLS models in predicting cognitive abilities from socio-demographics, lifestyles, and developmental adverse events. Using 44 features covering these areas, the predictive performance was around *r*=0.49, consistent across the two time points. (*Figure 4b*) shows the loadings and the proportion of variance explained by these PLS models. The first PLS component accounted for the highest proportion of variance (around 10%).

Based on its loadings, this first component was: (a) Positively influenced by features such as parental income and education, neighbourhood safety, and extracurricular activities, (b) Negatively influenced by features such as area deprivation, having a single parent, screen use, economic insecurities, lack of sleep, playing mature video games, watching mature movies, and lead exposure.

## Commonality analyses

We separately conducted the four sets of commonality analyses.

## Commonality analyses for proxy measures of cognitive abilities based on mental health and neuroimaging

At baseline, having both proxy measures based on mental health and neuroimaging in a linear mixed model explained 27% of the variance in cognitive abilities. Specifically, 9.8% of the variance in cognitive abilities was explained by mental health, which included the common effect between the two proxy measures (6.48%) and the unique effect of mental health (3.32%) (see *Tables 4–5* and *Figure 5*). This indicates that 66% of the relationship between cognitive abilities and mental health, i.e., (6.48 ÷ 9.8)×100, was shared with neuroimaging. The common effects varied considerably across different sets of neuroimaging features, ranging from approximately 0.08 to 2.78%, with the highest being the ENBack task fMRI: 2-Back vs. 0-Back (see *Figure 5—figure supplement 1*). The pattern of results was consistent across both time points.

## Commonality analyses for proxy measures of cognitive abilities based on mental health and PGSs

At baseline, having both proxy measures based on mental health and PGSs in a linear mixed model explained 11.8% of the variance in cognitive abilities. Specifically, 9.21% of the variance in cognitive abilities was explained by mental health, which included the common effect between the two proxy measures (1.93%) and the unique effect of mental health (7.28%) (see *Tables 6–7* and *Figure 5*). This indicates that 21% of the relationship between cognitive abilities and mental health, i.e., (1.93 ÷ 9.21) × 100, was shared with PGSs. The pattern of results was consistent across both time points.

**Table 2.** Performance metrics for predictive models, predicting cognitive abilities from the 45 sets of neuroimaging features in the baseline data.

The metrics were averaged across test sites with standard deviations in parentheses.

| Features | Correlation | $R^2$ | MAE | RMSE |
|---|---|---|---|---|
| Neuroimaging | 0.539 (0.073) | 0.291 (0.082) | 0.658 (0.039) | 0.839 (0.05) |
| ENback 2back vs 0back | 0.393 (0.048) | 0.147 (0.042) | 0.661 (0.038) | 0.841 (0.045) |
| ENback 2back | 0.367 (0.06) | 0.128 (0.048) | 0.667 (0.036) | 0.848 (0.043) |
| rsfMRI temporal variance | 0.3 (0.094) | 0.09 (0.054) | 0.728 (0.04) | 0.921 (0.045) |
| rsfMRI cortical FC | 0.299 (0.055) | 0.088 (0.034) | 0.734 (0.027) | 0.929 (0.032) |
| ENback emotion | 0.277 (0.06) | 0.07 (0.041) | 0.689 (0.031) | 0.876 (0.035) |
| Cortical thickness | 0.265 (0.1) | 0.072 (0.055) | 0.756 (0.026) | 0.96 (0.03) |
| T2 gray matter avg intensity | 0.264 (0.106) | 0.069 (0.064) | 0.752 (0.032) | 0.953 (0.035) |
| T1 gray matter avg intensity | 0.263 (0.103) | 0.063 (0.071) | 0.761 (0.033) | 0.965 (0.039) |
| ENback 0back | 0.261 (0.058) | 0.061 (0.038) | 0.688 (0.031) | 0.878 (0.035) |
| T1 white matter avg intensity | 0.26 (0.103) | 0.067 (0.063) | 0.76 (0.029) | 0.963 (0.035) |
| rsfMRI subcortical-network FC | 0.258 (0.083) | 0.066 (0.043) | 0.743 (0.033) | 0.94 (0.035) |
| ENback place | 0.239 (0.065) | 0.049 (0.041) | 0.695 (0.032) | 0.886 (0.038) |
| T2 white matter avg intensity | 0.238 (0.103) | 0.056 (0.056) | 0.756 (0.03) | 0.96 (0.031) |
| T2 normalised intensity | 0.236 (0.082) | 0.057 (0.041) | 0.755 (0.021) | 0.96 (0.024) |
| DTI | 0.23 (0.074) | 0.042 (0.048) | 0.762 (0.027) | 0.967 (0.029) |
| Cortical volume | 0.228 (0.095) | 0.053 (0.044) | 0.767 (0.02) | 0.971 (0.024) |
| MID Small Rew vs Neu anticipation | 0.223 (0.049) | 0.048 (0.022) | 0.743 (0.017) | 0.938 (0.02) |
| Cortical area | 0.218 (0.101) | 0.049 (0.046) | 0.768 (0.021) | 0.973 (0.025) |
| T1 normalised intensity | 0.215 (0.109) | 0.047 (0.049) | 0.769 (0.022) | 0.974 (0.028) |
| MID Reward vs Neutral anticipation | 0.214 (0.062) | 0.043 (0.028) | 0.745 (0.022) | 0.944 (0.024) |
| MID Loss vs Neutral anticipation | 0.214 (0.075) | 0.043 (0.034) | 0.745 (0.025) | 0.944 (0.028) |
| MID Small Loss vs Neu anticipation | 0.203 (0.073) | 0.038 (0.03) | 0.747 (0.026) | 0.945 (0.026) |
| MID Pos vs Neg Punishment Feedback | 0.202 (0.066) | 0.037 (0.027) | 0.745 (0.021) | 0.945 (0.026) |
| T1 subcortical avg intensity | 0.2 (0.087) | 0.037 (0.043) | 0.773 (0.023) | 0.979 (0.026) |
| MID Large Rew vs Neu anticipation | 0.2 (0.072) | 0.037 (0.03) | 0.747 (0.021) | 0.946 (0.024) |
| MID Pos vs Neg Reward Feedback | 0.198 (0.05) | 0.036 (0.02) | 0.748 (0.022) | 0.945 (0.028) |
| T1 summations | 0.196 (0.08) | 0.009 (0.059) | 0.784 (0.029) | 0.992 (0.033) |
| Sulcal depth | 0.18 (0.095) | 0.032 (0.039) | 0.777 (0.02) | 0.984 (0.026) |
| MID Large Loss vs Neu anticipation | 0.173 (0.066) | 0.026 (0.026) | 0.749 (0.022) | 0.95 (0.025) |
| subcortical volume | 0.17 (0.078) | 0.028 (0.029) | 0.775 (0.018) | 0.982 (0.021) |
| SST Any Stop vs Correct Go | 0.164 (0.065) | 0.022 (0.025) | 0.736 (0.038) | 0.935 (0.043) |
| T2 subcortical avg intensity | 0.158 (0.057) | 0.023 (0.023) | 0.77 (0.018) | 0.977 (0.02) |
| ENback Face vs Place | 0.148 (0.076) | 0.014 (0.028) | 0.712 (0.027) | 0.904 (0.034) |
| SST Incorrect Stop vs Correct Go | 0.147 (0.059) | 0.017 (0.02) | 0.738 (0.035) | 0.937 (0.04) |
| SST Correct Stop vs Correct Go | 0.145 (0.056) | 0.017 (0.018) | 0.739 (0.033) | 0.936 (0.038) |
| SST Correct Go vs Fixation | 0.145 (0.053) | 0.017 (0.017) | 0.74 (0.033) | 0.938 (0.036) |
| MID Large Rew vs Small anticipation | 0.133 (0.05) | 0.015 (0.014) | 0.757 (0.022) | 0.956 (0.025) |
| T2 summations | 0.114 (0.053) | 0.008 (0.022) | 0.777 (0.018) | 0.984 (0.016) |

*Table 2 continued on next page*

*Table 2 continued*

| Features | Correlation | $R^2$ | MAE | RMSE |
|---|---|---|---|---|
| SST Incorrect Go vs Correct Go | 0.11 (0.061) | 0.008 (0.015) | 0.744 (0.034) | 0.94 (0.038) |
| SST Correct Stop vs Incorrect Stop | 0.096 (0.068) | 0.005 (0.018) | 0.744 (0.033) | 0.943 (0.036) |
| MID Large vs Small Loss anticipation | 0.093 (0.063) | 0.006 (0.014) | 0.756 (0.024) | 0.96 (0.026) |
| SST Incorrect Go vs Incorrect Stop | 0.061 (0.039) | 0 (0.008) | 0.744 (0.032) | 0.943 (0.036) |
| ENback Positive vs Neutral Face | 0.024 (0.06) | −0.007 (0.012) | 0.716 (0.027) | 0.908 (0.034) |
| ENback Emotion vs Neutral Face | 0.019 (0.058) | −0.007 (0.01) | 0.716 (0.026) | 0.908 (0.033) |
| ENback Negative vs Neutral Face | 0.002 (0.058) | −0.007 (0.009) | 0.718 (0.024) | 0.911 (0.03) |

$R^2$=coefficient of determination; MAE = mean-absolute error; RMSE = root mean square error.

## Commonality analyses for proxy measures of cognitive abilities based on mental health and socio-demographics, lifestyles, and developmental adverse events

At baseline, having both proxy measures based on mental health and socio-demographics, lifestyles, and developmental adverse events in a linear mixed model explained 24.9% of the variance in cognitive abilities. Specifically, 9.75% of the variance in cognitive abilities was explained by mental health, which included the common effect between the two proxy measures (6.12%) and the unique effect of mental health (3.63%) (see *Tables 8–9* and *Figure 5*). This indicates that over 63% of the relationship between cognitive abilities and mental health, i.e., (6.12 ÷ 9.75) × 100, was shared with socio-demographics, lifestyles, and developmental adverse events. The pattern of results was consistent across both time points.

## Commonality analyses for proxy measures of cognitive abilities based on mental health, neuroimaging, PGSs and socio-demographics, lifestyles, and developmental adverse events

At baseline, having all four proxy measures based on mental health, neuroimaging, PGSs, and socio-demographics, lifestyles, and developmental adverse events in a linear mixed model explained 24.2% of the variance in cognitive abilities. Of the 8.97% of the variance in cognitive abilities explained by mental health, 7.05% represented common effects with the other proxy measures. This indicates that 79%, i.e., (7.05 ÷ 8.97) × 100, of the relationship between cognitive abilities and mental health was shared with the three other proxy measures (see *Tables 10–11* and *Figure 5*). Additionally, among the variance that socio-demographics, lifestyles, and developmental adverse events accounted for in the relationship between cognitive abilities and mental health, neuroimaging could capture 58%, while PGSs could capture 21%. The pattern of results was consistent across both time points.

## Discussion

We aim to understand the extent to which the relationship between cognitive abilities and mental health is represented in part by cognitive abilities at the neural and genetic levels of analysis. We began by quantifying the relationship between cognitive abilities and mental health, finding a medium-sized out-of-sample correlation of approximately *r*=0.36. This relationship was shared with neuroimaging (66% at baseline) and PGS (21% at baseline), based on two separate sets of commonality analyses. This suggests the significant roles of these two neurobiological units of analysis in shaping the relationship between cognitive abilities and mental health (*Morris and Cuthbert, 2012*). We also found that the relationship between cognitive abilities and mental health was partly shared with environmental factors, as measured by socio-demographics, lifestyles, and developmental adverse events (63% at baseline). In another set of commonality analysis, this variance due to socio-demographics, lifestyles, and developmental adverse events was explained by neuroimaging and PGS at 58% and 21%, respectively, at baseline. Accordingly, the neurobiological units of analysis for cognitive abilities

**Table 3.** Performance metrics for predictive models, predicting cognitive abilities from the 45 sets of neuroimaging features in the follow-up data.

| Features | Correlation | $R^2$ | MAE | RMSE |
|---|---|---|---|---|
| Neuroimaging | 0.524 (0.097) | 0.266 (0.112) | 0.645 (0.038) | 0.818 (0.053) |
| ENback 2back vs 0back | 0.402 (0.092) | 0.15 (0.075) | 0.671 (0.032) | 0.844 (0.041) |
| ENback 2back | 0.39 (0.083) | 0.14 (0.071) | 0.676 (0.036) | 0.848 (0.045) |
| ENback place | 0.32 (0.073) | 0.089 (0.049) | 0.695 (0.038) | 0.874 (0.047) |
| ENback emotion | 0.319 (0.076) | 0.089 (0.05) | 0.696 (0.04) | 0.876 (0.047) |
| rsfMRI cortical FC | 0.309 (0.093) | 0.081 (0.071) | 0.718 (0.037) | 0.908 (0.046) |
| ENback 0back | 0.299 (0.078) | 0.077 (0.057) | 0.7 (0.045) | 0.881 (0.052) |
| rsfMRI temporal variance | 0.297 (0.111) | 0.077 (0.071) | 0.718 (0.045) | 0.903 (0.052) |
| rsfMRI subcortical-network FC | 0.265 (0.092) | 0.056 (0.059) | 0.732 (0.039) | 0.92 (0.048) |
| Cortical thickness | 0.259 (0.106) | 0.055 (0.062) | 0.738 (0.034) | 0.932 (0.041) |
| Cortical volume | 0.243 (0.091) | 0.046 (0.049) | 0.744 (0.034) | 0.936 (0.039) |
| T1 white matter avg intensity | 0.243 (0.09) | 0.044 (0.057) | 0.742 (0.035) | 0.937 (0.042) |
| T1 gray matter avg intensity | 0.241 (0.105) | 0.04 (0.069) | 0.742 (0.039) | 0.939 (0.047) |
| Cortical area | 0.233 (0.092) | 0.041 (0.05) | 0.746 (0.032) | 0.939 (0.04) |
| T2 gray matter avg intensity | 0.226 (0.112) | 0.04 (0.064) | 0.743 (0.037) | 0.939 (0.049) |
| DTI | 0.218 (0.065) | 0.022 (0.052) | 0.747 (0.034) | 0.944 (0.041) |
| T2 white matter avg intensity | 0.213 (0.099) | 0.033 (0.057) | 0.747 (0.036) | 0.942 (0.045) |
| T1 summations | 0.213 (0.062) | 0.011 (0.046) | 0.756 (0.039) | 0.954 (0.044) |
| MID Pos vs Neg Punish Feedback | 0.208 (0.058) | 0.025 (0.033) | 0.743 (0.044) | 0.933 (0.049) |
| MID Pos vs Neg Reward Feedback | 0.196 (0.071) | 0.021 (0.042) | 0.742 (0.038) | 0.933 (0.042) |
| T2 normalised intensity | 0.195 (0.077) | 0.025 (0.035) | 0.749 (0.039) | 0.946 (0.045) |
| T1 subcortical avg intensity | 0.191 (0.094) | 0.002 (0.083) | 0.759 (0.039) | 0.957 (0.046) |
| sulcal depth | 0.185 (0.087) | 0.018 (0.048) | 0.756 (0.034) | 0.95 (0.043) |
| MID Reward vs Neutral anticipation | 0.185 (0.078) | 0.016 (0.039) | 0.746 (0.037) | 0.937 (0.04) |
| SST Any Stop vs Correct Go | 0.184 (0.079) | 0.018 (0.034) | 0.745 (0.047) | 0.934 (0.054) |
| T1 normalised intensity | 0.181 (0.077) | 0.018 (0.036) | 0.752 (0.038) | 0.95 (0.045) |
| ENback Face vs Place | 0.179 (0.075) | 0.019 (0.03) | 0.721 (0.039) | 0.907 (0.044) |
| subcortical volume | 0.178 (0.062) | 0.016 (0.032) | 0.752 (0.036) | 0.949 (0.041) |
| SST Correct Stop vs Correct Go | 0.175 (0.062) | 0.015 (0.026) | 0.746 (0.048) | 0.936 (0.053) |
| MID Large Rew vs Neu anticipation | 0.172 (0.055) | 0.012 (0.028) | 0.747 (0.04) | 0.939 (0.044) |
| SST Incorrect Stop vs Correct Go | 0.17 (0.085) | 0.015 (0.032) | 0.746 (0.051) | 0.936 (0.059) |
| T2 subcortical avg intensity | 0.157 (0.085) | 0.011 (0.033) | 0.755 (0.039) | 0.952 (0.043) |
| MID Small Rew vs Neu anticipation | 0.154 (0.086) | 0.007 (0.04) | 0.75 (0.04) | 0.941 (0.044) |
| MID Loss vs Neutral anticipation | 0.147 (0.07) | 0.004 (0.024) | 0.75 (0.04) | 0.942 (0.043) |
| SST Correct Go vs Fixation | 0.138 (0.065) | 0.005 (0.026) | 0.749 (0.046) | 0.938 (0.054) |
| SST Incorrect Go vs Correct Go | 0.122 (0.072) | 0.001 (0.03) | 0.752 (0.053) | 0.944 (0.059) |
| MID Large Loss vs Neu anticipation | 0.121 (0.074) | –0.004 (0.03) | 0.752 (0.04) | 0.942 (0.044) |
| T2 summations | 0.116 (0.07) | –0.003 (0.029) | 0.763 (0.041) | 0.96 (0.048) |

*Table 3 continued on next page*

*Table 3 continued*

| Features | Correlation | $R^2$ | MAE | RMSE |
|---|---|---|---|---|
| MID Small Loss vs Neu Anticipation | 0.106 (0.071) | –0.005 (0.021) | 0.755 (0.041) | 0.948 (0.044) |
| SST Correct Stop vs Incorrect Stop | 0.09 (0.086) | –0.006 (0.023) | 0.754 (0.049) | 0.947 (0.057) |
| MID Large vs Small Loss Anticipation | 0.064 (0.07) | –0.012 (0.025) | 0.756 (0.043) | 0.948 (0.048) |
| MID Large vs Small Rew anticipation | 0.063 (0.059) | –0.012 (0.018) | 0.759 (0.042) | 0.952 (0.046) |
| SST Incorrect Go vs Incorrect Stop | 0.038 (0.067) | –0.014 (0.019) | 0.756 (0.052) | 0.95 (0.059) |
| ENback Positive vs Neutral Face | 0.006 (0.069) | –0.013 (0.018) | 0.732 (0.037) | 0.919 (0.044) |
| ENback Negative vs Neutral Face | –0.012 (0.031) | –0.012 (0.015) | 0.735 (0.039) | 0.923 (0.043) |
| ENback Emotion vs Neutral Face | –0.027 (0.067) | –0.014 (0.016) | 0.733 (0.038) | 0.921 (0.045) |

The metrics were averaged across test sites with standard deviations in parentheses. $R^2$=coefficient of determination; MAE = mean-absolute error; RMSE = root mean square error.

captured the environmental factors, consistent with RDoC's viewpoint (*Morris et al., 2022*). Notably, this pattern of results remained stable over two years in early adolescence.

Our predictive modelling revealed a medium-sized predictive relationship between cognitive abilities and mental health. This finding aligns with recent meta-analyses of case-control studies that link cognitive abilities and mental disorders across various psychiatric conditions (*Abramovitch et al., 2021*; *East-Richard et al., 2020*). Unlike previous studies, we estimated the predictive, out-of-sample relationship between cognitive abilities and mental disorders in a large normative sample of children. Although our predictive models, like other cross-sectional models, cannot determine the directionality of the effects, the strength of the relationship between cognitive abilities and mental health estimated here should be more robust than when calculated using the same sample as the model itself, known as in-sample prediction/association (*Marek et al., 2022*; *Yarkoni and Westfall, 2017*). Examining the PLS loadings of our predictive models revealed that the relationship was driven by various aspects of mental health, including thought and externalising symptoms, as well as motivation. This suggests that there are multiple pathways—encompassing a broad range of emotional and behavioural problems and temperaments—through which cognitive abilities and mental health are linked.

Our predictive modelling created proxy measures of cognitive abilities based on two neurobiological units of analysis: neuroimaging and PGS (*Morris and Cuthbert, 2012*). For neuroimaging, inspired by recent BWAS benchmarks (*Engemann et al., 2020*; *Marek et al., 2022*), we used a multivariate modelling technique called opportunistic stacking, which integrates information across various MRI features and modalities. Combining 45 sets of neuroimaging features resulted in relatively high predictive performance (out-of-sample *r*=0.54 at baseline), compared to using any single set. This finding aligns with previous research that pooled multiple neuroimaging modalities (*Engemann et al., 2020*; *Rasero et al., 2021*; *Tetereva et al., 2022*). This level of predictive performance is numerically higher than that found in a recent meta-analysis, which mainly included studies using only one set of neuroimaging features, with an *r* of 0.42 (*Vieira et al., 2022*). Moreover, this performance level in predicting cognitive abilities is nearly the same as our previous attempt using a similar stacking technique to integrate MRI modalities in young adult samples from the Human Connectome Project (HCP) (*Van Essen et al., 2013*), which achieved an out-of-sample *r*=0.57 (*Tetereva et al., 2022*). Similarly, in the current study, the top contributing set of neuroimaging features, the 2-Back vs. 0-Back task fMRI, was consistent with previous studies using the HCP (*Sripada et al., 2020*; *Tetereva et al., 2022*). Altogether, this demonstrates the robustness of our proxy measure of cognitive abilities based on multimodal neuroimaging. In addition to predictive performance, opportunistic stacking offers the added benefit of handling missing values (*Engemann et al., 2020*; *Pat et al., 2022b*), allowing us to retain data from 10,754 participants who completed the cognitive tasks at baseline and has at least one set of neuroimaging features. Consequently, with opportunistic stacking, we were more likely to retain MRI data from participants with higher fMRI noise, such as those with socioeconomic disadvantages (*Cosgrove et al., 2022*). More importantly, we demonstrated that the proxy measure based on multimodal neuroimaging explained the majority of the variance in the relationship between cognitive

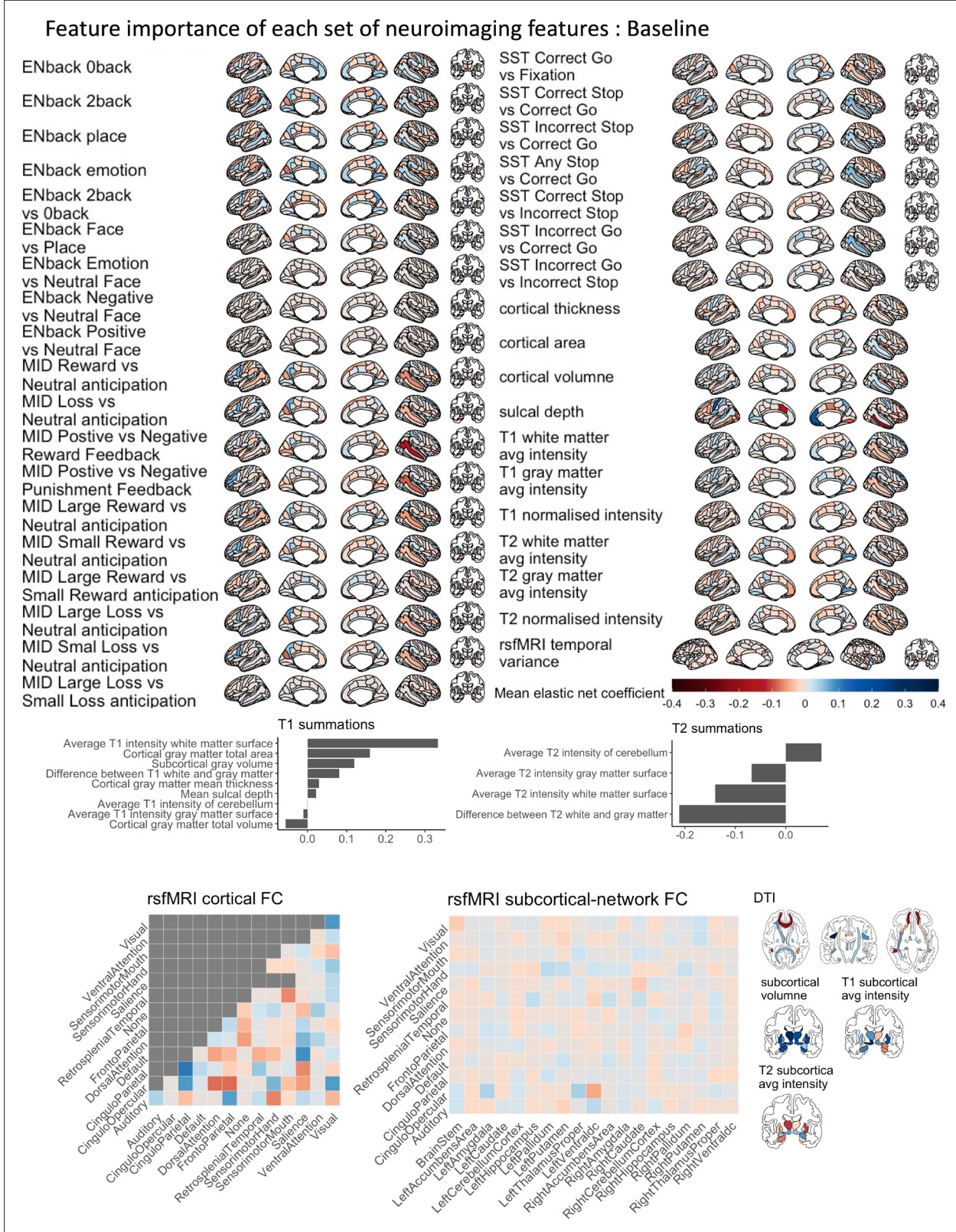

**Figure 3.** Feature importance of each set of neuroimaging features, predicting cognitive abilities in the baseline data. The feature importance was based on the Elastic Net coefficients, averaged across test sites. We did not order these sets of neuroimaging features according to their contribution to the stacking layer (see *Figure 2*). Larger versions of the feature importance for each set of neuroimaging features can be found in *Figure 3—figure supplements 1–11*. MID = Monetary Incentive Delay task; SST = Stop Signal Task; DTI = Diffusion Tensor Imaging; FC = functional connectivity.

*Figure 3 continued on next page*

*Figure 3 continued*

The online version of this article includes the following figure supplement(s) for figure 3:

**Figure supplement 1.** Feature importance of each set of neuroimaging features, predicting cognitive abilities in the follow-up data via Elastic Net.

**Figure supplement 2.** Feature importance of Nback task-fMRI features, predicting cognitive abilities in the baseline data via Elastic Net.

**Figure supplement 3.** Feature importance of MID task-fMRI features, predicting cognitive abilities in the baseline data via Elastic Net.

**Figure supplement 4.** Feature importance of SST task-fMRI features, predicting cognitive abilities in the baseline data via Elastic Net.

**Figure supplement 5.** Feature importance of resting-state functional MRI (rs-fMRI) features, predicting cognitive abilities in the baseline data via Elastic Net.

**Figure supplement 6.** Feature importance of structural MRI (sMRI) and dMRI features, predicting cognitive abilities in the baseline data via Elastic Net.

**Figure supplement 7.** Feature importance of Nback task-fMRI features, predicting cognitive abilities in the follow-up data via Elastic Net.

**Figure supplement 8.** Feature importance of monetary incentive delay (MID) task-fMRI features, predicting cognitive abilities in the follow-up data via Elastic Net.

**Figure supplement 9.** Feature importance of SST task-fMRI features, predicting cognitive abilities in the follow-up data via Elastic Net.

**Figure supplement 10.** Feature importance of resting-state functional MRI (rs-fMRI) features, predicting cognitive abilities in the follow-up data via Elastic Net.

**Figure supplement 11.** Feature importance of structural MRI (sMRI) and dMRI features, predicting cognitive abilities in the follow-up data via Elastic Net.

abilities and mental health, underscoring its significant role as a neurobiological unit of analysis for cognitive abilities (*Morris and Cuthbert, 2012*).

For PGS, we created a proxy measure based on three large-scale GWAS on cognitive abilities (*Davies et al., 2018*; *Lee et al., 2018*; *Savage et al., 2018*). Using PGS resulted in a numerically weaker predictive performance (out-of-sample $r$=0.25 at baseline) compared to multimodal neuroimaging. However, this predictive strength is still comparable to previous research. For instance, *Allegrini et al., 2019* used a different cohort of children and found $R^2$=0.053 when applying PGS based on *Savage et al., 2018* to predict the cognitive abilities of 12-year-old children. Given that PGS based on *Savage et al., 2018* also drove the prediction in the current study, as seen in its feature importance, this similar level of predictive performance between *Allegrini et al., 2019* and our study suggests consistency in the predictive performance of PGS. Despite this level of performance, PGS was able to explain some variance (21% at baseline) in the relationship between cognitive abilities and mental health, indicating some capacity of PGS as a neurobiological unit of analysis for cognitive abilities.

There are multiple potential reasons why PGS performed much poorer than multimodal neuroimaging. First, unlike genes, the brain changes throughout development and lifespan (*Bethlehem et al., 2022*), and so do cognitive abilities (*Hartshorne and Germine, 2015*). This dynamic nature might make multimodal neuroimaging a better tool for tracing cognitive abilities. Second, there might be a mismatch in the age of participants between the original GWAS (*Davies et al., 2018*; *Lee et al., 2018*; *Savage et al., 2018*) and the current study. While the original GWAS conducted meta-analyses pooling data from participants aged 5–102, these studies might draw more heavily from older cohorts with large participant numbers, such as the UK Biobank (*Sudlow et al., 2015*). *Allegrini et al., 2019* also demonstrated that PGS performs better in predicting cognitive abilities in older children (aged 16) compared to younger ones (aged 12). Therefore, a more child-specific PGS might be needed to explain more variance in children. Thirdly, the PGS used here included only common SNPs and not rare variants. Recent studies using whole-genome sequence data have found that rare variants contribute to the heritability of complex traits, such as height and body mass index (*Wainschtein et al., 2022*). Given that cognitive abilities are also complex traits, future studies might need to examine if including rare variants can improve the predictive performance of PGS.

Similarly, our predictive modelling created proxy measures of cognitive abilities for environmental factors based on socio-demographics, lifestyles, and developmental adverse events. In line with previous work (*Kirlic et al., 2021*; *Pat et al., 2022b*), we could predict unseen children's cognitive abilities based on their socio-demographics, lifestyles, and developmental adverse events with a medium-to-high out-of-sample $r$=0.49 (at baseline). This prediction was driven more strongly by socio-demographics (e.g. parent's income and education, neighbourhood safety, area deprivation, single parenting), somewhat weaker by lifestyles (e.g. extracurricular activities, sleep, screen time,

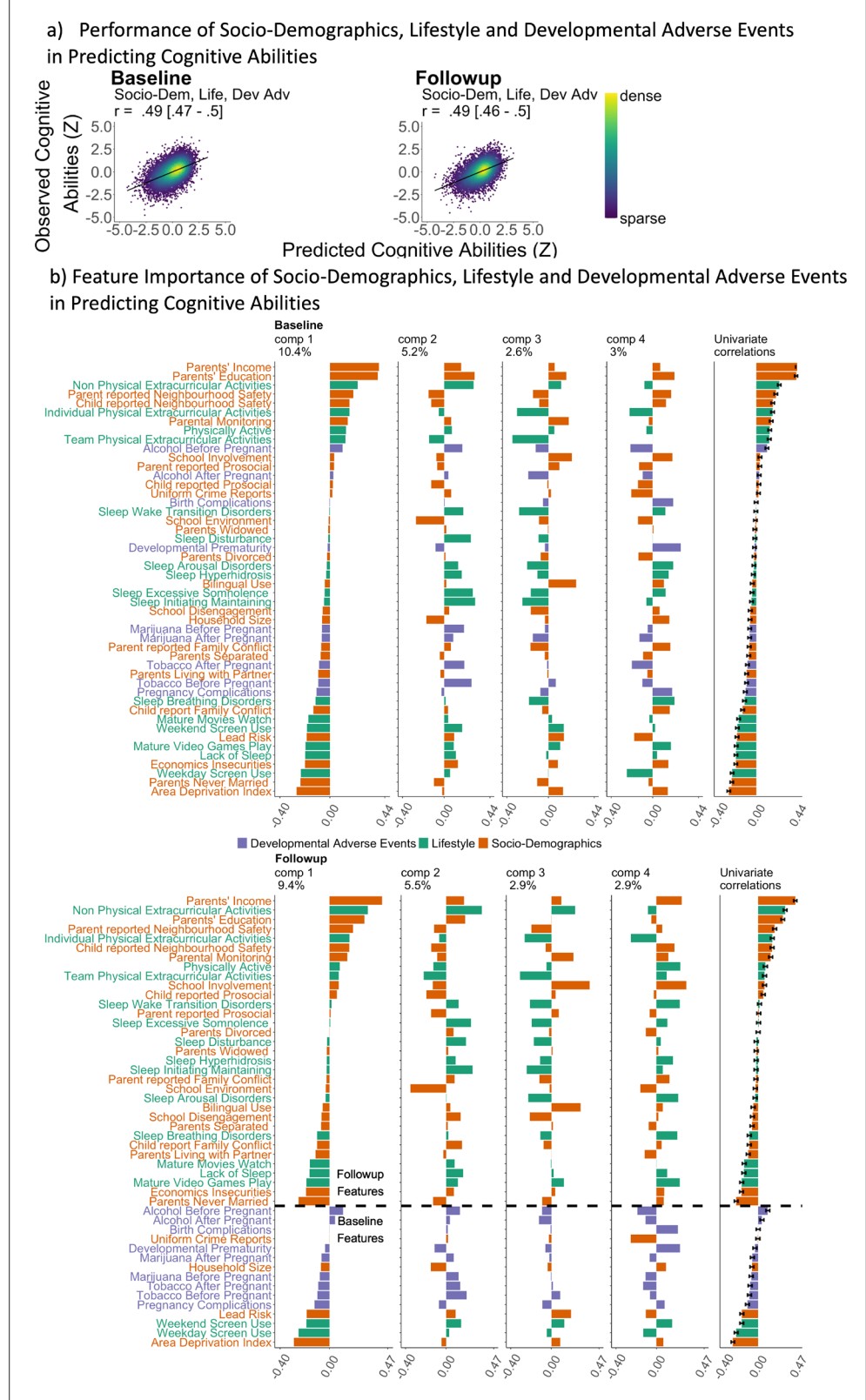

**Figure 4.** Predictive models, predicting cognitive abilities from socio-demographics, lifestyles, and developmental adverse events via Partial Least Square (PLS). (**a**) Scatter plots between observed vs predicted cognitive abilities based on socio-demographics, lifestyles, and developmental adverse events. Cognitive abilities are based on the second-order latent variable, the g-factor, based on a confirmatory factor analysis of six cognitive tasks. All data

*Figure 4 continued on next page*

*Figure 4 continued*

points are from test sets. *r* is the average Pearson's *r* across 21 test sites. The parentheses following the *r* indicate bootstrapped 95% CIs, calculated based on observed vs predicted cognitive abilities from all test sites combined. (**b**) Feature importance of socio-demographics, lifestyles, and developmental adverse events, predicting cognitive abilities via Partial Least Square. The features were ordered based on the loading of the first component. Univariate correlations were Pearson's correlation between each feature and cognitive abilities. Error bars reflect 95% CIs of the correlations. Different types of environmental factors were filled with different colours: orange for socio-demographics, purple for developmental adverse events and green for lifestyle. A dashed horizontal line in the follow-up feature importance figure distinguishes whether the variables were collected at baseline or follow-up.

video gaming, mature movie watching, and parental monitoring), and much weaker by developmental adverse events (e.g. pregnancy complications). Importantly, proxy measures based on socio-demographics, lifestyles, and developmental adverse events captured a large proportion of the relationship between cognitive abilities and mental health. Furthermore, this variance captured by socio-demographics, lifestyles, and developmental adverse events overlapped mainly with the neurobiological proxy measures. This reiterates RDoC's central tenet that understanding the neurobiology of a functional domain, such as cognitive abilities, could help us understand the extent to which environments influence mental health (*Cuthbert and Insel, 2013*; *Insel et al., 2010*). More importantly, all the results regarding neuroimaging, PGS, and socio-demographics, lifestyles, and developmental adverse events were reliable across two years during a sensitive period for adolescents.

This study has several limitations that might affect its generalisability. Firstly, the range of mental health variables was not exhaustive. While we covered various emotional and behavioural problems (*Achenbach et al., 2017*) and temperaments, including behavioural inhibition/activation (*Carver and White, 1994*) and impulsivity (*Zapolski et al., 2010*), we may still miss other critical mental health variables, such as psychotic-like experiences, eating disorder symptoms, and mania. Similarly, our ABCD samples were young and community-based, likely limiting the severity of their psychopathological issues (*Kessler et al., 2007*). Future work needs to test if the results found here are generalisable to adults and participants with stronger severity. Next, for cognitive abilities, while the six cognitive tasks (*Luciana et al., 2018*; *Thompson et al., 2019*) covered most of the RDoC cognitive abilities/systems constructs, we still missed variability in some domains, such as perception (*Morris*

**Table 4.** Results of linear-mixed models using proxy measures of cognitive abilities based on mental health and/or neuroimaging as regressors to explain cognitive abilities across test sites in the baseline.

| Response | Cognitive abilities | | | Cognitive abilities | | | Cognitive abilities | | |
|---|---|---|---|---|---|---|---|---|---|
| *Regressors* | Estimates | CI | p | Estimates | CI | p | Estimates | CI | p |
| (Intercept) | 0.02 | −0.00–0.03 | 0.058 | 0.02 | −0.00–0.04 | 0.057 | 0.02 | −0.00–0.03 | 0.067 |
| mental savg | 0.00 | −0.02–0.02 | 0.895 | 0.00 | −0.02–0.02 | 0.985 | | | |
| mental cws | 0.19 | 0.17–0.20 | <0.001 | 0.31 | 0.29–0.33 | <0.001 | | | |
| neuroimaging savg | −0.01 | −0.02–0.01 | 0.507 | | | | −0.01 | −0.02–0.01 | 0.523 |
| neuroimaging cws | 0.43 | 0.41–0.44 | <0.001 | | | | 0.48 | 0.47–0.50 | <0.001 |
| Random Effects | | | | | | | | | |
| σ² | 0.55 | | | 0.54 | | | 0.57 | | |
| τ₀₀ | 0.17 SITE_ID_L:REL_FAMILY_ID | | | 0.35 SITE_ID_L:REL_FAMILY_ID | | | 0.18 SITE_ID_L:REL_FAMILY_ID | | |
| ICC | 0.24 | | | 0.39 | | | 0.24 | | |
| N | 21 SITE_ID_L | | | 21 SITE_ID_L | | | 21 SITE_ID_L | | |
| | 9001 REL_FAMILY_ID | | | 9001 REL_FAMILY_ID | | | 9001 REL_FAMILY_ID | | |
| Observations | 10728 | | | 10728 | | | 10728 | | |
| Marginal R² | 0.272 | | | 0.098 | | | 0.238 | | |
| Conditional R² | 0.444 | | | 0.452 | | | 0.423 | | |

cws = values centred within each site; savg = values averaged within each site.

**Table 5.** Results of linear-mixed models using proxy measures of cognitive abilities based on mental health and/or neuroimaging as regressors to explain cognitive abilities across test sites in the follow-up.

| Response | Cognitive abilities | | | Cognitive abilities | | | Cognitive abilities | | |
|---|---|---|---|---|---|---|---|---|---|
| Regressors | Estimates | CI | p | Estimates | CI | p | Estimates | CI | p |
| (Intercept) | 0.82 | 0.80–0.84 | <0.001 | 0.82 | 0.80–0.85 | <0.001 | 0.82 | 0.80–0.84 | <0.001 |
| mental savg | 0.02 | 0.00–0.04 | 0.047 | 0.02 | 0.00–0.05 | 0.037 | | | |
| mental cws | 0.19 | 0.17–0.21 | <0.001 | 0.31 | 0.29–0.33 | <0.001 | | | |
| neuroimaging savg | 0.02 | 0.00–0.05 | 0.021 | | | | 0.03 | 0.01–0.05 | 0.012 |
| neuroimaging cws | 0.42 | 0.40–0.44 | <0.001 | | | | 0.47 | 0.45–0.49 | <0.001 |
| Random Effects | | | | | | | | | |
| $\sigma^2$ | 0.41 | | | 0.45 | | | 0.42 | | |
| $\tau_{00}$ | 0.24 $_{SITE\_ID\_L:REL\_FAMILY\_ID}$ | | | 0.37 $_{SITE\_ID\_L:REL\_FAMILY\_ID}$ | | | 0.27 $_{SITE\_ID\_L:REL\_FAMILY\_ID}$ | | |
| ICC | 0.37 | | | 0.46 | | | 0.40 | | |
| N | 21 $_{SITE\_ID\_L}$ | | | 21 $_{SITE\_ID\_L}$ | | | 21 $_{SITE\_ID\_L}$ | | |
| | 5434 $_{REL\_FAMILY\_ID}$ | | | 5434 $_{REL\_FAMILY\_ID}$ | | | 5434 $_{REL\_FAMILY\_ID}$ | | |
| Observations | 6315 | | | 6315 | | | 6315 | | |
| Marginal $R^2$ | 0.286 | | | 0.104 | | | 0.245 | | |
| Conditional $R^2$ | 0.552 | | | 0.513 | | | 0.545 | | |

cws = values centred within each site; savg = values averaged within each site.

**Table 6.** Results of linear-mixed models using proxy measures of cognitive abilities based on mental health and/or polygenic scores as regressors to explain cognitive abilities across test sites in the baseline.

| Response | Cognitive abilities | | | Cognitive abilities | | | Cognitive abilities | | |
|---|---|---|---|---|---|---|---|---|---|
| Regressors | Estimates | CI | p | Estimates | CI | p | Estimates | CI | p |
| (Intercept) | 0.23 | 0.21–0.26 | <0.001 | 0.23 | 0.21–0.25 | <0.001 | 0.23 | 0.21–0.26 | <0.001 |
| mental savg | 0.06 | 0.02–0.09 | 0.004 | 0.13 | 0.10–0.15 | <0.001 | | | |
| mental cws | 0.25 | 0.23–0.27 | <0.001 | 0.25 | 0.23–0.27 | <0.001 | | | |
| PGS savg favg | −0.08 | −0.12 to −0.05 | <0.001 | | | | −0.13 | −0.15 to −0.10 | <0.001 |
| PGS cws cwf | 0.05 | 0.03–0.07 | <0.001 | | | | 0.06 | 0.04–0.08 | <0.001 |
| Random Effects | | | | | | | | | |
| $\sigma^2$ | 0.51 | | | 0.52 | | | 0.53 | | |
| $\tau_{00}$ | 0.27 $_{SITE\_ID\_L:REL\_FAMILY\_ID}$ | | | 0.26 $_{SITE\_ID\_L:REL\_FAMILY\_ID}$ | | | 0.32 $_{SITE\_ID\_L:REL\_FAMILY\_ID}$ | | |
| ICC | 0.34 | | | 0.33 | | | 0.38 | | |
| N | 21 $_{SITE\_ID\_L}$ | | | 21 $_{SITE\_ID\_L}$ | | | 21 $_{SITE\_ID\_L}$ | | |
| | 4734 $_{REL\_FAMILY\_ID}$ | | | 4734 $_{REL\_FAMILY\_ID}$ | | | 4734 $_{REL\_FAMILY\_ID}$ | | |
| Observations | 5766 | | | 5766 | | | 5766 | | |
| Marginal $R^2$ | 0.098 | | | 0.092 | | | 0.026 | | |
| Conditional $R^2$ | 0.408 | | | 0.394 | | | 0.394 | | |

cws = values centred within each site; savg = values averaged within each site; cws,cwf = values centred within each family first and then within each site; savg,favg = values averaged within each family first and then within each site. PGS = polygenic scores.

*and Cuthbert, 2012*). Additionally, several children (3274) did not complete all six cognitive tasks at follow-up, which might create a discrepancy between baseline and follow-up samples. However, the differences in social demographics, lifestyles, and developmental adverse events between participants who provided cognitive scores in the follow-up were minimal (Cohen's d ranging from 0.007 to 0.092, see *Table 12*). Moreover, given that we found a similar pattern of predictive performance

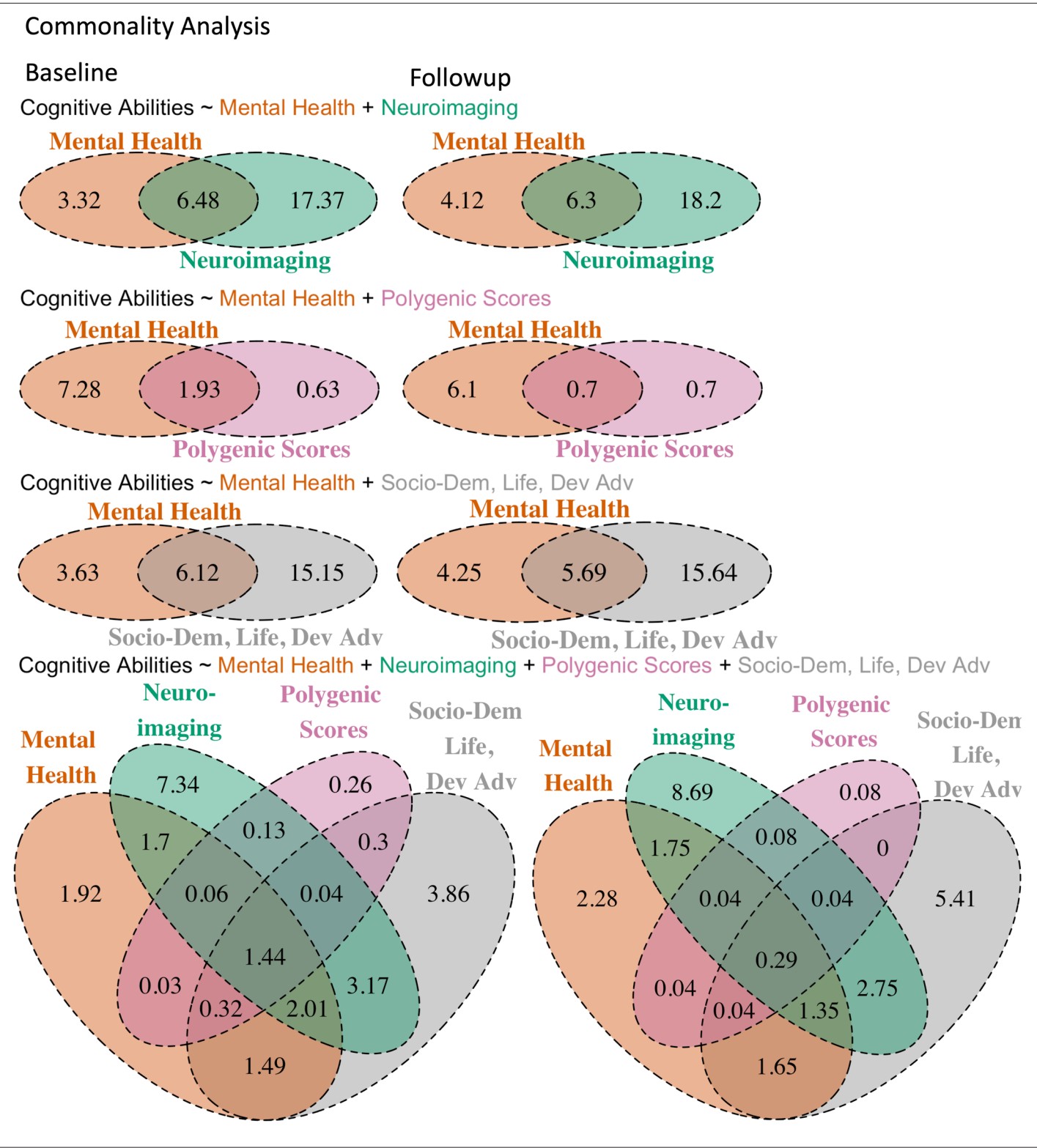

**Figure 5.** Venn diagrams showing common and unique effects of proxy measures of cognitive abilities based on mental health, neuroimaging, polygenic scores, and/or socio-demographics, lifestyles and developmental adverse events in explaining cognitive abilities across test sites. We computed the common and unique effects in % based on the marginal $R^2$ of four sets of linear-mixed models.

The online version of this article includes the following figure supplement(s) for figure 5:

**Figure supplement 1.** Stacked bar plots showing common and unique effects of proxy measures of cognitive abilities based on each set of neuroimaging features in explaining cognitive abilities across test sites.

**Table 7.** Results of linear-mixed models using proxy measures of cognitive abilities based on mental health and/or polygenic scores as regressors to explain cognitive abilities across test sites in the follow-up.

| Response | Cognitive abilities | | | Cognitive abilities | | | Cognitive abilities | | |
|---|---|---|---|---|---|---|---|---|---|
| Predictors | Estimates | CI | p | Estimates | CI | p | Estimates | CI | p |
| (Intercept) | 1.06 | 1.03–1.09 | <0.001 | 1.06 | 1.03–1.09 | <0.001 | 1.06 | 1.03–1.09 | <0.001 |
| mental savg | 0.03 | –0.00–0.07 | 0.063 | 0.07 | 0.05–0.10 | <0.001 | | | |
| mental cws | 0.22 | 0.19–0.25 | <0.001 | 0.22 | 0.20–0.25 | <0.001 | | | |
| PGS savg favg | –0.07 | –0.10 to –0.04 | <0.001 | | | | –0.09 | –0.12 to –0.06 | <0.001 |
| PGS cws cwf | 0.04 | 0.02–0.06 | <0.001 | | | | 0.05 | 0.03–0.07 | <0.001 |
| Random Effects | | | | | | | | | |
| $\sigma^2$ | 0.42 | | | 0.43 | | | 0.43 | | |
| $\tau_{00}$ | 0.32 SITE_ID_L:REL_FAMILY_ID | | | 0.31 SITE_ID_L:REL_FAMILY_ID | | | 0.37 SITE_ID_L:REL_FAMILY_ID | | |
| ICC | 0.43 | | | 0.42 | | | 0.46 | | |
| N | 21 SITE_ID_L | | | 21 SITE_ID_L | | | 21 SITE_ID_L | | |
| | 3370 REL_FAMILY_ID | | | 3370 REL_FAMILY_ID | | | 3370 REL_FAMILY_ID | | |
| Observations | 4036 | | | 4036 | | | 4036 | | |
| Marginal $R^2$ | 0.075 | | | 0.068 | | | 0.013 | | |
| Conditional $R^2$ | 0.470 | | | 0.460 | | | 0.469 | | |

cws = values centred within each site; savg = values averaged within each site; cws,cwf = values centred within each family first and then within each site; savg,favg = values averaged within each family first and then within each site. PGS = polygenic scores.

**Table 8.** Results of linear-mixed models using proxy measures of cognitive abilities based on mental health and/or socio-demographics, lifestyles, and developmental adverse events as regressors to explain cognitive abilities across test sites in the baseline.

| Response | Cognitive abilities | | | Cognitive abilities | | | Cognitive abilities | | |
|---|---|---|---|---|---|---|---|---|---|
| *Regressors* | *Estimates* | *CI* | *p* | *Estimates* | *CI* | *p* | *Estimates* | *CI* | *p* |
| (Intercept) | 0.01 | –0.01–0.02 | 0.525 | 0.01 | –0.01–0.03 | 0.385 | 0.01 | –0.01–0.02 | 0.558 |
| mental savg | –0.00 | –0.02–0.02 | 0.917 | –0.00 | –0.02–0.02 | 0.930 | | | |
| mental cws | 0.20 | 0.18–0.22 | <0.001 | 0.31 | 0.29–0.33 | <0.001 | | | |
| sdl savg | 0.00 | –0.02–0.02 | 0.819 | | | | 0.00 | –0.01–0.02 | 0.792 |
| sdl cws | 0.40 | 0.38–0.41 | <0.001 | | | | 0.46 | 0.44–0.48 | <0.001 |
| Random Effects | | | | | | | | | |
| $\sigma^2$ | 0.52 | | | 0.53 | | | 0.54 | | |
| $\tau_{00}$ | 0.22 SITE_ID_L:REL_FAMILY_ID | | | 0.35 SITE_ID_L:REL_FAMILY_ID | | | 0.24 SITE_ID_L:REL_FAMILY_ID | | |
| ICC | 0.30 | | | 0.40 | | | 0.31 | | |
| N | 21 SITE_ID_L | | | 21 SITE_ID_L | | | 21 SITE_ID_L | | |
| | 9390 REL_FAMILY_ID | | | 9390 REL_FAMILY_ID | | | 9390 REL_FAMILY_ID | | |
| Observations | 11294 | | | 11294 | | | 11294 | | |
| Marginal $R^2$ | 0.249 | | | 0.098 | | | 0.213 | | |
| Conditional $R^2$ | 0.474 | | | 0.458 | | | 0.456 | | |

cws = values centred within each site; savg = values averaged within each site; sdl = socio-demographics, lifestyles and developmental adverse events.

across the two time points, we believe excluding the children who did not complete the cognitive tasks at follow-up should not alter our conclusions.

Furthermore, while we used comprehensive multimodal MRI from 45 sets of features for neuro-imaging, three fMRI tasks were not chosen based on their relevance to cognitive abilities (*Casey*

**Table 9.** Results of linear-mixed models using proxy measures of cognitive abilities based on mental health and/or socio-demographics, lifestyles and developmental adverse events as regressors to explain cognitive abilities across test sites in the follow-up.

| Response | Cognitive abilities | | | Cognitive abilities | | | Cognitive abilities | | |
|---|---|---|---|---|---|---|---|---|---|
| *Regressors* | *Estimates* | *CI* | *p* | *Estimates* | *CI* | *p* | *Estimates* | *CI* | *p* |
| (Intercept) | 0.83 | 0.81–0.85 | <0.001 | 0.83 | 0.81–0.86 | <0.001 | 0.83 | 0.81–0.85 | <0.001 |
| mental savg | 0.01 | –0.01–0.03 | 0.185 | 0.01 | –0.01–0.04 | 0.198 | | | |
| mental cws | 0.20 | 0.18–0.22 | <0.001 | 0.30 | 0.28–0.32 | <0.001 | | | |
| sdl savg | 0.00 | –0.02–0.02 | 0.957 | | | | 0.00 | –0.02–0.02 | 0.757 |
| sdl cws | 0.39 | 0.37–0.41 | <0.001 | | | | 0.44 | 0.42–0.47 | <0.001 |
| **Random Effects** | | | | | | | | | |
| $\sigma^2$ | 0.42 | | | 0.45 | | | 0.43 | | |
| $\tau_{00}$ | 0.27 $_{SITE\_ID\_L:REL\_FAMILY\_ID}$ | | | 0.37 $_{SITE\_ID\_L:REL\_FAMILY\_ID}$ | | | 0.30 $_{SITE\_ID\_L:REL\_FAMILY\_ID}$ | | |
| ICC | 0.39 | | | 0.45 | | | 0.41 | | |
| N | 21 $_{SITE\_ID\_L}$ | | | 21 $_{SITE\_ID\_L}$ | | | 21 $_{SITE\_ID\_L}$ | | |
| | 6217 $_{REL\_FAMILY\_ID}$ | | | 6217 $_{REL\_FAMILY\_ID}$ | | | 6217 $_{REL\_FAMILY\_ID}$ | | |
| Observations | 7382 | | | 7382 | | | 7382 | | |
| Marginal $R^2$ | 0.256 | | | 0.099 | | | 0.213 | | |
| Conditional $R^2$ | 0.543 | | | 0.508 | | | 0.535 | | |

cws = values centred within each site; savg = values averaged within each site; sdl = socio-demographics, lifestyles and developmental adverse events.

*et al., 2018*). It is possible to obtain higher predictive performance based on other fMRI tasks. For all analyses involving PGS, we limited our participants to children of European ancestry due to the lack of summary statistics from well-powered GWAS for cognitive abilities in non-European participants. This prevented us from fully leveraging the diverse samples in the ABCD study (*Garavan et al., 2018*). Future GWAS work with more diverse samples is needed to ensure equity and fairness in developing neurobiological units of analysis for cognitive abilities. Lastly, we relied on 44 variables of socio-demographics, lifestyles, and developmental adverse events included in the study, which might have missed some variables relevant to cognitive abilities (e.g. nutrition). The ABCD study (*Casey et al., 2018*) is ongoing, and future data might address some of these limitations.

Overall, aligning with the RDoC perspective (*Morris and Cuthbert, 2012*), our findings support the use of neurobiological units of analysis for cognitive abilities, as assessed through multimodal neuroimaging and Polygenic Scores (PGS). These measures explain (a) the relationship between cognitive abilities and mental health and (b) the variance in this cognitive-ability-and-mental-health relationship attributable to environmental factors. Our results emphasise the importance of considering both neurobiology and environmental factors, such as socio-demographics, lifestyles, and adverse childhood events, to gain a comprehensive understanding of the aetiology of mental health (*Insel et al., 2010*; *Morris et al., 2022*).

## Materials and methods
### The ABCD study
We used data from the AABCD Study Curated Annual Release 5.1 (DOI:10.15154/z563-zd24) from two time points. The baseline included data from 11,868 children (5677 females and 3 others, aged 9–10 years), while the two-year follow-up included data from the same children two years later (10,908 children, 5181 females and 3 others). Although the ABCD collected data from 22 sites across the United States, we excluded data from Site 22 since this site only provided data from 35 children at baseline and none at follow-up (*Garavan et al., 2018*). We also excluded 69 children based on the Snellen Vision Screener (*Luciana et al., 2018*; *Snellen, 1862*). These children either could not read any line on the chart, could only read the largest line, or could read up to the fourth line clearly but had difficulty reading stimuli

**Table 10.** Results of linear-mixed models using proxy measures of cognitive abilities based on mental health, neuroimaging, polygenic scores and/or socio-demographics, lifestyles and developmental adverse events as regressors to explain cognitive abilities across test sites in the baseline.

| Response | Cognitive abilities | | | Cognitive abilities | | |
|---|---|---|---|---|---|---|
| *Regressors* | Estimates | CI | p | Estimates | CI | p |
| (Intercept) | 0.24 | 0.21–0.26 | **<0.001** | 0.24 | 0.21–0.26 | **<0.001** |
| mental savg | 0.00 | −0.05–0.05 | 0.975 | 0.09 | 0.05–0.12 | **<0.001** |
| mental cws | 0.14 | **0.11–0.16** | **<0.001** | 0.18 | 0.15–0.20 | **<0.001** |
| neuroimaging savg | 0.01 | −0.03–0.05 | 0.533 | 0.05 | 0.01–0.09 | **0.006** |
| neuroimaging cws | 0.26 | **0.24–0.29** | **<0.001** | 0.31 | 0.28–0.33 | **<0.001** |
| PGS savg favg | −0.04 | −0.08–0.00 | 0.070 | | | |
| PGS cws cwf | 0.05 | 0.03–0.07 | **<0.001** | | | |
| sdl savg | 0.09 | 0.03–0.16 | **0.006** | | | |
| sdl cws | 0.18 | 0.16–0.21 | **<0.001** | | | |
| $\sigma^2$ | 0.50 | | | 0.52 | | |
| $\tau_{00}$ | 0.15 SITE_ID_L:REL_FAMILY_ID | | | 0.17 SITE_ID_L:REL_FAMILY_ID | | |
| ICC | 0.23 | | | 0.25 | | |
| N | 21 SITE_ID_L | | | 21 SITE_ID_L | | |
| Observations | 5520 | | | 5520 | | |
| Marginal $R^2$ | 0.241 | | | 0.197 | | |
| Conditional $R^2$ | 0.416 | | | 0.395 | | |
| *Regressors* | Estimates | CI | p | Estimates | CI | p |
| (Intercept) | 0.24 | 0.21–0.26 | <0.001 | 0.24 | 0.21–0.26 | <0.001 |
| mental savg | 0.06 | 0.03–0.10 | 0.001 | 0.00 | −0.04–0.05 | 0.890 |
| mental cws | 0.24 | 0.22–0.27 | <0.001 | 0.19 | 0.16–0.21 | <0.001 |
| neuroimaging savg | | | | | | |
| neuroimaging cws | | | | | | |
| PGS savg favg | −0.08 | −0.12 to −0.05 | **<0.001** | | | |
| PGS cws cwf | 0.06 | 0.04–0.08 | **<0.001** | | | |
| sdl savg | | | | 0.14 | 0.09–0.19 | <0.001 |
| sdl cws | | | | 0.25 | 0.22–0.27 | <0.001 |
| $\sigma^2$ | 0.51 | | | 0.52 | | |
| $\tau_{00}$ | 0.27 SITE_ID_L:REL_FAMILY_ID | | | 0.20 SITE_ID_L:REL_FAMILY_ID | | |
| ICC | 0.34 | | | 0.28 | | |
| N | 21 SITE_ID_L | | | 21 SITE_ID_L | | |
| | 4571 REL_FAMILY_ID | | | 4571 REL_FAMILY_ID | | |
| Observations | 5520 | | | 5520 | | |
| Marginal $R^2$ | 0.097 | | | 0.163 | | |
| Conditional $R^2$ | 0.408 | | | 0.395 | | |

cws = values centred within each site; savg = values averaged within each site; cws,cwf = values centred within each family first and then within each site; savg,favg = values averaged within each family first and then within each site; PGS = polygenic scores; sdl = socio-demographics, lifestyles and developmental adverse events.

on an iPad used for administering cognitive tasks (explained below). We listed the number of participants following each inclusion and exclusion criteria for each variable in *Figure 6*, *Figure 6—figure supplement 1* and *Tables 13–14*. Institutional Review Boards at each site approved the study protocols. Please see *Clark et al., 2018* for ethical details, such as informed consent and confidentiality.

**Table 11.** Results of linear-mixed models using proxy measures of cognitive abilities based on mental health, neuroimaging, polygenic scores and/or socio-demographics, lifestyles, and developmental adverse events as regressors to explain cognitive abilities across test sites in the follow-up.

| Response | Cognitive abilities | | | Cognitive abilities | | |
|---|---|---|---|---|---|---|
| *Regressors* | *Estimates* | *CI* | *p* | *Estimates* | *CI* | *p* |
| (Intercept) | 1.05 | 1.02–1.08 | **<0.001** | 1.05 | 1.02–1.08 | **<0.001** |
| mental savg | 0.05 | –0.01–0.10 | 0.100 | 0.06 | 0.03–0.10 | **<0.001** |
| mental cws | 0.13 | 0.11–0.16 | **<0.001** | 0.17 | 0.14–0.20 | **<0.001** |
| neuroimaging savg | 0.00 | –0.06–0.06 | 0.935 | 0.03 | –0.01–0.06 | 0.146 |
| neuroimaging cws | 0.27 | 0.24–0.30 | **<0.001** | 0.31 | 0.28–0.33 | **<0.001** |
| PGS savg favg | 0.00 | –0.03–0.04 | 0.833 | | | |
| PGS cws cwf | 0.04 | 0.02–0.06 | **<0.001** | | | |
| sdl savg | 0.04 | –0.04–0.12 | 0.349 | | | |
| sdl cws | 0.20 | 0.17–0.23 | **<0.001** | | | |
| $\sigma^2$ | 0.38 | | | 0.40 | | |
| $\tau_{00}$ | 0.23 SITE_ID_L:REL_FAMILY_ID | | | 0.25 SITE_ID_L:REL_FAMILY_ID | | |
| ICC | 0.38 | | | 0.39 | | |
| N | 21 SITE_ID_L | | | 21 SITE_ID_L | | |
| | 2930 REL_FAMILY_ID | | | 2930 REL_FAMILY_ID | | |
| Observations | 3423 | | | 3423 | | |
| Marginal $R^2$ | 0.242 | | | 0.190 | | |
| Conditional $R^2$ | 0.527 | | | 0.506 | | |
| *Regressors* | *Estimates* | *CI* | *p* | *Estimates* | *CI* | *p* |
| (Intercept) | 1.05 | 1.02–1.08 | **<0.001** | 1.05 | 1.02–1.08 | **<0.001** |
| mental savg | 0.08 | 0.04–0.11 | **<0.001** | 0.05 | –0.00–0.10 | 0.074 |
| mental cws | 0.23 | 0.20–0.26 | **<0.001** | 0.18 | 0.15–0.21 | **<0.001** |
| neuroimaging savg | | | | | | |
| neuroimaging cws | | | | | | |
| PGS savg favg | 0.00 | - 0.03–0.04 | 0.844 | | | |
| PGS cws cwf | 0.05 | 0.03–0.07 | **<0.001** | | | |
| PGS savg favg | 0.00 | - 0.03–0.04 | 0.844 | | | |
| PGS cws cwf | 0.05 | 0.03–0.07 | <0.001 | | | |
| sdl savg | | | | 0.04 | –0.01–0.09 | 0.092 |
| sdl cws | | | | 0.25 | 0.22–0.28 | **<0.001** |
| $\sigma^2$ | 0.41 | | | 0.42 | | |
| $\tau_{00}$ | 0.33 SITE_ID_L:REL_FAMILY_ID | | | 0.27 SITE_ID_L:REL_FAMILY_ID | | |
| ICC | 0.45 | | | 0.39 | | |
| N | 21 SITE_ID_L | | | 21 SITE_ID_L | | |
| | 2930 REL_FAMILY_ID | | | 2930 REL_FAMILY_ID | | |
| Observations | 3423 | | | 3423 | | |
| Marginal $R^2$ | 0.076 | | | 0.153 | | |
| Conditional $R^2$ | 0.491 | | | 0.486 | | |

cws = values centred within each site; savg = values averaged within each site; cws,cwf = values centred within each family first and then within each site; savg,favg = values averaged within each family first and then within each site; PGS = polygenic scores; sdl = socio-demographics, lifestyles and developmental adverse events.

**Table 12.** The differences in social demographics, lifestyles, and developmental adverse events between participants who provided cognitive scores in the follow-up.

We used social demographics, lifestyles, and developmental adverse events collected at baseline.

| Variable names | Having cognitive scores in the follow-up. | Not having cognitive scores in the follow-up. | Test statistics |
|---|---|---|---|
| Age in months | Mean (sd): 119.3 (7.5) | Mean (sd): 118.3 (7.6) | Yuen's $t(3783)$=6.05, $p < 0.001$, Cohen's $d = 0.092$ |
| Sex | Male = 3918 (52.4%) Female = 3564 (47.6%) Intersex-Male=1 (0.0%) Intersex-female=0 (0.0%) Do not know = 0 (0.0%) | Male = 1776 (53.2%) Female = 1563 (46.8%) Intersex-Male=2 (0.1%) Intersex-female=0(0.0%) Do not know = 0 (0.0%) | $(X^2 = 4, N = 10824)$=6, $p$=0.199 |
| Body Mass Index | Mean (sd): 18.7 (4.1) | Mean (sd): 18.9 (4.4) | Yuen's $t$ (3658)=1.605, $p$=0.109, Cohen's $d$=0.023 |
| Race | White = 4190 (56.0%) Black = 918 (12.3%) Hispanic = 1441 (19.3%) Asian = 157(2.1%) Other = 777 (10.4%) | White = 1611 (48.2%) Black = 612 (18.3%) Hispanic = 689 (20.6%) Asian = 68(2.0%) Other = 360 (10.8%) | $X^2$(16, N=10823)=20, $p$ = 0.22 |
| Bilingual Use | Mean (sd): 1 (1.7) | Mean (sd): 1 (1.7) | Yuen's $t$(3776)=0.696, $p$=0.486, Cohen's $d$=0.011 |
| Parent Marital Status | Married = 5239 (70.5%) Widowed = 59(0.8%) Divorced = 684 (9.2%) Separated = 264 (3.6%) NeverMarried = 806(10.8%) LivingWithPartner = 381 (5.1%) | Married = 2194 (66.0%) Widowed = 29(0.9%) Divorced = 290 (8.7%) Separated = 135 (4.1%) NeverMarried = 460(13.8%) LivingWithPartner = 214 (6.4%) | $X^2$(25, N=10755)=30, $p$=0.224 |
| Parents' Education | Mean (sd): 16.6 (2.6) | Mean (sd): 16.3 (2.8) | Yuen's $t$(3262)=4.175, $p$<0.001, Cohen's $d$=0.068 |
| Parents' Income | Mean (sd): 7.4 (2.3) | Mean (sd): 7.2 (2.5) | Yuen's $t$(2854)=2.243, $p$=0.025, Cohen's $d$=0.034 |
| Household Size | Mean (sd): 4.7 (1.5) | Mean (sd): 4.7 (1.6) | Yuen's $t$(3718)=0.39, $p$=0.697, Cohen's $d$=0.007 |
| Economics Insecurities | Mean (sd): 0.4 (1.1) | Mean (sd): 0.5 (1.1) | Yuen's $t$(1982)=2.65, $p$=0.008, Cohen's $d$=0.033 |
| Area Deprivation Index | Mean (sd): 94.6 (20.7) | Mean (sd): 94.9 (21.2) | Yuen's $t$(3297)=1.686, $p$=0.092, Cohen's $d$=0.029 |
| Lead Risk | Mean (sd): 5 (3.1) | Mean (sd): 5.1 (3.1) | Yuen's $t$(3374)=1.797, $p$=0.072, Cohen's $d$=0.027 |
| Uniform Crime Reports | Mean (sd): 12.1 (5.5) | Mean (sd): 12 (6.1) | Yuen's $t$(3370)=0.873, $p$=0.383, Cohen's $d$=0.014 |
| Parent reported Neighbourhood Safety | Mean (sd): 11.8 (2.9) | Mean (sd): 11.6 (3) | Yuen's $t$(3382)=1.799, $p$=0.072, Cohen's $d$=0.025 |
| Child reported Neighbourhood Safety | Mean (sd): 4.1 (1.1) | Mean (sd): 4 (1.1) | Yuen's $t$(3786)=2.258, $p$=0.024, Cohen's $d$=0.036 |
| School Environment | Mean (sd): 20 (2.8) | Mean (sd): 19.8 (2.9) | Yuen's $t$(3787)=1.763, $p$=0.078, Cohen's $d$=0.029 |
| School Involvement | Mean (sd): 13.1 (2.3) | Mean (sd): 12.9 (2.4) | Yuen's $t$(3790)=3.203, $p$=0.001, Cohen's $d$=0.05 |
| School Disengagement | Mean (sd): 3.7 (1.4) | Mean (sd): 3.8 (1.5) | Yuen's $t$(3800)=2.171, $p$=0.03, Cohen's $d$=0.035 |
| Lack of Sleep | Mean (sd): 1.7 (0.8) | Mean (sd): 1.7 (0.8) | Yuen's $t$(3860)=3.084, $p$=0.002, Cohen's $d$=0.05 |
| Sleep Disturbance | Mean (sd): 1.9 (*Abramovitch et al., 2021*) | Mean (sd): 1.9 (*Abramovitch et al., 2021*) | Yuen's $t$(3877)=1.567, $p$=0.117, Cohen's $d$=0.025 |
| Sleep Initiating Maintaining | Mean (sd): 11.7 (3.7) | Mean (sd): 11.9 (3.8) | Yuen's $t$(3862)=2.481, $p$=0.013, Cohen's $d$=0.038 |
| Sleep Breathing Disorders | Mean (sd): 3.7 (1.2) | Mean (sd): 3.8 (1.3) | Yuen's $t$(3834)=1.43, $p$=0.153, Cohen's $d$=0.022 |
| Sleep Arousal Disorders | Mean (sd): 3.4 (0.9) | Mean (sd): 3.4 (*Abramovitch et al., 2021*) | Yuen's $t$(3885)=0.966, $p$=0.334, Cohen's $d$=0.013 |
| Sleep Wake Transition Disorders | Mean (sd): 8.2 (2.6) | Mean (sd): 8.1 (2.6) | Yuen's $t$(3828)=1.198, $p$=0.231, Cohen's $d$=0.022 |

*Table 12 continued on next page*

*Table 12 continued*

| Variable names | Having cognitive scores in the follow-up. | Not having cognitive scores in the follow-up. | Test statistics |
|---|---|---|---|
| Sleep Excessive Somnolence | Mean (sd): 6.9 (2.4) | Mean (sd): 7 (2.5) | Yuen's $t(3836)$=0.131, $p$=0.896, Cohen's $d$=0.007 |
| Sleep Hyperhidrosis | Mean (sd): 2.4 (1.2) | Mean (sd): 2.5 (1.2) | Yuen's $t(4375)$=1.755, $p$=0.079, Cohen's $d$=0.029 |
| Individual Physical Extracurricular Activities | Mean (sd): 5 (5.7) | Mean (sd): 4.7 (5.4) | Yuen's $t(4173)$=2.933, $p$=0.003, Cohen's $d$=0.044 |
| Team Physical Extracurricular Activities | Mean (sd): 8.4 (7.7) | Mean (sd): 7.8 (7.4) | Yuen's $t(4007)$=3.604, $p$<0.001, Cohen's $d$=0.055 |
| Non Physical Extracurricular Activities | Mean (sd): 5.1 (6.3) | Mean (sd): 4.8 (6.1) | Yuen's $t(4075)$=2.961, $p$=0.003, Cohen's $d$=0.047 |
| Physically Active | Mean (sd): 3.5 (2.3) | Mean (sd): 3.4 (2.3) | Yuen's $t(3838)$=2.094, $p$=0.036, Cohen's $d$=0.033 |
| Mature Video Games Play | Mean (sd): 0.5 (0.8) | Mean (sd): 0.6 (0.9) | Yuen's $t(3816)$=1.396, $p$=0.163, Cohen's $d$=0.022 |
| Mature Movies Watch | Mean (sd): 0.4 (0.6) | Mean (sd): 0.4 (0.7) | Yuen's $t(3728)$=4.038, $p$<0.001, Cohen's $d$=0.065 |
| Weekday Screen Use | Mean (sd): 3.3 (3) | Mean (sd): 3.6 (3.3) | Yuen's $t(3220)$=4.161,$p$<0.001, Cohen's $d$=0.069 |
| Weekend Screen Use | Mean (sd): 4.5 (3.5) | Mean (sd): 4.8 (3.7) | Yuen's $t(3521)$=3.218, $p$=0.001, Cohen's $d$=0.053 |
| Tobacco Before Pregnant | No = 6328 (86.7%) Yes = 974 (13.3%) | No = 2838 (86.7%) Yes = 436 (13.3%) | $X^2(1,$=10576)=0, $p$=1 |
| Tobacco After Pregnant | No = 6968 (95.2%) Yes = 351 (4.8%) | No = 3081 (94.2%) Yes = 190 (5.8%) | $X^2(1,$=10590)=0, $p$=1 |
| Alcohol Before Pregnant | No = 5174 (73.4%) Yes = 1871 (26.6%) | No = 2380 (75.4%) Yes = 775 (24.6%) | $X^2(1,$=10200)=0, $p$=1 |
| Alcohol After Pregnant | No = 7096 (97.1%) Yes = 210 (2.9%) | No = 3175 (97.4%) Yes = 85 (2.6%) | $X^2(1,$=10566)=0, $p$=1 |
| Marijuana Before Pregnant | No = 6874 (94.5%) Yes = 399 (5.5%) | No = 3044 (93.9%) Yes = 199 (6.1%) | $X^2(1,$=10516)=0, $p$=1 |
| Marijuana After Pregnant | No = 7182 (98.2%) Yes = 130 (1.8%) | No = 3191 (97.7%) Yes = 74 (2.3%) | $X^2(1,$=10577)=0, $p$=1 |
| Developmental Prematurity | No = 5945 (80.3%) Yes = 1458 (19.7%) | No = 2735 (83.0%) Yes = 561 (17.0%) | $X^2(1, N$=10699)=0, $p$=1 |
| Birth Complications | Mean (sd): 0.4 (0.8) | Mean (sd): 0.4 (0.7) | Yuen's $t(3591)$=0.121, $p$=0.904, Cohen's $d$=0.007 |
| Pregnancy Complications | Mean (sd): 0.6 (*Abramovitch et al., 2021*) | Mean (sd): 0.6 (*Abramovitch et al., 2021*) | Yuen's $t(3543)$=1.19, $p$=0.234, Cohen's $d$=0.018 |
| Parental Monitoring | Mean (sd): 4.4 (0.5) | Mean (sd): 4.4 (0.5) | Yuen's $t(3810)$=0.451, $p$=0.652, Cohen's $d$=0.009 |
| Parent-reported Family Conflict | Mean (sd): 2.5 (1.9) | Mean (sd): 2.6 (2) | Yuen's $t(3805)$=1.404, $p$=0.16, Cohen's $d$=0.023 |
| Child report Family Conflict | Mean (sd): 2 (1.9) | Mean (sd): 2.1 (2) | Yuen's $t(3809)$=1.751, $p$=0.08, Cohen's $d$=0.026 |
| Parent reported Prosocial | Mean (sd): 1.8 (0.4) | Mean (sd): 1.8 (0.4) | Yuen's $t(3817)$=0.288, $p$=0.774, Cohen's $d$=0.007 |
| Child reported Prosocial | Mean (sd): 1.7 (0.4) | Mean (sd): 1.7 (0.4) | Yuen's $t(3849)$=2.529, $p$=0.011, Cohen's $d$=0.041 |

## Measures: cognitive abilities

Cognitive abilities were assessed using six cognitive tasks collected with an iPad during a 70 min session outside of MRI at baseline and two-year follow-up (*Luciana et al., 2018*; *Thompson et al., 2019*). The first task was Picture Vocabulary, which measured language comprehension (*Gershon et al., 2014*). The second task was Oral Reading Recognition, which measured language decoding (*Bleck et al., 2013*). The third task was Flanker, which measured conflict monitoring and inhibitory control (*Eriksen and Eriksen, 1974*). The fourth task was Pattern Comparison Processing, which measured the speed of processing patterns (*Carlozzi et al., 2013*). The fifth task was Picture Sequence Memory, which

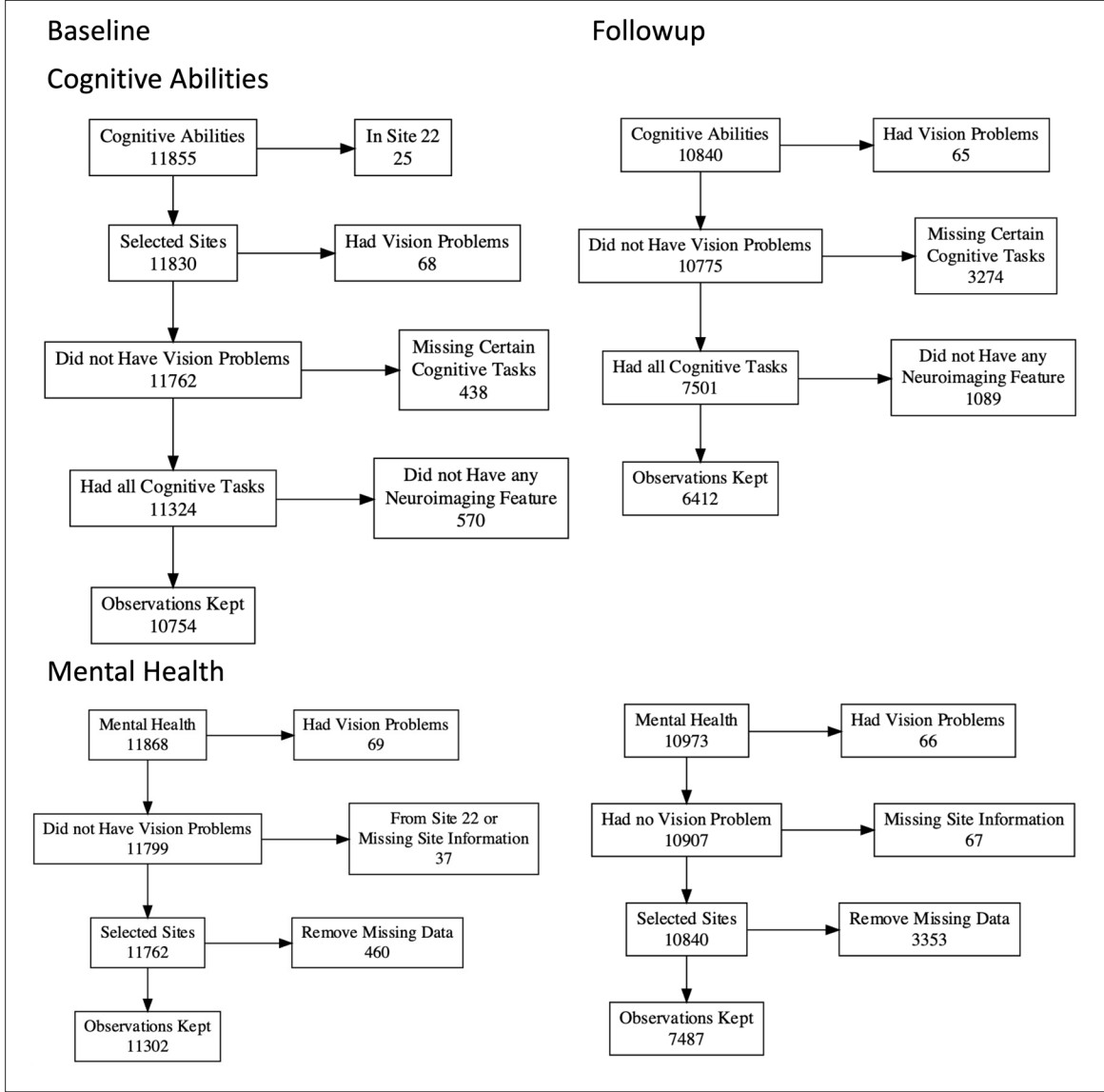

**Figure 6.** Flow diagram of participants' inclusion and exclusion criteria. Here, we show the criteria for cognitive abilities and mental health across the two time points.

The online version of this article includes the following figure supplement(s) for figure 6:

**Figure supplement 1.** Flow diagram of participants' inclusion and exclusion criteria.

measured episodic memory (*Bauer et al., 2013*). The sixth task was Rey-Auditory Verbal Learning, which measured memory recall after distraction and a short delay (*Daniel and Wahlstrom, 2014*). Rey-Auditory Verbal Learning was sourced from Pearson Assessment, while the other five cognitive tasks were from the NIH Toolbox (*Bleck et al., 2013*; *Luciana et al., 2018*). The ABCD study administered the Dimensional Change Card Sort and List Sorting Working Memory tasks from the NIH Toolbox (*Bleck et al., 2013*) only at baseline, not at the two-year follow-up (see DOI: 10.15154/z563-zd24). Consequently, these two tasks were not analysed in the current study. Additionally, 3,274 children at follow-up did not complete some of these tasks and were therefore excluded from the follow-up data analysis.

We operationalised individual differences in cognitive abilities across the six cognitive tasks as a factor score of a latent variable, the 'g-factor.' To estimate this factor score, we fit the standardised performance of the six cognitive tasks to a second-order confirmatory factor analysis (CFA) of a 'g-factor' model, similar to previous work (*Ang et al., 2020*; *Pat et al., 2022a*; *Pat et al., 2022b*;

**Table 13.** Exclusion criteria for neuroimaging features in the baseline.

| Neuroimaging features | Data provided | Did not pass quality control | Had vision problems | From site 22 | Had any missing feature | Flagged as outliers | Observations kept |
|---|---|---|---|---|---|---|---|
| ENback 0back | 11771 | 3996 | 38 | 21 | 8 | 292 | 7416 |
| ENback 2back | 11771 | 3996 | 38 | 21 | 12 | 281 | 7423 |
| ENback 2back vs 0back | 11771 | 3996 | 38 | 21 | 10 | 397 | 7309 |
| ENback emotion | 11771 | 3996 | 38 | 21 | 10 | 303 | 7403 |
| ENback Emotion vs Neutral Face | 11771 | 3996 | 38 | 21 | 11 | 480 | 7225 |
| ENback Face vs Place | 11771 | 3996 | 38 | 21 | 10 | 391 | 7315 |
| ENback Negative vs Neutral Face | 11771 | 3996 | 38 | 21 | 11 | 454 | 7251 |
| ENback Positive vs Neutral Face | 11771 | 3996 | 38 | 21 | 10 | 500 | 7206 |
| ENback place | 11771 | 3996 | 38 | 21 | 11 | 331 | 7374 |
| MID Reward vs Neutral anticipation | 11771 | 2596 | 51 | 22 | 11 | 250 | 8841 |
| MID Loss vs Neutral anticipation | 11771 | 2596 | 51 | 22 | 11 | 245 | 8846 |
| MID Positive vs Negative Reward Feedback | 11771 | 2596 | 51 | 22 | 12 | 338 | 8752 |
| MID Positive vs Negative Punishment Feedback | 11771 | 2596 | 51 | 22 | 10 | 334 | 8758 |
| MID Large Reward vs Neutral anticipation | 11771 | 2596 | 51 | 22 | 12 | 241 | 8849 |
| MID Small Reward vs Neutral anticipation | 11771 | 2596 | 51 | 22 | 10 | 270 | 8822 |
| MID Large Reward vs Small Reward anticipation | 11771 | 2596 | 51 | 22 | 13 | 266 | 8823 |
| MID Large Loss vs Neutral anticipation | 11771 | 2596 | 51 | 22 | 11 | 250 | 8841 |
| MID Small Loss vs Neutral anticipation | 11771 | 2596 | 51 | 22 | 11 | 282 | 8809 |
| MID Large Loss vs Small Loss anticipation | 11771 | 2596 | 51 | 22 | 12 | 307 | 8783 |
| SST Any Stop vs Correct Go | 11771 | 3672 | 45 | 20 | 14 | 227 | 7793 |
| SST Correct Go vs Fixation | 11771 | 3672 | 45 | 20 | 13 | 262 | 7759 |
| SST Correct Stop vs Correct Go | 11771 | 3672 | 45 | 20 | 13 | 236 | 7785 |
| SST Correct Stop vs Incorrect Stop | 11771 | 3672 | 45 | 20 | 14 | 292 | 7728 |
| SST Incorrect Go vs Correct Go | 11771 | 3672 | 45 | 20 | 15 | 481 | 7538 |
| SST Incorrect Go vs Incorrect Stop | 11771 | 3672 | 45 | 20 | 14 | 366 | 7654 |
| SST Incorrect Stop vs Correct Go | 11771 | 3672 | 45 | 20 | 13 | 246 | 7775 |
| rsfMRI temporal variance | 11771 | 2397 | 62 | 25 | 14 | 682 | 8591 |
| rsfMRI subcortical-network FC | 11771 | 2397 | 62 | 25 | 14 | 1 | 9272 |
| rsfMRI cortical FC | 11771 | 2397 | 62 | 25 | 14 | 3 | 9270 |
| T1 subcortical avg intensity | 11771 | 501 | 66 | 27 | 0 | 60 | 11117 |
| T1 white matter avg intensity | 11771 | 501 | 66 | 27 | 12 | 13 | 11152 |
| T1 gray matter avg intensity | 11771 | 501 | 66 | 27 | 12 | 11 | 11154 |
| T1 normalised intensity | 11771 | 501 | 66 | 27 | 12 | 2 | 11163 |
| T1 summations | 11771 | 501 | 66 | 27 | 12 | 34 | 11131 |
| cortical thickness | 11771 | 501 | 66 | 27 | 12 | 2 | 11163 |
| cortical area | 11771 | 501 | 66 | 27 | 12 | 1 | 11164 |
| cortical volume | 11771 | 501 | 66 | 27 | 12 | 0 | 11165 |
| subcortical volume | 11771 | 501 | 66 | 27 | 0 | 215 | 10962 |
| sulcal depth | 11771 | 501 | 66 | 27 | 12 | 1106 | 10059 |
| T2 subcortical avg intensity | 11771 | 1217 | 58 | 25 | 0 | 67 | 10404 |
| T2 white matter avg intensity | 11771 | 1217 | 58 | 25 | 10 | 56 | 10405 |

*Table 13 continued on next page*

*Table 13 continued*

| Neuroimaging features | Data provided | Did not pass quality control | Had vision problems | From site 22 | Had any missing feature | Flagged as outliers | Observations kept |
|---|---|---|---|---|---|---|---|
| T2 gray matter avg intensity | 11771 | 1217 | 58 | 25 | 10 | 55 | 10406 |
| T2 normalised intensity | 11771 | 1217 | 58 | 25 | 10 | 12 | 10449 |
| T2 summations | 11771 | 1217 | 58 | 25 | 10 | 14 | 10447 |
| DTI | 11771 | 1577 | 57 | 13 | 0 | 24 | 10100 |

*Thompson et al., 2019*). In this CFA, we treated the g-factor as the second-order latent variable that underpinned three first-order latent variables, each with two manifest variables: (1) 'language,' underlying Picture Vocabulary and Oral Reading Recognition, (2) 'mental flexibility,' underlying Flanker and Pattern Comparison Processing, and (3) 'memory recall,' underlying Picture Sequence Memory and Rey-Auditory Verbal Learning.

We fixed the variance of the latent factors to one and applied the Maximum Likelihood with Robust standard errors (MLR) approach with Huber-White standard errors and scaled test statistics. To provide information about the internal consistency of the g-factor, we calculated OmegaL2 (*Jorgensen et al., 2022*). We used the *lavaan* (*Rosseel, 2012*) (version 0.6–15), *semTools* (*Jorgensen et al., 2022*), and *semPlots* (*Epskamp, 2015*) packages for this CFA of cognitive abilities.

We found the second-order 'g-factor' model to fit cognitive abilities well across the six cognitive tasks. This is evidenced by several indices if we apply the model to the whole baseline data: scaled and robust CFI (0.994), TLI (0.986), RMSEA (0.031, 90% CI [0.024-0.037]), robust SRMR (0.013), and OmegaL2 (0.78). See *Figure 7* for the standardised weights of this CFA model. This enabled us to use the factor score of the latent variable 'g-factor' as the target for our predictive models.

## Measures: mental health

Mental health was assessed using two sets of features. The first set involved parental reports of children's emotional and behavioural problems, as measured by the Child Behaviour Checklist (CBCL) (*Achenbach et al., 2017*). We used eight summary scores: anxious/depressed, withdrawn, somatic complaints, social problems, thought problems, attention problems, rule-breaking behaviours, and aggressive behaviours. For CBCL, caretakers rated each item as 0=not true (as far as you know), 1=somewhat or sometimes true, and 2=very true or often true. The third set assessed children's temperaments, conceptualised as risk factors for mental issues (*Johnson et al., 2003*; *Whiteside and Lynam, 2003*), using the Urgency, Premeditation, Perseverance, Sensation Seeking, and Positive Urgency (UPPS-P) Impulsive Behaviour Scale (*Zapolski et al., 2010*) and the Behavioural Inhibition System/Behavioural Activation System (BIS/BAS) (*Carver and White, 1994*). We used nine summary scores: negative urgency, lack of planning, sensation seeking, positive urgency, lack of perseverance, BIS, BAS reward responsiveness, BAS drive, and BAS fun. *Supplementary file 1* and *Supplementary file 2* provide summary statistics, histograms, and missing values for measures of mental health. They also include the actual variable names listed in the data dictionary and their calculations.

## Measures: neuroimaging

Neuroimaging data were based on the tabulated brain-MRI data pre-processed by the ABCD. We organized the brain-MRI data into 45 sets of neuroimaging features, covering task-fMRI (including ENBack, stop signal (SST), and monetary incentive delay (MID) tasks), resting-state fMRI, structural MRI, and diffusion tensor imaging (DTI). The ABCD provided details on MRI acquisition and image processing elsewhere (*Hagler et al., 2019*; *Yang and Jernigan, 2023*).

The ABCD study provided recommended exclusion criteria for neuroimaging data based on automated and manual quality control (*Yang and Jernigan, 2023*). Specifically, the study created an exclusion flag for each neuroimaging feature (with the prefix 'imgincl' in the 'abcd_imgincl01' table) based on criteria involving image quality, MR neurological screening, behavioural performance, and the number of repetition times (TRs), among others. We removed the entire set of neuroimaging features from each participant if any of its features were flagged or missing. We also detected outliers with over three interquartile ranges from the nearest quartile for each neuroimaging feature. We excluded a particular set of neuroimaging features from each participant when this set had outliers over 5% of

**Table 14.** Exclusion criteria for neuroimaging features in the follow-up.

| Neuroimaging features | Data provided | Did not pass quality control | Had vision problems | Had any missing feature | Flagged as outliers | Observations kept |
|---|---|---|---|---|---|---|
| ENback 0back | 8123 | 1804 | 35 | 11 | 216 | 6057 |
| ENback 2back | 8123 | 1804 | 35 | 13 | 186 | 6085 |
| ENback 2back vs 0back | 8123 | 1804 | 35 | 14 | 294 | 5976 |
| ENback emotion | 8123 | 1804 | 35 | 13 | 202 | 6069 |
| ENback Emotion vs Neutral Face | 8123 | 1804 | 35 | 13 | 347 | 5924 |
| ENback Face vs Place | 8123 | 1804 | 35 | 13 | 295 | 5976 |
| ENback Negative vs Neutral Face | 8123 | 1804 | 35 | 11 | 355 | 5918 |
| ENback Positive vs Neutral Face | 8123 | 1804 | 35 | 13 | 342 | 5929 |
| ENback place | 8123 | 1804 | 35 | 12 | 234 | 6038 |
| MID Reward vs Neutral anticipation | 8123 | 1379 | 40 | 8 | 153 | 6543 |
| MID Loss vs Neutral anticipation | 8123 | 1379 | 40 | 8 | 154 | 6542 |
| MID Positive vs Negative Reward Feedback | 8123 | 1379 | 40 | 9 | 192 | 6503 |
| MID Positive vs Negative Punishment Feedback | 8123 | 1379 | 40 | 9 | 197 | 6498 |
| MID Large Reward vs Neutral anticipation | 8123 | 1379 | 40 | 8 | 142 | 6554 |
| MID Small Reward vs Neutral anticipation | 8123 | 1379 | 40 | 8 | 163 | 6533 |
| MID Large Reward vs Small Reward anticipation | 8123 | 1379 | 40 | 8 | 155 | 6541 |
| MID Large Loss vs Neutral anticipation | 8123 | 1379 | 40 | 8 | 150 | 6546 |
| MID Smal Loss vs Neutral anticipation | 8123 | 1379 | 40 | 9 | 179 | 6516 |
| MID Large Loss vs Small Loss anticipation | 8123 | 1379 | 40 | 9 | 173 | 6522 |
| SST Any Stop vs Correct Go | 8123 | 2036 | 33 | 7 | 123 | 5924 |
| SST Correct Go vs Fixation | 8123 | 2036 | 33 | 7 | 173 | 5874 |
| SST Correct Stop vs Correct Go | 8123 | 2036 | 33 | 7 | 163 | 5884 |
| SST Correct Stop vs Incorrect Stop | 8123 | 2036 | 33 | 7 | 187 | 5860 |
| SST Incorrect Go vs Correct Go | 8123 | 2036 | 33 | 7 | 345 | 5702 |
| SST Incorrect Go vs Incorrect Stop | 8123 | 2036 | 33 | 7 | 267 | 5780 |
| SST Incorrect Stop vs Correct Go | 8123 | 2036 | 33 | 7 | 131 | 5916 |
| rsfMRI temporal variance | 8123 | 1152 | 49 | 14 | 512 | 6396 |
| rsfMRI subcortical-network FC | 8123 | 1152 | 49 | 14 | 3 | 6905 |
| rsfMRI cortical FC | 8123 | 1152 | 49 | 14 | 3 | 6905 |
| T1 subcortical avg intensity | 8123 | 227 | 51 | 0 | 32 | 7813 |
| T1 white matter avg intensity | 8123 | 227 | 51 | 10 | 8 | 7827 |
| T1 gray matter avg intensity | 8123 | 227 | 51 | 10 | 9 | 7826 |
| T1 normalised intensity | 8123 | 227 | 51 | 10 | 0 | 7835 |
| T1 summations | 8123 | 227 | 51 | 10 | 19 | 7816 |
| cortical thickness | 8123 | 227 | 51 | 10 | 2 | 7833 |
| cortical area | 8123 | 227 | 51 | 10 | 0 | 7835 |
| cortical volume | 8123 | 227 | 51 | 10 | 0 | 7835 |
| subcortical volume | 8123 | 227 | 51 | 0 | 112 | 7733 |
| sulcal depth | 8123 | 227 | 51 | 10 | 890 | 6945 |
| T2 subcortical avg intensity | 8123 | 600 | 50 | 0 | 39 | 7434 |
| T2 white matter avg intensity | 8123 | 600 | 50 | 10 | 47 | 7416 |
| T2 gray matter avg intensity | 8123 | 600 | 50 | 10 | 49 | 7414 |

*Table 14 continued on next page*

*Table 14 continued*

| Neuroimaging features | Data provided | Did not pass quality control | Had vision problems | Had any missing feature | Flagged as outliers | Observations kept |
|---|---|---|---|---|---|---|
| T2 normalised intensity | 8123 | 600 | 50 | 10 | 5 | 7458 |
| T2 summations | 8123 | 600 | 50 | 10 | 14 | 7449 |
| DTI | 8123 | 638 | 47 | 0 | 15 | 7423 |

the total number of its neuroimaging features. For instance, for the 2-Back vs 0-Back contrast from the ENBack task-fMRI, we had 167 features (i.e. brain regions) based on the brain parcellation atlas used by the ABCD. If (a) one of the 167 features had an exclusion flag, (b) a participant had a vision problem, (c) any of the 167 features was missing, (d) at least nine features (i.e. over 5%) were outliers, then we would remove this 2-Back vs 0-Back contrast from a particular participant but still keep other sets of neuroimaging features that did not meet these criteria (see –13 for the number of participants after each exclusion criterion for each set of neuroimaging features).

We standardised each neuroimaging feature across participants and harmonised variation across MRI scanners using ComBat (*Fortin et al., 2017*; *Johnson et al., 2007*; *Nielson et al., 2018*). Note that under predictive modelling, we discuss strategies we implemented to avoid data leakage and to model the data with missing values using the opportunistic stacking technique (*Engemann et al., 2020*; *Pat et al., 2022b*).

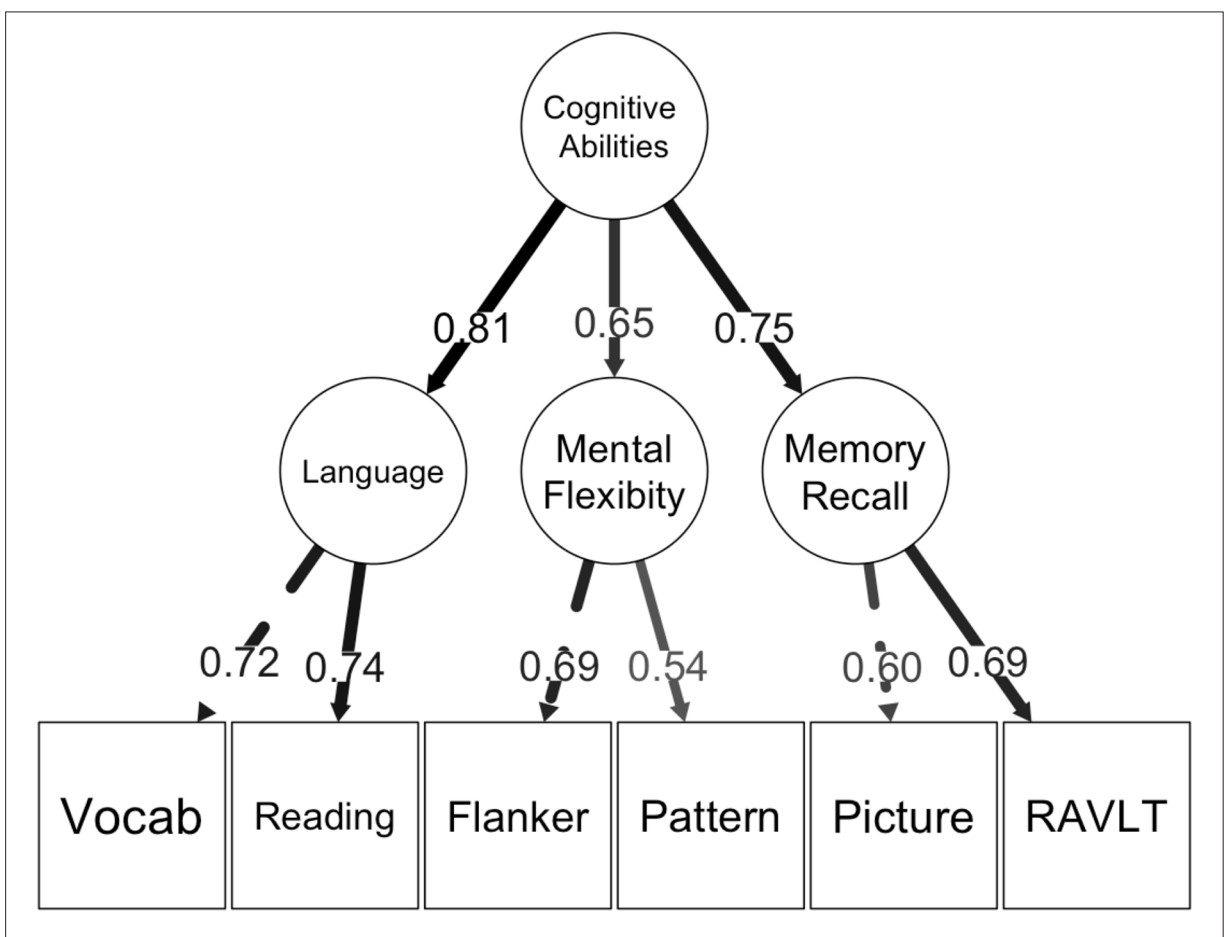

**Figure 7.** Standardised weights of the second-order 'g-factor' model. These weights were derived from confirmatory factor analysis, fitted on cognitive abilities across six cognitive tasks from the entire baseline dataset. The actual weights used for predictive modelling were slightly different, as the predictive modelling was based on leave-one-site-out cross-validation, which trained on data from all but one site.

## Sets of neuroimaging features 1-26: task-fMRI

We used unthresholded generalised-linear model (GLM) contrasts, averaged across two runs (*Bolt et al., 2017*; *Pat et al., 2023*; *Pat et al., 2022b*) for task-fMRI sets of features. These contrasts were embedded in the brain parcels based on the FreeSurfer's atlases (*Dale et al., 1999*): 148 cortical-surface Destrieux parcels (*Destrieux et al., 2010*) and subcortical-volumetric 19 ASEG parcels (*Fischl et al., 2002*), resulting in 167 features in each task-fMRI set.

## Sets of neuroimaging features 1-9: ENBack task-fMRI

The 'ENBack' or emotional n-back task was designed to elicit fMRI activity related to working memory to neutral and emotional stimuli (*Barch et al., 2013*). Depending on the block, the children were asked whether an image matched the image shown two trials earlier (2-Back) or at the beginning (0-Back). In this task version, the images shown included emotional faces and places. Thus, in addition to working memory, the task also allowed us to extract fMRI activity related to emotion processing and facial processing. We used the following contrasts as nine separate sets of neuroimaging features for ENBack task-fMRI: 2-Back vs 0-Back, Face vs Place, Emotion vs Neutral Face, Positive vs Neutral Face, Negative vs Neutral Face, 2-Back, 0-Back, Emotion, and Place.

## Sets of neuroimaging features 10-19: MID task-fMRI

The MID task was designed to elicit fMRI activity related to reward processing (*Knutson et al., 2000*). In this task, children responded to a stimulus shown on a screen. If they responded before the stimulus disappeared, they could either win $5 (Large Reward), win $0.2 (Small Reward), lose $5 (Large Loss), lose $0.2 (Small Loss), or not win or lose any money (Neutral), depending on the conditions. At the end of each trial, they were shown feedback on whether they won money (Positive Reward Feedback), did not win money (Negative Reward Feedback), avoided losing money (Positive Punishment Feedback), or lost money (Negative Punishment Feedback). We used the following contrasts as ten separate sets of neuroimaging features for MID task-fMRI: Large Reward vs Small Reward anticipation, Small Reward vs Neutral anticipation, Large Reward vs Neutral anticipation, Large Loss vs Small Loss anticipation, Small Loss vs Neutral anticipation, Large Loss vs Neutral anticipation, Loss vs Neutral anticipation, Reward vs Neutral anticipation, Positive vs Negative Reward Feedback, and Positive vs Negative Punishment Feedback.

## Sets of Neuroimaging Features 20-26: Stop-Signal Task (SST) task-fMRI

The SST was designed to elicit fMRI activity related to inhibitory control (*Whelan et al., 2012*). Children were asked to withhold or interrupt their motor response to a 'Go' stimulus whenever they saw a 'Stop' signal. We used two additional quality-control exclusion criteria for the SST task: *tfmri_sst_beh_glitchflag* and *tfmri_sst_beh_violatorflag*, which notified glitches as recommended (*Bissett et al., 2021*; *Garavan et al., 2018*). We used the following contrasts as seven separate sets of neuroimaging features for SST task-fMRI: Incorrect Go vs Incorrect Stop, Incorrect Go vs Correct Go, Correct Stop vs Incorrect Stop, Any Stop vs Correct Go, Incorrect Stop vs Correct Go, Correct Stop vs Correct Go, and Correct Go vs Fixation.

## Sets of neuroimaging features 27-29: rs-fMRI

The ABCD study collected rs-fMRI data for 20 min while children viewed a crosshair. The study described the pre-processing procedure elsewhere (*Hagler et al., 2019*). The investigators parcellated the cortical surface into 333 regions and the subcortical volume into 19 regions using Gordon's (*Gordon et al., 2016*) and ASEG (*Fischl et al., 2002*) atlases, respectively. They grouped the cortical-surface regions into 13 predefined large-scale cortical networks (*Gordon et al., 2016*). These large-scale cortical networks included auditory, cingulo-opercular, cingulo-parietal, default-mode, dorsal-attention, frontoparietal, none, retrosplenial-temporal, salience, sensorimotor-hand, sensorimotor-mouth, ventral-attention, and visual networks. Note that the term 'None' refers to regions that did not belong to any network. They then correlated time series from these regions and applied Fisher's z-transformation to the correlations. We included three sets of neuroimaging features for rs-fMRI. The first set was cortical functional connectivity (FC) with 91 features, including the mean values of the correlations between pairs of regions within the same large-scale cortical network and

between large-scale cortical networks. The second set was subcortical-network FC with 247 features, including the mean values of the correlations between each of the 19 subcortical regions and the 13 large-scale cortical networks. The third set was temporal variance with 352 features (i.e. 333 cortical and 19 subcortical regions), representing the variance across time calculated for each parcellated region. Temporal variance reflects the magnitude of low-frequency oscillations (*Yang and Jernigan, 2023*).

## Sets of neuroimaging features 30-44: sMRI

The ABCD study collected T1-weighted and T2-weighted 3D sMRI images and quantified them into various measures, mainly through FreeSurfer v7.1.1 (*Yang and Jernigan, 2023*). Similar to task-fMRI, we used 148 cortical-surface Destrieux (*Destrieux et al., 2010*) and subcortical-volumetric 19 ASEG (*Fischl et al., 2002*) atlases, resulting in 167 features. We included 15 sets of neuroimaging features for sMRI: cortical thickness, cortical area, cortical volume, sulcal depth, T1 white-matter averaged intensity, T1 grey-matter averaged intensity, T1 normalised intensity, T2 white-matter averaged intensity, T2 grey-matter averaged intensity, T2 normalised intensity, T1 summations, T2 summations, T1 subcortical averaged intensity, T2 subcortical averaged intensity and subcortical volume. Note: see *Figure 3* for the neuroimaging features included in T1 and T2 summations, and those figures are enlarged in *Figure 3—figure supplements 6 and 11* for baseline and follow-up respectively.

## Sets of neuroimaging features 45: DTI

We included fractional anisotropy (FA) derived from DTI as another set of neuroimaging features. FA characterizes the directionality of diffusion within white matter tracts, which is thought to indicate the density of fiber packing (*Alexander et al., 2007*). The ABCD study used AtlasTrack (*Hagler et al., 2009*; *Hagler et al., 2019*) to segment major white matter tracts. These included the corpus callosum, forceps major, forceps minor, cingulate and parahippocampal portions of the cingulum, fornix, inferior fronto-occipital fasciculus, inferior longitudinal fasciculus, pyramidal/corticospinal tract, superior longitudinal fasciculus, temporal lobe portion of the superior longitudinal fasciculus, anterior thalamic radiations, and uncinate. Given that ten tracts were separately labelled for each hemisphere, there were 23 neuroimaging features in this set.

## Measures: polygenic scores

Genetic profiles were constructed based on PGS of cognitive abilities. The ABCD study provides detailed notes on genotyping in another source (*Uban et al., 2018*). Briefly, the study genotyped saliva and whole blood samples using Smokescreen Array. The investigators then quality-controlled the data using calling signals and variant call rates, applied the Ricopili pipeline and imputed the data with TOPMED (see https://topmedimpute.readthedocs.io/). The study also identified problematic plates and data points with a subject-matching issue. Additional quality control was applied to these data, specifically excluding SNPs with a minor allele frequency of < 5%, removing SNPs with excessive deviation in Hardy-Weinberg Equilibrium of $p < 10^{-10}$, and finally remove individuals with excessive homozygosity / heterozygosity. Defined as heterozygosity observed at greater than 4 standard deviations above or below the sample mean.

We calculated PGS using three definitions from three large-scale genome-wide association studies (GWAS) on cognitive abilities: n=300,486 participants aged 16–102 (*Davies et al., 2018*), n=257,84 participants aged 8–96 (*Lee et al., 2018*) and n=269,867 participants aged 5–98 (*Savage et al., 2018*). These GWAS synthesised findings from different cohorts that collected cognitive tasks. Due to the diversity in cognitive tasks used across cohorts, they defined cognitive abilities in unique ways. For instance, *Lee et al., 2018* utilised principal component analysis to consolidate various cognitive task scores into a single measure within each cohort from the Cognitive Genomics Consortium (COGENT) consortia (*Lencz et al., 2014*), but only focused on the verbal-numerical reasoning (VNR) test within the UK Biobank cohort (*Sudlow et al., 2015*). In a similar approach, *Davies et al., 2018* employed principal component analysis to capture cognitive abilities from different cohorts within both CHARGE consortium data sets (*Psaty et al., 2009*) and COGENT (*Lencz et al., 2014*). They also focused on VNR testing within UK Biobank (*Sudlow et al., 2015*). Similarly, *Savage et al., 2018* calculated a singular score for cognitive abilities using 'a single sum score, mean score, or factor score' collated

from various tasks across thirteen cohort studies alongside logistic regression in one case-control study.

Participants in these GWAS were of European ancestry. Because PGS has a lower predictive ability when target samples (i.e. in our case, ABCD children) do not have the same ancestry as those of the discovery GWAS sample (*Duncan et al., 2019*), we restricted all analyses involving PGS to 5776 children of European ancestry. These children were within four standard deviations from the mean of the top four principal components (PCs) of the super-population individuals in the 1000 Genomes Project Consortium Phase 3 reference (*Auton et al., 2015*).

We employed the P-threshold approach (*Choi et al., 2020*). In this approach, we defined 'risk' alleles as those associated with cognitive abilities in the three discovery GWASs (*Davies et al., 2018*; *Lee et al., 2018*; *Savage et al., 2018*) at ten different PGS thresholds: 0.5, 0.1, 0.05, 0.01, 0.001, 0.0001, 0.00001, 0.000001, 0.0000001, 0.00000001. We then computed PGS as the Z-scored, weighted mean number of linkage-independent risk alleles in approximate linkage equilibrium derived from imputed autosomal SNPs. We selected the best PGS threshold for each of the three definitions by choosing the PGS threshold that demonstrated the strongest correlation between its PGS and cognitive abilities in the ABCD (i.e. the g-factor factor score). Refer to the section on predictive modelling below for strategies we implemented to avoid data leakage due to this selection of the PGS threshold and the family structure in the ABCD.

## Measures: sociodemographics, lifestyles, and developmental adverse events

Environmental factors were based on 44 features, covering socio-demographics, lifestyles, and developmental adverse events. This included (a) 14 features for child social-demographics (*Zucker et al., 2018*), including bilingual use (*Dick et al., 2019*), parental marital status, parental education, parental income, household size, economic insecurities, area deprivation index (*Kind et al., 2014*), lead risk (*Frostenson, 2016*), crime report (*Federal Bureau Of Investigation, 2012*), neighbourhood safety (*Echeverria et al., 2004*), school environment, involvement and disengagement (*Stover et al., 2010*), (b) five features for child social interactions from Parent Monitoring scale (*Chilcoat and Anthony, 1996*), Child Report of Behaviour Inventory (*Schaefer, 1965*), Strength and Difficulties Questionnaire (*Goodman et al., 2003*) and Moos Family Environment Scale (*Moos et al., 1974*), (c) eight features from child's sleep problems based on the Sleep Disturbance scale (*Bruni et al., 1996*), (d) four features for child's physical activities from Youth Risk Behaviour Survey (*Dolsen et al., 2019*; *Hunsberger et al., 2015*), (e) four features for child screen use (*Bagot et al., 2018*), (f) six features for parental use of alcohol, tobacco and marijuana before and after pregnancy from the Developmental History Questionnaire (*Kessler et al., 2009*; *Merikangas et al., 2009*), and (g) three features for developmental adverse events from the Developmental History Questionnaire, including prematurity and birth and pregnancy complications (*Kessler et al., 2009*; *Merikangas et al., 2009*). Note that we treated developmental adverse events from the Developmental History Questionnaire as environmental factors, as these events are either parental behaviours (e.g. parental use of alcohol, tobacco and marijuana) or parental medical conditions (e.g. pregnancy complications) that affect children. *Supplementary file 3* and *Supplementary file 4* provide summary statistics, histograms, and missing values for measures of socio-demographics, lifestyles and developmental adverse events. They also include the actual variable names listed in the data dictionary and their calculations.

## Predictive modelling

For building predictive multivariate models, we implemented a nested leave-one-site-out cross-validation. Specifically, we treated one out of 21 sites as a test set and the rest as a training set for training predictive models. We then repeated the model-building process until every site was a test set once and reported overall predictive performance across all test sites. Within each training set, we applied 10-fold cross-validation to tune the hyperparameters of the predictive models. The nested leave-one-site-out cross-validation allowed us to ensure the generalisability of our predictive models to unseen sites. This is important because different sites involved different MRI machines, experimenters, and participants of other demographics (*Garavan et al., 2018*). Next, data from children from the same family were collected from the same site. Accordingly, using leave-one-site-out also prevented data leakage due to family structure, which might inflate the predictive performance of the

models, particularly those involving polygenic scores. Still, given the different number of participants in each site, one drawback for the nested leave-one-site-out cross-validation is that we ended up with some test sets with fewer participants than others. Accordingly, we provided a supplemental analysis using the classical nested cross-validation, which included ten non-overlapping outer folds, randomly chosen without considering the site information, as test sets and ten inner folds for hyperparameter tuning (see *Figure 8*). Briefly, the results of the leave-one-site-out cross-validation and classical nested cross-validation were close to each other, albeit classical nested cross-validation having slightly higher performance.

To demonstrate the stability of the results across two years, we built the predictive models (including hyperparameter tuning) separately for baseline and follow-up data. We separately applied standardisation to the baseline training and test sets for both the target and features to prevent data leakage between training and test sets. To ensure similarity in the data scale across two time points, we used the mean and standard deviation of the baseline training and test sets to standardise the follow-up training and test sets, respectively. For cognitive abilities, which were used as the target for all predictive models, we applied this standardisation strategy both before CFA (i.e. to the behavioural performance of the six cognitive tasks) and after CFA (i.e. to the g-factor factor scores). Moreover, we only estimated the CFA of cognitive abilities using the baseline training set to ensure that the predictive models of the two time points had the same target. We then applied this estimated CFA model to the baseline test set and follow-up training and test sets. We examined the predictive performance of the models via the relationship between predicted and observed cognitive abilities, using Pearson's correlation ($r$), coefficient of determination ($R^2$, calculated using the sum of square definition), mean-absolute error (MAE), and root mean square error (RMSE).

## Predicting cognitive abilities from mental health

We developed predictive models to predict cognitive abilities from three sets of mental health features: CBCL and temperaments. We separately modelled each of these two sets and also simultaneously modelled the two sets by concatenating them into one set of features called 'mental health.' We implemented PLS (*Wold et al., 2001*) as a multivariate algorithm for these predictive models. Note that while PLS is sometimes used for reducing the dimensionality of features within a dataset, here we utilised PLS in a predictive framework: we tuned and estimated PLS loadings in each training set and applied the final model to the corresponding test set. PLS decomposes features into components that capture not only the features' variance but also the target's variance (*Wold et al., 2001*). PLS has an advantage in dealing with collinear features (*Dormann et al., 2013*), typical for mental health issues (*Caspi and Moffitt, 2018*).

PLS has one hyperparameter, the number of components. In our grid search, we tested the number of components, ranging from one to the total number of features. We selected the number of components based on the drop in root mean square error (RMSE). We kept increasing the number of components until the component did not reduce 0.1% of the total RMSE. We fit PLS using the *mixOmics* package (*Rohart et al., 2017*) with the *tidymodels* package as a wrapper (*Kuhn and Wickham, 2025*).

To understand how PLS made predictions, we examined loadings and the proportion of variance explained. Loadings for each PLS component show how much each feature contributes to each PLS component. The proportion of variance explained shows how much variance each PLS component captures compared to the total variance. We then compared loadings and the proportion of variance explained with the univariate Pearson's correlation between each feature and the target. Note that because we could not guarantee that each training set would result in the same PLS components, we calculated loadings and the proportion of variance explained on the full data without splitting them into training and test sets. It is important to note that the loadings and the proportion of variance explained are for understanding the models, but for assessing the predictive performance and computing a proxy measure of cognitive abilities (i.e., the predicted values), we still relied on the nested leave-one-site-out cross-validation.

## Predicting cognitive abilities from neuroimaging

We developed predictive models to predict cognitive abilities from 45 sets of neuroimaging features. To avoid data leakage, we detected the outliers separately in the baseline training, baseline test, follow-up training, and follow-up test sets. Similarly, to harmonise neuroimaging features across

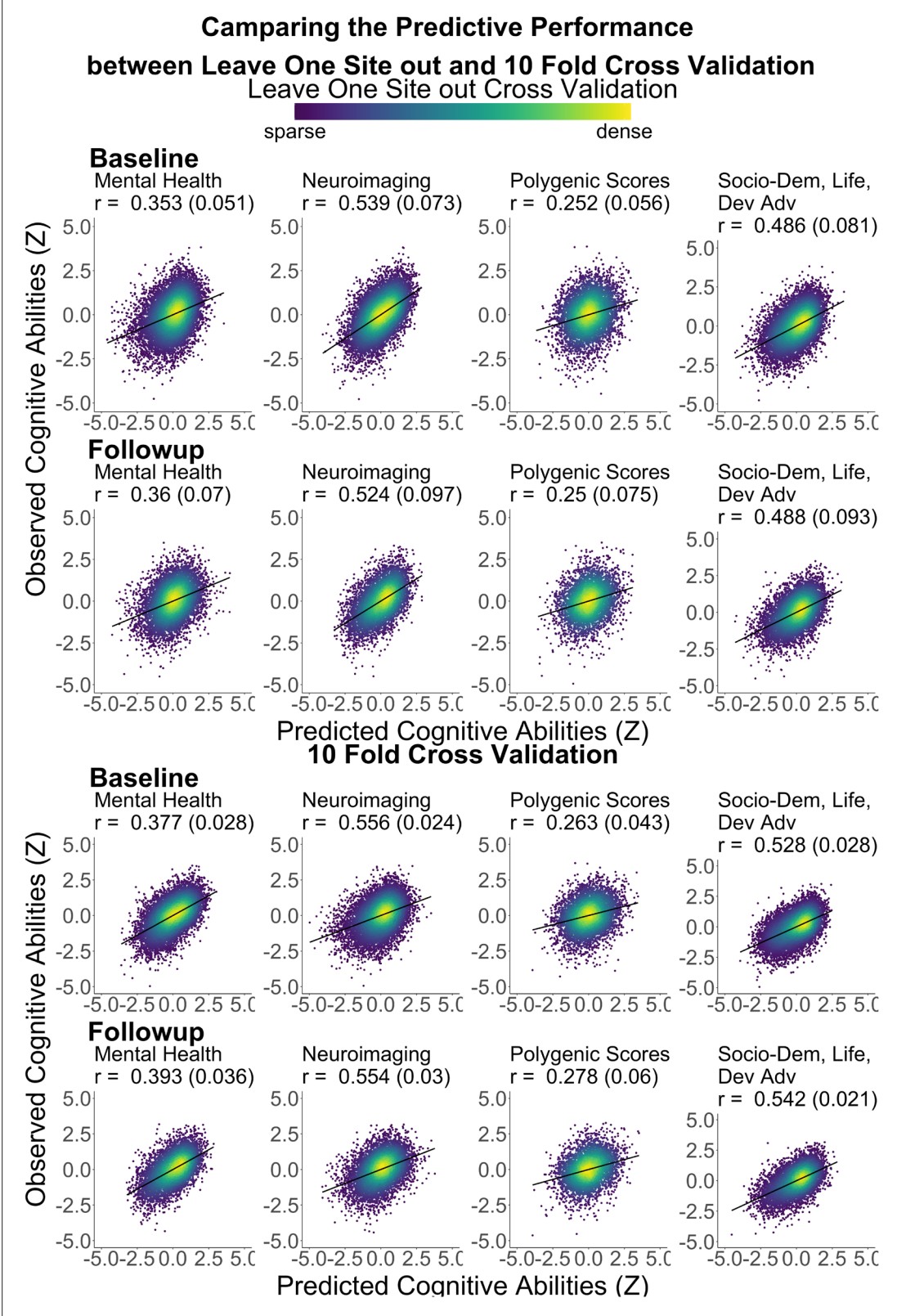

**Figure 8.** Predictive performance of leave one site out cross-validation vs 10-fold cross validation.

different sites while avoiding data leakage, we applied ComBat (*Fortin et al., 2017*; *Johnson et al., 2007*; *Nielson et al., 2018*) to the training set. We then applied ComBat to the test set, using the ComBatted training set as a reference batch.

Unlike PLS used above for predictive models from mental health, we chose to apply opportunistic stacking (*Engemann et al., 2020*; *Pat et al., 2022b*) when building predictive models from neuro-imaging. As we showed previously (*Pat et al., 2022b*), opportunistic stacking allowed us to handle missingness in the neuroimaging data without sacrificing predictive performance. Missingness in children's MRI data is expected, given high levels of noise (e.g. movement artifact) (*Fassbender et al., 2017*). For the ABCD, if we applied listwise exclusion using the study's exclusion criteria and outlier detection, we would have to exclude around 68% and 74%, at baseline and follow-up, respectively of the children with MRI data from any set of neuroimaging features flagged (*Pat et al., 2022b*) (see *Figure 9*). With opportunistic stacking, we only required each participant to have at least one out of 45 sets of neuroimaging features available. Therefore, we needed to exclude just around 9% and 41%, at baseline and follow-up respectively, of the children (see *Figure 9*). Our opportunistic stacking method kept 10,754 and 6412 participants at baseline and follow-up, respectively, while listwise deletion only kept 3784 and 2788 participants, respectively. We previously showed that the predictive performance of the models with opportunistic stacking is similar to that with listwise exclusion (*Pat et al., 2022b*).

Opportunistic stacking (*Engemann et al., 2020*; *Pat et al., 2022b*) involves two layers of modelling: set-specific and stacking layers. In the set-specific layer, we predicted cognitive abilities separately from each set of neuroimaging features using Elastic Net (*Zou and Hastie, 2005*). While being a linear and non-interactive algorithm, Elastic Net performs relatively well in predicting behaviours from neuroimaging MRI, often on par with, if not better than, other non-linear and interactive algorithms, such as support vector machine with non-linear kernel, XGBoost and Random Forest (*Pat et al., 2023*; *Tetereva et al., 2022*; *Vieira et al., 2022*). Moreover, Elastic Net coefficients are readily explainable, enabling us to explain how the models drew information from each neuroimaging feature when making a prediction (*Molnar, 2019*; *Pat et al., 2023*).

Elastic Net simultaneously minimises the weighted sum of the features' coefficients. Its loss function can be written as:

$$L_{\text{enet}}(\hat{\boldsymbol{\beta}}) = \frac{\sum_{i=1}^{n}(y_i - \mathbf{x}_i\hat{\boldsymbol{\beta}})^2}{2n} + \lambda \left( \frac{1-\alpha}{2} \sum_{j=1}^{m} \hat{\beta}_j^2 + \alpha \sum_{j=1}^{m} |\hat{\beta}_j| \right), \quad (1)$$

where $\mathbf{x_i}$ is a row vector of all the features in observation $i$, and $\beta$ is a column vector of features' coefficient. There are two hyperparameters: (1) the penalty $(\lambda)$ constraining the magnitude of the coefficients and (2) the mixture (α) deciding whether the model is more of a sum of squared coefficients (known as Ridge) or a sum of absolute values of the coefficients (known as Least Absolute Shrinkage and Selection Operator, LASSO). Using grid search, we chose the pair of penalty and mixture based on the lowest root mean square error (RMSE). The penalty was selected from 20 numbers, ranging from $10^{-10}$ to 10, equally spaced with the $log_{10}$ scale, and the penalty was selected from 11 numbers, ranging from 0 to 1 on a linear scale.

Training the set-specific layer resulted in the predicted values of cognitive abilities, one from each set of neuroimaging features. The stacking layer, then, took these predicted values across 45 sets of neuroimaging features and treated them as features to predict cognitive abilities, thereby drawing information across (as opposed to within) sets of neuroimaging features. Importantly, we used the same training set across both layers, ensuring no data leakage between training and test sets. Opportunistic stacking dealt with missing values from each set of neuroimaging features by, first, duplicating each feature (i.e. each of 45 predicted values from the set-specific layer) into two features, resulting in 90 features. We then replaced the missing values in the duplicated features with either unrealistically large (1000) or small (−1000) values. Accordingly, we could keep the data as long as at least one set of neuroimaging features had no missing value. Using these duplicated and imputed features, we predicted cognitive abilities from different sets of neuroimaging features using Random Forest (*Breiman, 2001*). Ultimately, the stacking layer resulted in a predicted value of cognitive abilities based on 45 sets of neuroimaging features.

Random Forest generates several regression trees by bootstrapping observations and including a random subset of features at each split (*Breiman, 2001*). To make a prediction, Random Forest

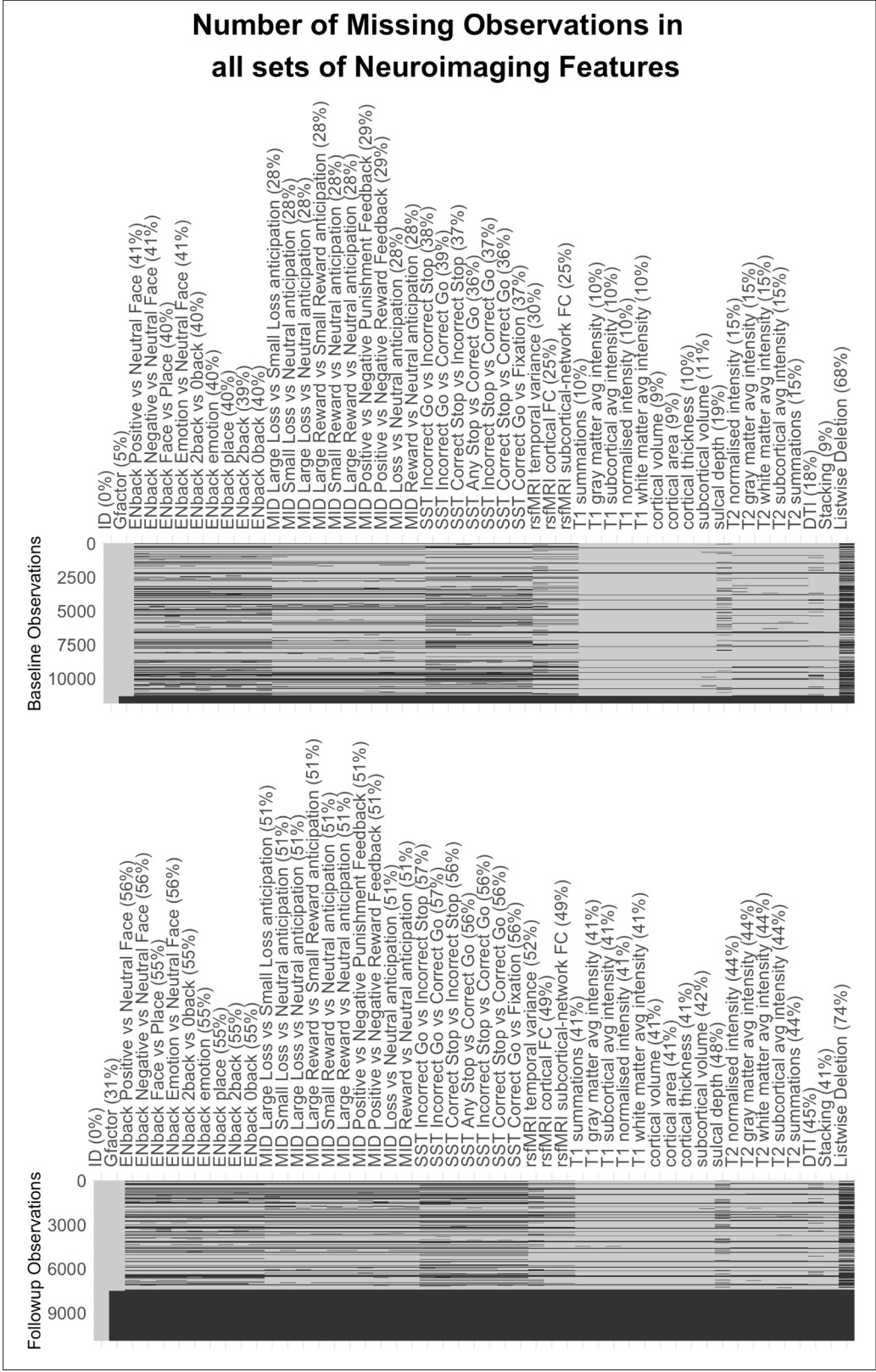

**Figure 9.** Illustration of data missingness (black) versus presence (grey) across different sets of neuroimaging features. This figure compares the number of observations in the analysis. Opportunist stacking (referred to as stacking here) requires only at least one neuroimaging feature to be present, thus allowing the inclusion of more neuroimaging features compared to listwise deletion.

aggregates predicted values across bootstrapped trees, known as bagging. We used 500 trees and turned two hyperparameters. First, 'mtry' was the number of features selected at each branch. Second, 'min_n' was the minimum number of observations in a node needed for the node to be split further. Using a Latin hypercube grid search of 3000 numbers (*Dupuy et al., 2015*; *Sacks et al., 1989*; *Santner et al., 2003*), we chose the pair of mtry, ranging from 1 to 90, and min_n, ranging from 2 to 2000, based on the lowest root mean square error (RMSE).

To understand how opportunistic stacking made predictions, we plotted Elastic Net coefficients for the set-specific layer and SHapley Additive exPlanations (SHAP) (*Lundberg and Lee, 2017*) for the stacking layer, averaged across 21 test sites. For the set-specific layer, Elastic Net made a prediction based on the linear summation of its regularised, estimated coefficients, and thus, plotting the coefficient of each neuroimaging feature allowed us to understand the contribution of such feature. For the stacking layer, it is difficult to trace the contribution from each feature from Random Forest directly, given the use of bagging. To overcome this, we computed Shapley values instead (*Roth, 1988*). Shapley values indicate the weighted differences in a model output when each feature is included versus not included in all possible subsets of features. SHAP (*Lundberg and Lee, 2017*) is a method to estimate Shapley values efficiently. Thus, SHAP allowed us to visualise the contribution of each set of neuroimaging features to the prediction in the stacking layer. Given that we duplicated the predicted values from each set of neuroimaging features in the stacking layer, we combined the magnitude of SHAP across the duplicates.

We fit Elastic Net and Random Forest using the *glmnet* (*Friedman et al., 2010*) and ranger (*Wright and Ziegler, 2017*) packages, respectively, with the *tidymodels* (*Kuhn and Wickham, 2025*) package as a wrapper. We approximated the Shapley values (*Lundberg and Lee, 2017*) using the *fastshap* package (*Greenwell, 2023*). The brain plots were created via the *ggseg, ggsegDesterieux, ggsegJHU, and ggsegGordon* packages (*Mowinckel and Vidal-Piñeiro, 2020*).

## Predicting cognitive abilities from polygenic scores

We developed predictive models to predict cognitive abilities from polygenic scores, as reflected by PGS of cognitive abilities from three definitions (*Davies et al., 2018*; *Lee et al., 2018*; *Savage et al., 2018*). We first selected the PGS threshold for each of the three definitions that demonstrated the strongest correlation with cognitive abilities within the training set. This left three PGSs as features for our predictive models, one for each definition. To control for population stratification in genetics, we regressed each PGS on four genetic principal components separately for the training and test sets. Later, we treated the residuals of this regression for each PGS as each feature in our predictive models. Similar to the predictive models for the set-specific layer of the neuroimaging features, we used Elastic Net here as an algorithm. Given that the genetic data do not change over time, we used the same genetic features for baseline and follow-up predictive models. We selected participants based on ancestry for predictive models involving polygenic scores, leaving us with a much smaller number of children (n=5776 vs. n=11,868 in the baseline).

## Predicting cognitive abilities from socio-demographics, lifestyles, and developmental adverse events

We developed predictive models to predict cognitive abilities from socio-demographics, lifestyles and developmental adverse events, reflected in the 44 features. We implemented PLS (*Wold et al., 2001*) as an algorithm similar to the mental health features. To deal with missing values, we applied the following steps separately for baseline training, baseline test, follow-up training, and follow-up test sets. We first imputed categorical features using mode and converted them into dummy variables. We then standardised all features and imputed them using K-nearest neighbours with five neighbours. Note that in a particular site, the value in a specific feature was at 0 for all of the observations (e.g., site 3 having a crime report at 0 for all children), making it impossible for us to standardise this feature when using this site as a test set. In this case, we kept the value of this feature at 0 and did not standardise it.

Note that the ABCD study only provided some features in the baseline, but not the follow-up. Accordingly, we treated these baseline features as features in our follow-up predictive models and combined them with the other collected in the follow-up. *Supplementary file 4* listed all of the variables and their calculation.

**Table 15.** Medication reports in the baseline and follow-up.
This report is derived from the su_y_plus table and utilises the Anatomical Therapeutic Chemical (ATC) Classification System to group medications according to their functionality.

| Functionality | Baseline | Follow-up |
|---|---|---|
| Alimentary tract and metabolism | 144 | 145 |
| Blood and blood forming organs | 12 | 22 |
| Cardiovascular system | 124 | 142 |
| Dermatologicals | 108 | 64 |
| Genitourinary system and sex hormones | 72 | 76 |
| Systemic hormonal preparations, excl. sex hormones and insulins | 26 | 24 |
| Anti-infectives for systemic use | 56 | 35 |
| Antineoplastic and immunomodulating agents | 5 | 5 |
| Musculo-skeletal system | 145 | 183 |
| Nervous system | 710 | 729 |
| Antiparasitic products, insecticides and repellents | 5 | 4 |
| Respiratory system | 721 | 538 |
| Sensory organs | 42 | 42 |
| Various | 1 | 3 |

## Commonality analyses

Following the predictive modelling procedure above, we extracted predicted values from different sets of features at each test site and treated them as proxy measures of cognitive abilities (*Dadi et al., 2021*). The out-of-sample relationship between observed and proxy measures of cognitive abilities based on specific features reflects variation in cognitive abilities explained by those features. For instance, the relationship between observed and proxy measures of cognitive abilities based on mental health indicates the variation in cognitive abilities that could be explained by mental health. Capitalising on this variation, we then used commonality analyses (*Nimon et al., 2008*) to demonstrate the extent to which other proxy measures captured similar variance of cognitive abilities as mental health.

First, to control for the influences of biological sex, age at interview and medication information, we residualised those variables from observed cognitive abilities and each proxy measure of cognitive abilities. We defined medication using the su_y_plus table and generated dummy variables based on the medication's functionality, as categorized by the Anatomical Therapeutic Chemical (ATC) Classification System (refer to *Table 15*). We then applied random-intercept, linear-mixed models (*Raudenbush and Bryk, 2002*) to the data from all test sites, using the *lme4* package (*Bates et al., 2015*). In these models, we considered families to be nested within each site, which allow different families from each site can have an unique intercept. We treated different proxy measures of cognitive abilities as fixed-effect regressors to explain cognitive abilities. We, then, estimated marginal $R^2$ from the linear-mixed models, which describes the variance explained by all fixed effects included in the models (*Nakagawa and Schielzeth, 2013*; *Vonesh et al., 1996*) and multiplied the marginal $R^2$ by 100 to obtain a percentage. By including and excluding each proxy measure in the models, we were able to decompose marginal $R^2$ into unique (i.e. attributed to the variance, uniquely explained by a particular proxy measure) and common (i.e. attributed to the variance, jointly explained by a group of proxy measures) effects (*Nimon et al., 2008*). We focused on the common effects between a proxy measure based on mental health and other proxy measures in four sets of commonality analyses. Note that each of the four sets of commonality analyses used different numbers of participants, depending on the data availability.

## Commonality analyses for proxy measures of cognitive abilities based on mental health and neuroimaging

Here, we included proxy measures of cognitive abilities based on mental health and/or neuroimaging. Specifically, for each proxy measure, we added two regressors in the models: the values centred within each site (denoted $cws$) and the site average (denoted $savg$). For instance, we applied the following *lme4* syntax for the models with both proxy measures:

$$\mathbf{y} = \beta_0 + \beta_1 \mathbf{x}_{mental\,health\,(cws)} + \beta_2 \mathbf{x}_{mental\,health\,(savg)} + \beta_3 \mathbf{x}_{neuroimaging\,(cws)}$$
$$+ \beta_4 \mathbf{x}_{neuroimaging\,(savg)} + \left(1 | site : family\right), \tag{2}$$

We computed unique and common effects (*Nimon et al., 2008*) as follows:

$$Unique_{mental\,health} = R^2_{mental\,health,\,neuroimaging} - R^2_{neuroimaging}$$
$$Unique_{neuroimaging} = R^2_{mental\,health,neuroimaging} - R^2_{mental\,health} \tag{3}$$
$$Common_{mental\,health,neuroimaging} = R^2_{mental\,health,neuroimaging} - Unique_{mental\,health} - Unique_{neuroimaging},$$

where the subscript of $R^2$ indicates which proxy measures were included in the model.

In addition to using the proxy measures based on neuroimaging from the stacking layer, we also conducted commonality analyses on proxy measures based on neuroimaging from each set of neuroimaging features. This allows us to demonstrate which sets of neuroimaging features showed higher common effects with the proxy measures based on mental health. Note that to include as many participants in the models as possible, we dropped missing values based on the availability of data in each set of neuroimaging features included in the models (i.e., not applying listwise deletion across sets of neuroimaging features).

## Commonality analyses for proxy measures of cognitive abilities based on mental health and polygenic scores

Here, we included proxy measures of cognitive abilities based on mental health and/or polygenic scores. Since family members had more similar genetics than non-members, we changed our centring strategy to polygenic scores. With the proxy measure based on polygenic scores, we applied (1) centring on two levels: centring its values within each family first and then within each site (denoted $cws, cwf$) (2) averaging on two levels: averaging of its values within each family first and then within each site (denoted $savg, favg$). Accordingly, we used the following *lme4* syntax for the models with both proxy measures:

$$\mathbf{y} = \beta_0 + \beta_1 \mathbf{x}_{mental\,health\,(cws)} + \beta_2 \mathbf{x}_{mental\,health\,(savg)} + \beta_3 \mathbf{x}_{PGS\,(cws,cwf)} + \beta_4 \mathbf{x}_{PGS\,(savg,favg)}$$
$$+ \left(1 | site : family\right) \tag{4}$$

We computed unique and common effects as follows:

$$Unique_{mental\,health} = R^2_{mental\,health,\,PGS} - R^2_{PGS}$$
$$Unique_{PGS} = R^2_{mental\,health,PGS} - R^2_{mental\,health} \tag{5}$$
$$Common_{mental\,health,PGS} = R^2_{mental\,health,PGS} - Unique_{mental\,health} - Unique_{PGS},$$

## Commonality analyses for proxy measures of cognitive abilities based on mental health and socio-demographics, lifestyles, and developmental adverse events

Here, we included proxy measures of cognitive abilities based on mental health and/or socio-demographics, lifestyles, and developmental adverse events. We applied the following *lme4* syntax for the models with both proxy measures:

$$\mathbf{y} = \beta_0 + \beta_1 \mathbf{x}_{mental\ health(cws)} + \beta_2 \mathbf{x}_{mental\ health(savg)} + \beta_3 \mathbf{x}_{soc\ lif\ dev(cws)} + \beta_4 \mathbf{x}_{soc\ lif\ dev(savg)}$$
$$+ \left(1 | site : family\right), \tag{6}$$

Where *soc lif dev* shorts for socio-demographics, lifestyles, and developmental adverse events. We computed unique and common effects (*Nimon et al., 2008*) as follows:

$$Unique_{mental\ health} = R^2_{mental\ health,\ soc\ lif\ dev} - R^2_{soc\ lif\ dev}$$
$$Unique_{soc\ lif\ dev} = R^2_{mental\ health,soc\ lif\ dev} - R^2_{mental\ health} \tag{7}$$
$$Common_{mental\ health,soc\ life\ dev} = R^2_{mental\ health,soc\ life\ dev} - Unique_{mental\ health} - Unique_{soc\ life\ dev},$$

## Commonality analyses for proxy measures of cognitive abilities based on mental health, neuroimaging, polygenic scores and socio-demographics, lifestyles, and developmental adverse events

Here, we included proxy measures of cognitive abilities based on mental health, neuroimaging, polygenic scores and/or socio-demographics, lifestyles, and developmental adverse events. We applied the following *lme4* syntax for the model with all proxy measures included:

$$\mathbf{y} = \beta_0 + \beta_1 \mathbf{x}_{mental\ health(cws)} + \beta_2 \mathbf{x}_{mental\ health(savg)} + \beta_3 \mathbf{x}_{neuroimaging(cws)} + \beta_4 \mathbf{x}_{neuroimaging(savg)}$$
$$+ \beta_5 \mathbf{x}_{PGS(cws,cwf)} + \beta_6 \mathbf{x}_{PGS(savg,favg)} + \beta_7 \mathbf{x}_{soc\ lif\ dev(cws)} + \beta_8 \mathbf{x}_{soc\ lif\ dev(savg)} + \left(1 | site : family\right), \tag{8}$$

We computed unique and common effects (*Nimon et al., 2008*; *Nimon et al., 2017*) as follows:

$$Unique_{mh} = R^2_{mh,b,s,g} - R^2_{b,s,g}$$
$$Unique_{b} = R^2_{mh,b,s,g} - R^2_{mh,s,g}$$
$$Unique_{s} = R^2_{mh,b,s,g} - R^2_{mh,b,g}$$
$$Unique_{g} = R^2_{mh,b,s,g} - R^2_{mh,b,s}$$
$$Common_{mh,b} = -R^2_{s,g} + R^2_{mh,s,g} + R^2_{b,s,g} - R^2_{mh,b,s,g}$$
$$Common_{mh,s} = -R^2_{b,g} + R^2_{mh,b,g} + R^2_{s,b,g} - R^2_{mh,b,s,g}$$
$$Common_{mh,g} = -R^2_{b,s} + R^2_{mh,b,s} + R^2_{s,b,g} - R^2_{mh,b,s,g}$$
$$Common_{b,s} = -R^2_{mh,g} + R^2_{mh,b,g} + R^2_{mh,s,g} - R^2_{mh,b,s,g}$$
$$Common_{b,g} = -R^2_{mh,s} + R^2_{mh,b,s} + R^2_{mh,s,g} - R^2_{mh,b,s,g} \tag{9}$$
$$Common_{mh,b,s} = -R^2_{g} + R^2_{mh,g} + R^2_{b,g} + R^2_{s,g} - R^2_{mh,b,g} - R^2_{mh,s,g} - R^2_{b,s,g} + R^2_{mh,b,s,g}$$
$$Common_{mh,b,g} = -R^2_{s} + R^2_{mh,s} + R^2_{b,s} + R^2_{s,g} - R^2_{mh,b,s} - R^2_{mh,s,g} - R^2_{b,s,g} + R^2_{mh,b,s,g}$$
$$Common_{mh,s,g} = -R^2_{b} + R^2_{mh,b} + R^2_{mh,s} + R^2_{b,g} - R^2_{mh,b,s} - R^2_{mh,b,g} - R^2_{b,s,g} + R^2_{mh,b,s,g}$$
$$Common_{b,s,g} = -R^2_{mh} + R^2_{mh,b} + R^2_{mh,s} + R^2_{mh,g} - R^2_{mh,b,s} - R^2_{mh,b,g} - R^2_{mh,s,g} + R^2_{mh,b,s,g}$$
$$Common_{mh,b,s,g} = R^2_{mh} + R^2_{b} + R^2_{s} + R^2_{g} - R^2_{mh,b} - R^2_{mh,s} - R^2_{mh,g} - R^2_{b,s} - R^2_{b,g} - R^2_{s,g} + R^2_{mh,b,s}$$
$$+ R^2_{mh,b,g} + R^2_{mh,s,g} + R^2_{b,s,g} - R^2_{mh,b,s,g},$$

where *mh*, *b*, *g*, and *s* denote mental health, brain (i.e. neuroimaging), genetic profile (i.e. polygenic scores) and/or socio-demographics, lifestyles, and developmental adverse events, respectively.

## Acknowledgements

Data used in the preparation of this article were obtained from the Adolescent Brain Cognitive Development (ABCD) Study (https://abcdstudy.org), held in the NIMH Data Archive (NDA). This is a multisite, longitudinal study designed to recruit more than 10,000 children aged 9–10 and follow them over 10 years into early adulthood. The ABCD Study is supported by the National Institutes of Health

and additional federal partners under award numbers U01DA041048, U01DA050989, U01DA051016, U01DA041022, U01DA051018, U01DA051037, U01DA050987, U01DA041174, U01DA041106, U01DA041117, U01DA041028, U01DA041134, U01DA050988, U01DA051039, U01DA041156, U01DA041025, U01DA041120, U01DA051038, U01DA041148, U01DA041093, U01DA041089, U24DA041123, U24DA041147. A full list of supporters is available at https://abcdstudy.org/federal-partners.html. A listing of participating sites and a complete listing of the study investigators can be found at https://abcdstudy.org/consortium_members/. ABCD consortium investigators designed and implemented the study and/or provided data but did not necessarily participate in the analysis or writing of this report. This manuscript reflects the views of the authors and may not reflect the opinions or views of the NIH or ABCD consortium investigators. The ABCD data repository grows and changes over time. The ABCD data used in this report came from DOI:10.15154/z563-zd24. The authors wish to acknowledge the use of New Zealand eScience Infrastructure (NeSI) high-performance computing facilities, consulting support, and/or training services as part of this research. New Zealand's national facilities are provided by NeSI and funded jointly by NeSI's collaborator institutions and through the Ministry of Business, Innovation & Employment's Research Infrastructure programme. URL https://www.nesi.org.nz. Yue Wang and Narun Pat were supported by Health Research Council Funding (21/618 and 24/838), the Neurological Foundation of New Zealand (grant number 2350 PRG) and the University of Otago.

## Additional information

### Funding

| Funder | Grant reference number | Author |
|---|---|---|
| Health Research Council of New Zealand | 21/618 | Narun Pat |
| Health Research Council of New Zealand | 24/838 | Narun Pat |
| Neurological Foundation of New Zealand | 2350 PRG | Narun Pat |
| University of Otago | | Narun Pat |

The funders had no role in study design, data collection and interpretation, or the decision to submit the work for publication.

### Author contributions

Yue Wang, Conceptualization, Data curation, Formal analysis, Visualization, Methodology, Writing – original draft; Richard Anney, Software, Validation; Narun Pat, Conceptualization, Resources, Data curation, Software, Formal analysis, Supervision, Funding acquisition, Validation, Methodology, Writing – original draft, Project administration, Writing – review and editing

### Author ORCIDs

Yue Wang ⬩ https://orcid.org/0009-0001-0444-8975
Richard Anney ⬩ https://orcid.org/0000-0002-6083-407X
Narun Pat ⬩ https://orcid.org/0000-0003-1459-5255

### Ethics

Human subjects: We used data from the Adolescent Brain Cognitive Development (ABCD) Study Curated Annual Release 5.1 (DOI:10.15154/z563-zd24) from two time points. Institutional Review Boards at each site approved the study protocols. Please see Clark et al., 2018 for ethical details, such as informed consent and confidentiality.

Reviewer #1 (Public review): https://doi.org/10.7554/eLife.105537.3.sa1
Reviewer #2 (Public review): https://doi.org/10.7554/eLife.105537.3.sa2
Author response https://doi.org/10.7554/eLife.105537.3.sa3

# Additional files

## Supplementary files

Supplementary file 1. Summary statistics of the measures of mental health in the baseline. Med = Median IQR = interquartile range; CV = Coefficient of variation; CBCL = Child Behavioural Checklist, reflecting children's emotional and behavioural problems; UPPS-P = Urgency, Premedition, Perseverance, Sensation seeking, and Positive urgency Impulsive Behaviour Scale; BAS = Behavioural Activation System. Under the variable names, there are information about the method to compute these variables and the original variables names in ABCD data dictionary.

Supplementary file 2. Summary statistics of the measures of mental health in the follow-up. Med = Median IQR = interquartile range; CV = Coefficient of variation; CBCL = Child Behavioural Checklist, reflecting children's emotional and behavioural problems; UPPS-P = Urgency, Premedition, Perseverance, Sensation seeking, and Positive urgency Impulsive Behaviour Scale; BAS = Behavioural Activation System. Under the variable names, there are information about the method to compute these variables and the original variables names in ABCD data dictionary.

Supplementary file 3. Summary statistics of the measures of socio-demographics, lifestyles and developmental adverse events in the baseline. Med = Median IQR = interquartile range; CV = Coefficient of variation. Under the variable names, there are information about the method to compute these variables and the original variables names in ABCD data dictionary.

Supplementary file 4. Summary statistics of the measures of socio-demographics, lifestyles and developmental adverse events in the follow-up. We only provided variables that were repeatedly correction in the follow-up here. Med = Median IQR = interquartile range; CV = Coefficient of variation. Under the variable names, there are information about the method to compute these variables and the original variables names in ABCD data dictionary.

MDAR checklist

## Data availability

We used publicly available ABCD 5.1 data (DOI: 10.15154/z563-zd24) provided by the ABCD study, held in the NIMH Data Archive. We uploaded the R analysis script and detailed outputs here (https://github.com/HAM-lab-Otago-University/Commonality-Analysis-ABCD5.1, copy archived at *HAM-lab-Otago-University, 2025*).

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
