## [Editor Report · eLife Assessment]

This **important** study examines the relationship between cognition and mental health and investigates how brain, genetics, and environmental measures mediate that relationship. The methods and results are **compelling** and well-executed. Overall, this study will be of interest in the field of population neuroscience and in studies of mental health.

---

## [Referee Report · Reviewer #1 (Public review)]

Summary:

This work integrates two timepoints from the Adolescent Brain Cognitive Development Study to understand how neuroimaging, genetic and environmental data contribute to the predictive power of mental health variables in predicting cognition in a large early adolescent sample. Their multimodal and multivariate prediction framework involves a novel opportunistic stacking model to handle complex types of information to predict variables that are important in understanding mental health-cognitive performance associations.

Strengths:

The authors are commended for incorporating and directly comparing the contribution of multiple imaging modalities (task fMRI, resting state fMRI, diffusion MRI, structural MRI), neurodevelopmental markers, environmental factors and polygenic risk scores in a novel multivariate framework (via opportunistic stacking), as well as interpreting mental health-cognition associations with latent factors derived from Partial Least Squares. The authors also use a large well-characterized and diverse cohort of adolescents from the Adolescent Brain Cognitive Development (ABCD) Study. The paper is also strengthened by commonality analyses to understand the shared and unique contribution of different categories of factors (e.g., neuroimaging vs mental health vs polygenic scores vs sociodemographic and adverse developmental events) in explaining variance in cognitive performance

Weaknesses:

The paper is framed with an over-reliance on the RDoC framework in the introduction, despite deviations from the RDoC framework in the methods. The field is also learning more about RDoC's limitations when mapping cognitive performance to biology. The authors also focus on a single general factor of cognition as the core outcome of interest as opposed to different domains of cognition. The authors could consider predicting mental health rather than cognition. Using mental health as a predictor could be limited by the included 9-11 year age range at baseline (where mental health concerns are likely to be low or not well captured), as well as the nature of how the data was collected, i.e., either by self-report or from parent/caregiver report.

Comments on revisions:

The authors have done an excellent job of addressing my comments. I have no other suggestions to add. Great work!

---

## [Referee Report · Reviewer #2 (Public review)]

Summary:

This paper by Wang et al. uses rich brain, behaviour, and genetics data from the ABCD cohort to ask how well cognitive abilities can be predicted from mental health related measures, and how brain and genetics influence that prediction. They obtain an out of sample correlation of 0.4, with neuroimaging (in particular task fMRI) proving the key mediator. Polygenic scores contributed less.

Strengths:

This paper is characterized by the intelligent use of a superb sample (ABCD) alongside strong statistical learning methods and a clear set of questions. The outcome - the moderate level of prediction between brain, cognition, genetics and mental health - is interesting, and particularly important is the dissection of which features best mediate that prediction and how developmental and lifestyle factors play a role.

Weaknesses:

There are relatively few weaknesses to this paper. It has already undergone review at a different journal, and the authors clearly took the original set of comments into account in revising their paper. Overall, while the ABCD sample is superb for the questions asked, it would have been highly informative to extend the analyses to datasets containing more participants with neurological/psychiatric diagnoses (e.g. HBN, POND) or extending it into adolescent/early adult onset psychopathology cohorts. But it is fair enough that the authors want to leave that for future work.

---

## [Author Response]

The following is the authors’ response to the original reviews

**Public Reviews:**

**Reviewer #1 (Public review):**
Summary:This work integrates two timepoints from the Adolescent Brain Cognitive Development (ABCD) Study to understand how neuroimaging, genetic, and environmental data contribute to the predictive power of mental health variables in predicting cognition in a large early adolescent sample. Their multimodal and multivariate prediction framework involves a novel opportunistic stacking model to handle complex types of information to predict variables that are important in understanding mental health-cognitive performance associations.Strengths:The authors are commended for incorporating and directly comparing the contribution of multiple imaging modalities (task fMRI, resting state fMRI, diffusion MRI, structural MRI), neurodevelopmental markers, environmental factors, and polygenic risk scores in a novel multivariate framework (via opportunistic stacking), as well as interpreting mental health-cognition associations with latent factors derived from partial least squares. The authors also use a large well-characterized and diverse cohort of adolescents from the ABCD Study. The paper is also strengthened by commonality analyses to understand the shared and unique contribution of different categories of factors (e.g., neuroimaging vs mental health vs polygenic scores vs sociodemographic and adverse developmental events) in explaining variance in cognitive performanceWeaknesses:The paper is framed with an over-reliance on the RDoC framework in the introduction, despite deviations from the RDoC framework in the methods. The field is also learning more about RDoC's limitations when mapping cognitive performance to biology. The authors also focus on a single general factor of cognition as the core outcome of interest as opposed to different domains of cognition. The authors could consider predicting mental health rather than cognition. Using mental health as a predictor could be limited by the included 9-11 year age range at baseline (where many mental health concerns are likely to be low or not well captured), as well as the nature of how the data was collected, i.e., either by self-report or from parent/caregiver report.

Thank you so much for your encouragement.

We appreciate your comments on the strengths of our manuscript.

Regarding the weaknesses, the reliance on the RDoC framework is by design. Even with its limitations, following RDoC allows us to investigate mental health holistically. In our case, RDoC enabled us to focus on (a) a functional domain (i.e., cognitive ability), (b) the biological units of analysis of this functional domain (i.e., neuroimaging and polygenic scores), (c) potential contribution of environments, and (d) the continuous individual deviation in this domain (as opposed to distinct categories). We are unaware of any framework with all these four features.

Focusing on modelling biological units of analysis of a functional domain, as opposed to mental health per se, has some empirical support from the literature. For instance, in Marek and colleagues’ (2022) study, as mentioned by a previous reviewer, fMRI is shown to have a more robust prediction for cognitive ability than mental health. Accordingly, our reasons for predicting cognitive ability instead of mental health in this study are motivated theoretically (i.e., through RDoC) and empirically (i.e., through fMRI findings). We have clarified this reason in the introduction of the manuscript.

We are aware of the debates surrounding the actual structure of functional domains where the originally proposed RDoC’s specific constructs might not fit the data as well as the data-driven approach (Beam et al., 2021; Quah et al., 2025). However, we consider this debate as an attempt to improve the characterisation of functional domains of RDoC, not an effort to invalidate its holistic, neurobiological and basicfunctioning approach. Our use of a latent-variable modelling approach through factor analyses moves towards a data-driven direction. We made the changes to the second-to-last paragraph in the introduction to make this point clear:

“In this study, inspired by RDoC, we (a) focused on cognitive abilities as a functional domain, (b) created predictive models to capture the continuous individual variation (as opposed to distinct categories) in cognitive abilities, (c) computed two neurobiological units of analysis of cognitive abilities: multimodal neuroimaging and PGS, and (d) investigated the potential contributions of environmental factors. To operationalise cognitive abilities, we estimated a latent variable representing behavioural performance across various cognitive tasks, commonly referred to as general cognitive ability or the gfactor (Deary, 2012). The g-factor was computed from various cognitive tasks pertinent to RDoC constructs, including attention, working memory, declarative memory, language, and cognitive control. However, using the g-factor to operationalise cognitive abilities caused this study to diverge from the original conceptualisation of RDoC, which emphasises studying separate constructs within cognitive abilities (Morris et al., 2022; Morris & Cuthbert, 2012). Recent studies suggest an improvement to the structure of functional domains by including a general factor, such as the g-factor, in the model, rather than treating each construct separately (Beam et al., 2021; Quah et al., 2025). The g-factor in children is also longitudinally stable and can forecast future health outcomes (Calvin et al., 2017; Deary et al., 2013). Notably, our previous research found that neuroimaging predicts the g-factor more accurately than predicting performance from separate individual cognitive tasks (Pat et al., 2023). Accordingly, we decided to conduct predictive models on the g-factor while keeping the RDoC’s holistic, neurobiological, and basic-functioning characteristics.”

**Reviewer #2 (Public review):**
Summary:This paper by Wang et al. uses rich brain, behaviour, and genetics data from the ABCD cohort to ask how well cognitive abilities can be predicted from mental-health-related measures, and how brain and genetics influence that prediction. They obtain an out-ofsample correlation of 0.4, with neuroimaging (in particular task fMRI) proving the key mediator. Polygenic scores contributed less.Strengths:This paper is characterized by the intelligent use of a superb sample (ABCD) alongside strong statistical learning methods and a clear set of questions. The outcome - the moderate level of prediction between the brain, cognition, genetics, and mental health - is interesting. Particularly important is the dissection of which features best mediate that prediction and how developmental and lifestyle factors play a role.

Thank you so much for the encouragement.

Weaknesses:There are relatively few weaknesses to this paper. It has already undergone review at a different journal, and the authors clearly took the original set of comments into account in revising their paper. Overall, while the ABCD sample is superb for the questions asked, it would have been highly informative to extend the analyses to datasets containing more participants with neurological/psychiatric diagnoses (e.g. HBN, POND) or extend it into adolescent/early adult onset psychopathology cohorts. But it is fair enough that the authors want to leave that for future work.

Thank you very much for providing this valuable comment and for your flexibility.

For the current manuscript, we have drawn inspiration from the RDoC framework, which emphasises the variation from normal to abnormal in normative samples (Morris et al., 2022). The ABCD samples align well with this framework.

We hope to extend this framework to include participants with neurological and psychiatric diagnoses in the future. We have begun applying neurobiological units of analysis for cognitive abilities, assessed through multimodal neuroimaging and polygenic scores (PGS), to other datasets containing more participants with neurological and psychiatric diagnoses. However, this is beyond the scope of the current manuscript. We have listed this as one of the limitations in the discussion section:

“Similarly, our ABCD samples were young and community-based, likely limiting the severity of their psychopathological issues (Kessler et al., 2007). Future work needs to test if the results found here are generalisable to adults and participants with stronger severity.”

In terms of more practical concerns, much of the paper relies on comparing r or R2 measures between different tests. These are always presented as point estimates without uncertainty. There would be some value, I think, in incorporating uncertainty from repeated sampling to better understand the improvements/differences between the reported correlations.

This is a good suggestion. We have now included bootstrapped 95% confidence intervals in all of our scatter plots, showing the uncertainty of predictive performance.

The focus on mental health in a largely normative sample leads to the predictions being largely based on the normal range. It would be interesting to subsample the data and ask how well the extremes are predicted.

We appreciate this comment. Similar to our response to Reviewer 2’s Weakness #1, our approach has drawn inspiration from the RDoC framework, which emphasises the variation from normal to abnormal in normative samples (Morris et al., 2022). Subsampling the data would make us deviate from our original motivation.

Moreover, we used 17 mental healh variables in our predictive models: 8 CBCL subscales, 4 BIS/BAS subscales and 5 UPSS subscales. It is difficult to subsample them. Perhaps a better approach is to test the applicability of our neurobiological units of analysis for cognitive abilities (multimodal neuroimaging and PGS) in other datasets that include more extreme samples. We are working on this line of studies at the moment, and hope to show that in our future work.

Reviewer 2’s Weakness #4

A minor query - why are only cortical features shown in Figure 3?

We presented both cortical and subcortical features in Figure 3. The cortical features are shown on the surface space, while the subcortical features are displayed on the coronal plane. Below is an example of these cortical and subcortical features from the ENBack contrast. The subcortical features are presented in the far-right coronal image.

We separated the presentation of cortical and subcortical features because the ABCD uses the CIFTI format (https://www.humanconnectome.org/software/workbenchcommand/-cifti-help). CIFTI-format images combine cortical surface (in vertices) with subcortical volume (in voxels). For task fMRI, the ABCD parcellated cortical vertices using Freesurfer’s Destrieux atlas and subcortical voxels using Freesurfer’s automatically segmented brain volume (ASEG).

Due to the size of the images in Figure 3, it may have been difficult for Reviewer 2 to see the subcortical features clearly. We have now added zoomed-in versions of this figure as Supplementary Figures 4–13.

**Recommendations for the authors:**

**Reviewer #1 (Recommendations for the autors):**
(1) In the abstract, could the authors mention which imaging modalities contribute most to the prediction of cognitive abilities (e.g., working memory-related task fMRI)?

Thank you for the suggestion. Following this advice, we now mention which imaging modalities led to the highest predictive performance. Please see the abstract below.

“Cognitive abilities are often linked to mental health across various disorders, a pattern observed even in childhood. However, the extent to which this relationship is represented by different neurobiological units of analysis, such as multimodal neuroimaging and polygenic scores (PGS), remains unclear.

Using large-scale data from the Adolescent Brain Cognitive Development (ABCD) Study, we first quantified the relationship between cognitive abilities and mental health by applying multivariate models to predict cognitive abilities from mental health in children aged 9-10, finding an out-of-sample *r*=.36 . We then applied similar multivariate models to predict cognitive abilities from multimodal neuroimaging, polygenic scores (PGS) and environmental factors. Multimodal neuroimaging was based on 45 types of brain MRI (e.g., task fMRI contrasts, resting-state fMRI, structural MRI, and diffusion tensor imaging). Among these MRI types, the fMRI contrast, 2-Back vs. 0-Back, from the ENBack task provided the highest predictive performance (*r*=.4). Combining information across all 45 types of brain MRI led to the predictive performance of *r*=.54. The PGS, based on previous genome-wide association studies on cognitive abilities, achieved a predictive performance of *r*=.25. Environmental factors, including socio-demographics (e.g., parent’s income and education), lifestyles (e.g., extracurricular activities, sleep) and developmental adverse events (e.g., parental use of alcohol/tobacco, pregnancy complications), led to a predictive performance of *r*=.49.

In a series of separate commonality analyses, we found that the relationship between cognitive abilities and mental health was primarily represented by multimodal neuroimaging (66%) and, to a lesser extent, by PGS (21%). Additionally, environmental factors accounted for 63% of the variance in the relationship between cognitive abilities and mental health. The multimodal neuroimaging and PGS then explained 58% and 21% of the variance due to environmental factors, respectively. Notably, these patterns remained stable over two years.

Our findings underscore the significance of neurobiological units of analysis for cognitive abilities, as measured by multimodal neuroimaging and PGS, in understanding both (a) the relationship between cognitive abilities and mental health and (b) the variance in this relationship shared with environmental factors.”

(2) Could the authors clarify what they mean by "completing the transdiagnostic aetiology of mental health" in the introduction? (Second paragraph).

Thank you.

We intended to convey that understanding the transdiagnostic aetiology of mental health would be enhanced by knowing how neurobiological units of cognitive abilities, from the brain to genes, capture variations due to environmental factors. We realise this sentence might be confusing. Removing it does not alter the intended meaning of the paragraph, as we clarified this point later. The paragraph now reads:

“According to the National Institute of Mental Health’s Research Domain Criteria (RDoC) framework (Insel et al., 2010), cognitive abilities should be investigated not only behaviourally but also neurobiologically, from the brain to genes. It remains unclear to what extent the relationship between cognitive abilities and mental health is represented in part by different neurobiological units of analysis -- such as neural and genetic levels measured by multimodal neuroimaging and polygenic scores (PGS). To fully comprehend the role of neurobiology in the relationship between cognitive abilities and mental health, we must also consider how these neurobiological units capture variations due to environmental factors, such as sociodemographics, lifestyles, and childhood developmental adverse events (Morris et al., 2022). Our study investigated the extent to which (a) environmental factors explain the relationship between cognitive abilities and mental health, and (b) cognitive abilities at the neural and genetic levels capture these associations due to environmental factors. Specifically, we conducted these investigations in a large normative group of children from the ABCD study (Casey et al., 2018). We chose to examine children because, while their emotional and behavioural problems might not meet full diagnostic criteria (Kessler et al., 2007), issues at a young age often forecast adult psychopathology (Reef et al., 2010; Roza et al., 2003). Moreover, the associations among different emotional and behavioural problems in children reflect transdiagnostic dimensions of psychopathology (Michelini et al., 2019; Pat et al., 2022), making children an appropriate population to study the transdiagnostic aetiology of mental health, especially within a framework that emphasises normative variation from normal to abnormal, such as the RDoC (Morris et al., 2022).“

(3) It is unclear to me what the authors mean by this statement in the introduction: "Note that using the word 'proxy measure' does not necessarily mean that the predictive model for a particular measure has a high predictive performance - some proxy measures have better predictive performance than others".

We added this sentence to address a previous reviewer’s comment: “The authors use the phrasing throughout 'proxy measures of cognitive abilities' when they discuss PRS, neuroimaging, sociodemographics/lifestyle, and developmental factors. Indeed, the authors are able to explain a large proportion of variance with different combinations of these measures, but I think it may be a leap to call all of these proxy measures of cognition. I would suggest keeping the language more objective and stating these measures are associated with cognition.”

Because of this comment, we assumed that the reviewers wanted us to avoid the misinterpretation that a proxy measure implies high predictive performance. This term is used in machine learning literature (for instance, Dadi et al., 2021). We added the aforementioned sentence to ensure readers that using the term 'proxy measure' does not necessarily mean that the predictive model for a particular measure has high predictive performance. However, it seems that our intention led to an even more confusing message. Therefore, we decided to delete that sentence but keep an earlier sentence that explains the meaning of a proxy measure (see below).

“With opportunistic stacking, we created a ‘proxy’ measure of cognitive abilities (i.e., predicted value from the model) at the neural unit of analysis using multimodal neuroimaging.”

(4) Overall, despite comments from reviewers at another journal, I think the authors still refer to RDoC more than needed in the intro given the restructuring of the manuscript. For instance, at the end of page 4 and top of page 5, it becomes a bit confusing when the authors mention how they deviated from the RDoC framework, but their choice of cognitive domains is still motivated by RDoC. I think the chosen cognitive constructs are consistent with what is in ABCD and what other studies have incorporated into the g factor and do not require the authors to further justify their choice through RDoC. Also, there is emerging work showing that RDoC is limited in its ability to parse apart meaningful neuroimaging-based patterns; see for instance, Quah et al., Nature 2025 (https://doi.org/10.1038/s41467-025-55831-z).

Thank you very much for your comment. We have addressed it in our Response to Reviewer 1’s summary, strengths, and weaknesses above. We have rewritten the paragraph to clarify the relevance of our work to the RDoC framework and to recent studies aiming to improve RDoC constructs (including that from Quah and colleagues).

(5) I am still on the fence about the use of 'proxy measures of cognitive abilities' given that it is defined as the predictive performance of mental health measures in predicting cognition - what about just calling these mental health predictors? Also, it would be easier to follow this train of thought throughout the manuscript. But I leave it to the authors if they decide to keep their current language of 'proxy measure of cognition'.

Thank you so much for your flexibility. As we explained previously, this ‘proxy measures’ term is used in machine learning literature (for instance, Dadi et al., 2021). We thought about other terms, such as “score”, which is used in genetics, i.e., polygenic scores (Choi et al., 2020). and has recently been used in neuroimaging, i.e., neuroscore (Rodrigue et al., 2024). However, using a ‘score’ is a bit awkward for mental health and socio-demographics, lifestyle and developmental adverse events. Accordingly, we decided to keep the term ‘proxy measures’.

(6) It is unclear which cognitive abilities are being predicted in Figure 1, given the various domains that authors describe in their intro. Is it the g-factor from CFA? This should be clarified in all figure captions.

Yes, cognitive abilities are operationalised using a second-order latent variable, the g-factor from a CFA. We now added the following sentence to Figure 1, 2, 4 to make this point clearer. Thank you for the suggestion:

“Cognitive abilities are based on the second-order latent variable, the g-factor, based on a confirmatory factor analysis of six cognitive tasks.”

(7) I think it may also be worthwhile to showcase the explanatory power cognitive abilities have in predicting mental health or at least comment on this in the discussion. Certainly, there may be a bidirectional relationship here. The prediction direction from cognition to mental health may be an altogether different objective than what the paper currently presents, but many researchers working in psychiatry may take the stance (with support from the literature) that cognitive performance may serve as premorbid markers for later mental health concerns, particularly given the age range that the authors are working with in ABCD.

Thank you for this comment.

It is important to note that we do not make a directional claim in these cross-sectional analyses. The term "prediction" is used in a machine learning sense, implying only that we made an out-of-sample prediction (Yarkoni & Westfall, 2017). Specifically, we built predictive models on some samples (i.e., training participants) and applied our models to test participants who were not part of the model-building process. Accordingly, our predictive models cannot determine whether mental health “causes” cognitive abilities or vice versa, regardless of whether we treat mental health or cognitive abilities as feature/explanatory/independent variables or as target/response/outcome variables in the models. To demonstrate directionality, we would need to conduct a longitudinal analysis with many more repeated samples and use appropriate techniques, such as a cross-lagged panel model. It is beyond the scope of this manuscript and will need future releases of the ABCD data.

We decided to use cognitive abilities as a target variable here, rather than a feature variable, mainly for theoretical reasons. This work was inspired by the RDoC framework, which emphasises functional domains. Cognitive abilities is the functional domain in the current study. We created predictive models to predict cognitive abilities based on (a) mental health, (b) multimodal neuroimaging, (c) polygenic scores, and (d) environmental factors. We could not treat cognitive abilities as a functional domain if we used them as a feature variable. For instance, if we predicted mental health (instead of cognitive abilities) from multimodal neuroimaging and polygenic scores, we would no longer capture the neurobiological units of analysis for cognitive abilities.

We now made it clearer in the discussion that our use of predictive models cannot provide the directional of the effects

“Our predictive modelling revealed a medium-sized predictive relationship between cognitive abilities and mental health. This finding aligns with recent meta-analyses of case-control studies that link cognitive abilities and mental disorders across various psychiatric conditions (Abramovitch et al., 2021; East-Richard et al., 2020). Unlike previous studies, we estimated the predictive, out-of-sample relationship between cognitive abilities and mental disorders in a large normative sample of children. Although our predictive models, like other cross-sectional models, cannot determine the directionality of the effects, the strength of the relationship between cognitive abilities and mental health estimated here should be more robust than when calculated using the same sample as the model itself, known as in-sample prediction/association (Marek et al., 2022; Yarkoni & Westfall, 2017). Examining the PLS loadings of our predictive models revealed that the relationship was driven by various aspects of mental health, including thought and externalising symptoms, as well as motivation. This suggests that there are multiple pathways—encompassing a broad range of emotional and behavioural problems and temperaments—through which cognitive abilities and mental health are linked.”

(8) There is a lot of information packed into Figure 3 in the brain maps; I understand the authors wanted to fit this onto one page, and perhaps a higher resolution figure would resolve this, but the brain maps are very hard to read and/or compare, particularly the coronal sections.

Thank you for this suggestion. We agree with Reviewer 1 that we need to have a better visualisation of the feature-importance brain maps. To ensure that readers can clearly see the feature importance, we added a Zoom-in version of the feature-importance brain maps as Supplementary Figures 4 – 13.

(9) It would be helpful for authors to cluster features in the resting state functional connectivity correlation matrices, and perhaps use shorter names/acronyms for the labels.

Thank you for this suggestion.

We have now added a zoomed-in version of the feature importance for rs-fmri as Supplementary Figure 7 (for baseline) and 12 (for follow-up).

(10) (Figures 4a and 4b): please elaborate on "developmental adverse" in the title. I am assuming this is referring to childhood adverse events, or "developmental adversities".

Thank you so much for pointing this out. We meant ‘developmental adverse events’. We have made changes to this figure in the current manuscript.

(11) For the "follow-up" analyses, I would recommend the authors present this using only the features that are indeed available at follow-up, even if the list of features is lower, otherwise it becomes a bit confusing with the mix of baseline and follow-up features. Or perhaps the authors could make this more clear in the figures by perhaps having a different color for baseline vs follow-up features along the y-axis labels.

Thank you for this advice. We have now added an indicator in the plot to show whether the features were collected in the baseline or follow-up. We also added colours to indicate which type of environmental factors they were. It is now clear that the majority of the features that were collected at baseline, but were used for the followup predictive model, were developmental adverse events.

(12) Minor: Makowski et al 2023 reference can be updated to Makowski et al 2024, published in Cerebral Cortex.

Thank you for pointing this out. We have updated the citation accordingly.

References

Abramovitch, A., Short, T., & Schweiger, A. (2021). The C Factor: Cognitive dysfunction as a transdiagnostic dimension in psychopathology. Clinical Psychology Review, 86, 102007. https://doi.org/10.1016/j.cpr.2021.102007

Beam, E., Potts, C., Poldrack, R. A., & Etkin, A. (2021). A data-driven framework for mapping domains of human neurobiology. Nature Neuroscience, 24(12), 1733–1744. https://doi.org/10.1038/s41593-021-00948-9

Calvin, C. M., Batty, G. D., Der, G., Brett, C. E., Taylor, A., Pattie, A., Čukić, I., & Deary, I. J. (2017). Childhood intelligence in relation to major causes of death in 68 year follow-up: Prospective population study. BMJ, j2708. https://doi.org/10.1136/bmj.j2708

Casey, B. J., Cannonier, T., Conley, M. I., Cohen, A. O., Barch, D. M., Heitzeg, M. M., Soules, M. E., Teslovich, T., Dellarco, D. V., Garavan, H., Orr, C. A., Wager, T. D., Banich, M. T., Speer, N. K., Sutherland, M. T., Riedel, M. C., Dick, A. S., Bjork, J. M., Thomas, K. M., … ABCD Imaging Acquisition Workgroup. (2018). The Adolescent Brain Cognitive Development (ABCD) study: Imaging acquisition across 21 sites. Developmental Cognitive Neuroscience, 32, 43–54. https://doi.org/10.1016/j.dcn.2018.03.001

Choi, S. W., Mak, T. S.-H., & O’Reilly, P. F. (2020). Tutorial: A guide to performing polygenic risk score analyses. Nature Protocols, 15(9), Article 9. https://doi.org/10.1038/s41596-020-0353-1

Dadi, K., Varoquaux, G., Houenou, J., Bzdok, D., Thirion, B., & Engemann, D. (2021). Population modeling with machine learning can enhance measures of mental health. GigaScience, 10(10), giab071. https://doi.org/10.1093/gigascience/giab071

Deary, I. J. (2012). Intelligence. Annual Review of Psychology, 63(1), 453–482. https://doi.org/10.1146/annurev-psych-120710-100353

Deary, I. J., Pattie, A., & Starr, J. M. (2013). The Stability of Intelligence From Age 11 to Age 90 Years: The Lothian Birth Cohort of 1921. Psychological Science, 24(12), 2361–2368. https://doi.org/10.1177/0956797613486487

East-Richard, C., R. -Mercier, A., Nadeau, D., & Cellard, C. (2020). Transdiagnostic neurocognitive deficits in psychiatry: A review of meta-analyses. Canadian Psychology / Psychologie Canadienne, 61(3), 190–214. https://doi.org/10.1037/cap0000196

Insel, T., Cuthbert, B., Garvey, M., Heinssen, R., Pine, D. S., Quinn, K., Sanislow, C., & Wang, P. (2010). Research Domain Criteria (RDoC): Toward a New Classification Framework for Research on Mental Disorders. American Journal of Psychiatry, 167(7), 748–751. https://doi.org/10.1176/appi.ajp.2010.09091379

Kessler, R. C., Amminger, G. P., Aguilar-Gaxiola, S., Alonso, J., Lee, S., & Üstün, T. B. (2007). Age of onset of mental disorders: A review of recent literature. Current Opinion in Psychiatry, 20(4). https://journals.lww.com/co-psychiatry/abstract/2007/07000/age_of_onset_of_mental_disorders__a_review_of.10.aspx

Marek, S., Tervo-Clemmens, B., Calabro, F. J., Montez, D. F., Kay, B. P., Hatoum, A. S., Donohue, M. R., Foran, W., Miller, R. L., Hendrickson, T. J., Malone, S. M., Kandala, S., Feczko, E., Miranda-Dominguez, O., Graham, A. M., Earl, E. A., Perrone, A. J., Cordova, M., Doyle, O., … Dosenbach, N. U. F. (2022). eproducible brain-wide association studies require thousands of individuals. Nature, 603(7902), 654–660. https://doi.org/10.1038/s41586-022-04492-9

Michelini, G., Barch, D. M., Tian, Y., Watson, D., Klein, D. N., & Kotov, R. (2019). Delineating and validating higher-order dimensions of psychopathology in the Adolescent Brain Cognitive Development (ABCD) study. Translational Psychiatry, 9(1), 261. https://doi.org/10.1038/s41398-019-0593-4

Morris, S. E., & Cuthbert, B. N. (2012). Research Domain Criteria: Cognitive systems, neural circuits, and dimensions of behavior. Dialogues in Clinical Neuroscience, 14(1), 29–37.

Morris, S. E., Sanislow, C. A., Pacheco, J., Vaidyanathan, U., Gordon, J. A., & Cuthbert, B. N. (2022). Revisiting the seven pillars of RDoC. BMC Medicine, 20(1), 220. https://doi.org/10.1186/s12916-022-02414-0

Pat, N., Riglin, L., Anney, R., Wang, Y., Barch, D. M., Thapar, A., & Stringaris, A. (2022). Motivation and Cognitive Abilities as Mediators Between Polygenic Scores and Psychopathology in Children. Journal of the American Academy of Child and Adolescent Psychiatry, 61(6), 782-795.e3. https://doi.org/10.1016/j.jaac.2021.08.019

Pat, N., Wang, Y., Bartonicek, A., Candia, J., & Stringaris, A. (2023). Explainable machine learning approach to predict and explain the relationship between task-based fMRI and individual differences in cognition. Cerebral Cortex, 33(6), 2682–2703. https://doi.org/10.1093/cercor/bhac235

Quah, S. K. L., Jo, B., Geniesse, C., Uddin, L. Q., Mumford, J. A., Barch, D. M., Fair, D. A., Gotlib, I. H., Poldrack, R. A., & Saggar, M. (2025). A data-driven latent variable approach to validating the research domain criteria framework. Nature Communications, 16(1), 830. https://doi.org/10.1038/s41467-025-55831-z

Reef, J., Diamantopoulou, S., van Meurs, I., Verhulst, F., & van der Ende, J. (2010). Predicting adult emotional and behavioral problems from externalizing problem trajectories in a 24-year longitudinal study. European Child & Adolescent Psychiatry, 19(7), 577–585. https://doi.org/10.1007/s00787-010-0088-6

Rodrigue, A. L., Hayes, R. A., Waite, E., Corcoran, M., Glahn, D. C., & Jalbrzikowski, M. (2024). Multimodal Neuroimaging Summary Scores as Neurobiological Markers of Psychosis. Schizophrenia Bulletin, 50(4), 792–803. https://doi.org/10.1093/schbul/sbad149

Roza, S. J., Hofstra, M. B., Van Der Ende, J., & Verhulst, F. C. (2003). Stable Prediction of Mood and Anxiety Disorders Based on Behavioral and Emotional Problems in Childhood: A 14-Year Follow-Up During Childhood, Adolescence, and Young Adulthood. American Journal of Psychiatry, 160(12), 2116–2121. https://doi.org/10.1176/appi.ajp.160.12.2116

Yarkoni, T., & Westfall, J. (2017). Choosing Prediction Over Explanation in Psychology: Lessons From Machine Learning. Perspectives on Psychological Science, 12(6), 1100–1122. https://doi.org/10.1177/1745691617693393